# Provably Convergent Actor-Critic for MARL through Risk-aversion

**Yizhou Zhang** [1]   **Eric Mazumdar** [1]

## Abstract

Learning stationary policies in infinite-horizon general-sum Markov games (MGs) remains a fundamental open problem in Multi-Agent Reinforcement Learning (MARL). While stationary strategies are preferred for their practicality, computing stationary forms of classic game-theoretic equilibria is computationally intractable—a stark contrast to the comparative ease of solving single-agent RL or zero-sum games. To bridge this gap, we study Risk-averse Quantal response Equilibria (RQE), a solution concept rooted in behavioral game theory that incorporates risk aversion and bounded rationality. We demonstrate that RQE possesses strong regularity conditions that make it uniquely amenable to learning in MGs. We propose a novel single-timescale Actor-Critic algorithm characterized by a faster actor and a slower critic. Leveraging the regularity of RQE, we prove that this approach achieves global convergence with finite-sample guarantees. We empirically validate our algorithm in several environments to demonstrate superior convergence properties compared to risk-neutral baselines.

## 1. Introduction

Emerging paradigms in AI are fundamentally multi-agent in nature, requiring agents to interact strategically to achieve their goals. From autonomous driving and robotics to agentic markets, these interactions are often driven by misalignment between agent objectives. While Reinforcement Learning (RL) has become the dominant paradigm for isolated decision-making (Sutton & Barto, 1998), training agents to navigate strategic interactions requires drawing on ideas from *Multi-Agent* Reinforcement Learning (MARL) and game theory more broadly (Littman, 1994; Silver et al., 2016; Vinyals et al., 2019). This stems from the fact

that—unlike single-agent RL, which solves an optimization problem— MARL and game theory seek to solve problems of equilibrium computation (Lanctot et al., 2017).

The canonical framework for studying *dynamic* strategic interactions is the discounted general-sum Markov Game (MG) which dates back to seminal work by Shapley (Shapley, 1953). Despite the longevity of this framework, designing algorithms with provable guarantees of convergence to a meaningful equilibrium remains a largely open problem. Existing guarantees are generally limited to highly structured settings, like two-player zero-sum (Daskalakis et al., 2023) or cooperative games (Monderer & Shapley, 1996). This can be attributed to the fact that *Nash equilibria* are intractable (PPAD-complete) to compute even in general-sum normal-form games (Daskalakis et al., 2009; Chen et al., 2009). Furthermore, computing even stationary generalizations of weaker forms of equilibria like correlated and coarse correlated equilibria (which can be computed in normal-form games) has been shown to be intractable (Jin et al., 2022; Daskalakis et al., 2023). This in turn precludes the possibility that one could design algorithms to learn these strategies efficiently.

Consequently, recent work in MARL has pivoted toward finding non-stationary equilibria (Jin et al., 2024). While computationally tractable, these approaches suffer from crucial drawbacks: they require agents to maintain history-dependent policies whose complexity scales with the time horizon, and they fail to reflect the stationary policies typically employed in practice (Lowe et al., 2017; Samvelyan et al., 2019).

In this paper, we tackle the problem of learning stationary equilibria by adopting a different perspective. We build upon a line of work originating in behavioral game theory that models human decision-making via *strategic* risk aversion and bounded rationality. The resulting solution concept—Risk-Averse Quantal-Response Equilibria (RQE) (Mazumdar et al., 2025)—has recently been shown to be computationally tractable in MGs (Zhang & Mazumdar, 2025). The key insight of these works is that computational tractability can be achieved by making assumptions on agent behavior rather than game structure. In effect the behavioral features regularize the underlying game, rendering it monotone (Rockafellar & Wets, 2009) (a game-theoretic

---

[1]Department of Computing and Mathematical Sciences, California Institute of Technology, Pasadena, CA, USA. Correspondence to: Yizhou Zhang <yzhang8@caltech.edu>.

*Proceedings of the 43rd International Conference on Machine Learning*, Seoul, South Korea. PMLR 306, 2026. Copyright 2026 by the author(s).

analogue to convexity (Cai & Zheng, 2023)) and allowing for the definition of a contractive risk-adjusted Bellman operator.

We extend this line of work to derive a practical MARL algorithm that provably converges to RQE in general-sum MGs. Concretely, our contributions are:

- **Generalized Conditions for RQE Tractability:** We weaken the requirements on players' risk aversion and bounded rationality necessary for the game to possess a unique, computationally tractable RQE. We prove that a broader class of games than those considered in prior work (Mazumdar et al., 2025; Zhang & Mazumdar, 2025) admits an RQE that is unique and varies smoothly with respect to payoff matrices. We use these results to establish that the risk-adjusted Bellman operator is a contraction, which in turn allows us to derive convergent value-based learning methods. Critically, the assumptions under which our theory holds are on the agents' properties (e.g., degrees of risk-aversion and bounded rationality) and *not* on the underlying game. This makes our approach applicable to arbitrary general-sum games, provided we design agents appropriately—a key advantage in practical applications where game structure is fixed but the agent design is under our control.

- **A Provably Convergent Actor-Critic Algorithm:** We design a single-timescale iteration rule that approximates the contraction mapping. In contrast to standard Actor-Critic methods (where critics often update faster than actors), our approach updates the policy (actor) with a larger stepsize and the Q-function (critic) with a smaller stepsize. We provide a novel contraction-based analysis of coupled Lyapunov drift inequalities tailored to the single-timescale stepsizes to prove that this algorithm enjoys finite-sample convergence guarantees. To the best of our knowledge, this is the first MARL algorithm with global guarantees of convergence to stationary equilibria in general-sum discounted MGs that does not assume additional structure on the game.

- **Scalable Implementation and Evaluation:** We further adapt our algorithm to fit modern deep RL infrastructure by employing policy- and Q-networks and a replay buffer. We conduct experiments on three different MARL environments: A normal-form inspection game, a (Markov) gridworld cooperation game and an MPE Simple Tag experiment with fixed good agents. Our results confirm that RQE leads to more stable convergence patterns in the learning process and inherently risk-averse agent behaviors.

## 2. Preliminaries

In this section we present our problem setup. We first define RQE and the notion of monotone games in normal-form games, and then introduce infinite-horizon general-sum Markov game, a setting that extends normal-form games to MARL. Due to space limit, we defer our notations and basic definitions to Appendix A. For our theoretical results, we focus on the two-player case for simplicity, although our results can be easily extended to the $n$-player case.

### 2.1. RQE and Generalized Monotonicity in Normal-form Games

The first and simpler setting we consider in this work is a two-player general-sum bimatrix game where player (agent) $i \in \{1, 2\}$ has payoff matrix $R_i$ and pure strategy set (action set) $\mathcal{A}_i$. When players are risk-neutral, their objective is to maximize the expected utility, expressed as:

$$U_i(\pi_i, \pi_{-i}) = \mathbb{E}_{(a_1, a_2) \sim (\pi_1, \pi_2)}[R_i(a_1, a_2)] = \pi_i^T R_i \pi_{-i}; \quad (1)$$

for $i \in \{1, 2\}$ where $\pi_i \in \Delta_{|\mathcal{A}_i|}$ denotes the mixed strategy (policy) of agent $i$. However, since $\pi_{-i}$ is a mixed strategy, if player $i$ wants to be risk-averse against different possible realizations of the pure strategy $a_{-i}$ selected by $\pi_{-i}$, maximizing the expected utility may not be its desirable objective. Using the framework of convex risk measures (Föllmer & Schied, 2002), player $i$ minimizes a risk measure $\rho_{i, \pi_{-i}}(\mathbb{E}_{\pi_i}[R_i(a_i, a_{-i})])$ associated with $\pi_{-i}$. Leveraging the dual representation theorem for convex risk measures proposed in Föllmer & Schied (2002), prior work (Mazumdar et al., 2025) showed that the objective of player $i$ under risk aversion can be expressed as:

$$\underset{\pi_i}{\text{minimize}} \sup_{p_i \in \Delta_{|\mathcal{A}_{-i}|}} -\pi_i^T R_i p_i - D_i(p_i, \pi_{-i})/\tau_i \quad (2)$$

here $D_i$ is a penalty function as a regularization term in addition to the reward term induced by $R_i$. We can alternatively interpret (2) as follows: Instead of optimizing to play against player $-i$, player $i$ imagines an adversary (we call it adversary $i$), who decides $p_i$ that tries to minimize its expected payoff, while constrained by a term $D_i(p_i, \pi_{-i})$ (representing some distance notion) not to be too far away from the true policy $\pi_{-i}$ of player $-i$. The parameter $\tau_i$ characterizes the degree of risk-aversion of player $i$, where larger $\tau_i$ indicates player $i$ to be more risk-averse by making the adversary less constrained.

To incorporate the bounded rationality behavior in agents' strategies, we restrict their strategies to *quantal responses*, a behavioral economics notion positing that agents make probabilistic decisions that assign higher probabilities to actions with higher utility. Quantal responses can be realized by adding a proper convex regularizer $\nu_i$ to their objective function (Sokota et al., 2023; Mertikopoulos & Sandholm,

2016). Now the objective function for player $i$ becomes (to minimize):

$$f_i(\pi_i, \pi_{-i}; R_i)$$
$$= \sup_{p_i \in \Delta_{|\mathcal{A}_{-i}|}} -\pi_i^T R_i p_i - D_i(p_i, \pi_{-i})/\tau_i + \epsilon_i \nu_i(\pi_i). \quad (3)$$

where $\epsilon_i$ is the temperature that captures the regularization strength of player $i$. Notice that in reinforcement learning, the regularization term $\nu_i$ (often set to be entropy) is already widely used to encourage the policy to be randomized for better exploration. When context is clear, we drop the dependence of $f_i$ on $R_i$ and simply write $f_i(\pi_i, \pi_{-i})$.

Given each player's objective function as in (3), we define the risk-averse quantal response equilibrium (RQE) as the equilibrium point at which each player attains optimality:

**Definition 2.1** (Mazumdar et al., 2025, Definition 5)**.** A risk-averse quantal response equilibrium (RQE) of a two-player general-sum bimatrix game whose payoff matrix is given by $\mathbf{R} = (R_1, R_2)$ is a pair of mixed strategies $\pi^* = (\pi_1^*, \pi_2^*) \in \Delta_{|\mathcal{A}_1|} \times \Delta_{|\mathcal{A}_2|}$ such that

$$f_i(\pi_i^*, \pi_{-i}^*; \mathbf{R}) \le f_i(\pi_i, \pi_{-i}^*; \mathbf{R}), \forall \pi_i \in \Delta_{|\mathcal{A}_i|} \quad (4)$$

for both $i \in \{1, 2\}$. When the RQE is unique, we use $\texttt{RQE}_i$ to denote the value of player $i$ at this equilibrium:

$$\texttt{RQE}_i(\mathbf{R}) := f_i(\pi_i^*, \pi_{-i}^*; \mathbf{R}), \quad i \in \{1, 2\}. \quad (5)$$

Similar to other equilibrium notions like Nash, at an RQE $\pi^*$, no deviation from $\pi_i^*$ leads to a better risk-adjusted objective (3) for each player $i$.

Since the objective function (3) is in the form of a minimax optimization problem, it is hard to directly solve for RQE through (3). To simplify the computation, we treat this objective function from the view of a 4-player game, with 2 original players $i \in \{1, 2\}$ deciding $\pi_i \in \Delta_{|\mathcal{A}_i|}$ minimizing:

$$J_i(\pi_i, \pi_{-i}, p_i; R_i) = -\pi_i^T R_i p_i - D_i(p_i, \pi_{-i})/\tau_i + \epsilon_i \nu_i(\pi_i); \quad (6a)$$

and two adversaries deciding $p_i \in \Delta_{|\mathcal{A}_{-i}|}$ and minimizing:

$$\bar{J}_i(\pi_i, \pi_{-i}, p_i; R_i) = \pi_i^T R_i p_i + D_i(p_i, \pi_{-i})/\tau_i - \epsilon_i \nu_i(\pi_i). \quad (6b)$$

It is proven that the Nash equilibria of the 4-player game can be connected to the RQE in the original 2-player game in the following way:

**Proposition 2.2** (Mazumdar et al., 2025, Proposition 1)**.** *Let $z^* = (\pi^*, p^*)$ be a Nash equilibrium of the 4-player game characterized by (6a) and (6b). We have that $\pi^*$ is an RQE of the original two-player game characterized by (3). Furthermore, if $\pi^*$ is an RQE of the two-player game, then $(\pi^*, p^*)$ is a Nash equilibrium of the 4-player game where $p_i^* = \arg\max_{p_i \in \Delta_{|\mathcal{A}_{-i}|}} -\pi_i^* R_i p_i - D_i(p_i, \pi_{-i}^*)/\tau_i$.*

Proposition 2.2 implies that once we find a Nash equilibrium for the 4-player game, we also obtain an RQE of the original two-player game simply by taking its $\pi$ component.

Indeed, since $\bar{J}_i = -J_i$, the introduction of adversaries creates a partial zero-sum structure in each player-adversary pair, and relaxes the exact dependence of player $i$'s utility on $\pi_{-i}$ in (1) to the regularized adversary policy $p_i$ through $D_i(p_i, \pi_{-i})$. This structure is illustrated in Figure 1. While at first glance, elevating the original two-player game to 4 players makes the problem more complicated, it turns out to simplify equilibrium computation to a great extent.

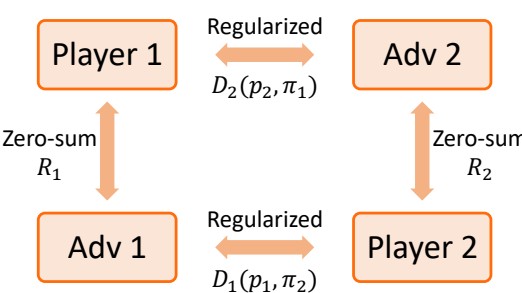

*Figure 1.* Illustration of the 4-player game structure.

Based on the intuition above, Zhang & Mazumdar (2025) studied the properties of Nash equilibria for the 4-player game through its *monotonicity*. In this work, we consider a generalized monotonicity notion originally referred to as *diagonally (strictly) concave* in Rosen (1965) as follows:

**Definition 2.3.** Let $\mathcal{Z}$ be a subset of $\mathbb{R}^n$ and $\lambda \in \mathbb{R}^n$ satisfying $\lambda > 0$. An operator $F : \mathcal{Z} \to \mathbb{R}^n$ is $\lambda$-monotone if:

$$\langle z - z', F(z) - F(z') \rangle_\lambda \ge 0, \forall z, z' \in \mathcal{Z},$$

it is $\lambda$-strictly monotone when the inequality is strict when $z \ne z'$, and is $(\mu, \lambda)$-strongly monotone if:

$$\langle z - z', F(z) - F(z') \rangle_\lambda \ge \mu \|z - z'\|_2^2, \forall z, z' \in \mathcal{Z}.$$

An $N$-player game where player $i$ chooses action $z_i$ from a compact and convex action space $\mathcal{Z}_i \subseteq \Delta_{|\mathcal{A}_i|}$ with cost function $J_i(z_i, z_{-i})$ is a $\lambda$-monotone (resp. $\lambda$-strictly monotone, $(\mu, \lambda)$-strongly monotone) game if its gradient operator $F : \mathcal{Z} \to \mathbb{R}^{\sum_{i \in [N]} |\mathcal{A}_i|}$ where $\mathcal{Z} = \prod_{i=1}^N \mathcal{Z}_i$ defined by $F_i(z) = \nabla_{z_i} J_i(z_i, z_{-i}) \in \mathbb{R}^{|\mathcal{A}_i|}$ is a $\lambda$-monotone (resp. $\lambda$-strictly monotone, $(\mu, \lambda)$-strongly monotone) operator.

Definition 2.3 generalizes the monotonicity definition in Zhang & Mazumdar (2025) through introducing a weight vector $\lambda$ that controls the weight of each direction in the decision space $\mathcal{Z}$. Monotone games are known to satisfy many desirable properties. For example, for a $\lambda$-strictly monotone game, there exists a unique Nash equilibrium. If the game is further $(\mu, \lambda)$-strongly monotone for some

$\lambda > 0$, projected preconditioned gradient descent $z_{t+1} = \text{Proj}_{\mathcal{Z}}(z_t - \eta \Lambda F(z_t))$ where $\Lambda = \text{Diag}(\lambda)$ is a diagonal matrix with diagonal entries equal $\lambda$ converges linearly to the Nash equilibrium. As we show later in Theorems 3.1 and 3.2, monotonicity brings many desirable properties to RQE, and the condition for the 4-player game to be monotone does not depend on players' payoff matrices.

## 2.2. RQE in Discounted Infinite-horizon Markov Games

In this section we generalize the RQE notion from normal form games to Markov games. A discounted two-player infinite-horizon general-sum Markov game is specified by a tuple $\mathcal{MG} = \{\mathcal{S}, \{\mathcal{A}_i\}_{i=1,2}, \{r_i\}_{i=1,2}, \gamma, P, \rho_0\}$ where $\mathcal{S}$ is the state space of the underlying MDP, $\mathcal{A}_i$ is the action space of player $i \in \{1, 2\}$, and we use the notation $\mathcal{A} = \mathcal{A}_1 \times \mathcal{A}_2$ to denote the product action space of both players. We assume $|\mathcal{S}|$ and $|\mathcal{A}_1|, |\mathcal{A}_2|$ to be finite. $r_i : \mathcal{S} \times \mathcal{A} \to [0, 1]$ is the reward function of player $i$, which we assume to be deterministic. We use $\mathbf{r}$ to denote the paired reward function $\mathbf{r} := (r_1, r_2)$. $\gamma \in [0, 1)$ is the discount factor and $P : \mathcal{S} \times \mathcal{A} \to \Delta_{\mathcal{S}}$ is the transition kernel, where $P(s'|s, \mathbf{a})$ is the probability of the next state being $s'$ given the current state $s$ and the current actions $\mathbf{a} = (a_1, a_2)$ of the players. We use $\rho_0 \in \Delta_{\mathcal{S}}$ to denote the initial state distribution.

We focus on *Markov policies*, the class of policies where the action selection probability only depends on the current state instead of the entire gameplay trajectory, i.e. $\pi = (\pi_1, \pi_2)$ where $\pi_i : \mathcal{S} \to \Delta_{|\mathcal{A}_i|}, i \in \{1, 2\}$. Given a product Markov policy $\pi$, without considering risk-aversion and bounded rationality, player $i$ has an expected discounted cumulative reward given by $\mathbb{E}_\pi[\sum_{t=0}^\infty \gamma^t r_i(s_t, a_t)]$.

To incorporate risk-aversion in discounted infinite-horizon Markov games, we slightly overload the notations in normal-form games and consider the following risk-adjusted objective of player $i$ that minimizes $f_i(\pi_i, \pi_{-i}) = \max_{p_i : \mathcal{S} \to \Delta_{|\mathcal{A}_{-i}|}} J_i(\pi_i, \pi_{-i}, p_i)$ where $J_i$ is defined as:

$$J_i(\pi_i, \pi_{-i}, p_i) = \mathbb{E}_{\pi, p, s_0 \sim \rho_0}\big[\sum_{t=0}^\infty \gamma^t \big( - r_i(s_t, \mathbf{a}_t) - D_i(p_i, \pi_{-i}; s_t)/\tau_i + \epsilon_i \nu_i(\pi_i; s_t)\big)\big] \quad (7)$$

where the joint actions $\mathbf{a}_t$ are sampled through $a_{i,t} \sim \pi_i(\cdot|s_t)$, $\mathbf{a}_{-i,t} \sim p_i(\cdot|s_t)$ and the next state $s_{t+1}$ is sampled from $s_{t+1} \sim P(\cdot|s_t, \mathbf{a}_t)$. The notations of $D_i(p_i, \pi_{-i}; s)$ and $\nu_i(\pi_i; s)$ are abbreviations of $D_i(p_i(\cdot|s), \pi_{-i}(\cdot|s))$ and $\nu_i(\pi_i(\cdot|s))$ respectively. Starting from now, we refer to the Markov game objective above when we don't include payoff matrices $\mathbf{R}$ in the argument of $J_i$ and refer to the normal form game objective (6a) when $\mathbf{R}$ are included. This adjusted objective can be viewed as each player playing against its imaginary adversary who controls $p_i$ that tries to minimize its discounted cumulative reward but constrained by a penalty term $D_i(\cdot, \cdot; s)$ at each possible state $s \in \mathcal{S}$ from $\pi_{-i}$ for all subsequent time steps. Given a set of origi-

nal player policies $\pi_i$ and adversarial policies $p_i$, we define the value function for each state $s \in \mathcal{S}$ as:

$$V_i^{\pi,p}(s) = \mathbb{E}_{\pi,p,s_0=s}\big[\sum_{t=0}^\infty \gamma^t\big( - r_i(s_t, \mathbf{a}_t) - D_i(p_i, \pi_{-i}; s_t)/\tau_i + \epsilon_i \nu_i(\pi_i; s_t)\big)\big] \quad (8)$$

so that $J_i(\pi, p) = \mathbb{E}_{s \sim \rho_0}[V_i^{\pi,p}(s)]$, and the $Q$ function as:

$$Q_i^{\pi,p}(s, \mathbf{a}) = -r_i(s, \mathbf{a}) + \gamma \mathbb{E}_{s' \sim P(\cdot|s, \mathbf{a})} V_i^{\pi,p}(s'). \quad (9)$$

Additionally, we use $f_i(\pi_i, \pi_{-i}; s)$ to denote the quantity $\max_{p_i : \mathcal{S} \to \Delta_{|\mathcal{A}_{-i}|}} V_i^{\pi,p}(s)$, so that $\mathbb{E}_{s \sim \rho}[f_i(\pi_i, \pi_{-i}; s)] = f_i(\pi_i, \pi_{-i})$. Notice that given the state $s$, we can view $Q_i^{\pi,p}(s, \cdot)$ as a payoff matrix of a normal form game, with associated strategies $\pi_i$ and $p_i$, it is easy to verify that:

$$V_i^{\pi,p}(s) = \pi_i(\cdot|s)^T Q_i^{\pi,p}(s, \cdot) p_i(\cdot|s) - D_i(p_i, \pi_{-i}; s)/\tau_i + \epsilon_i \nu_i(\pi_i; s). \quad (10)$$

This provides the connection between value function and the objective for the 4-player stage-game:

$$V_i^{\pi,p}(s) = J_i\left(\pi_i(\cdot|s), \pi_{-i}(\cdot|s), p_i(\cdot|s); -\mathbf{Q}^{\pi,p}(s, \cdot)\right). \quad (11)$$

Here $J_i$ denotes the objective function (6a) for normal form games, where we regard $-\mathbf{Q}^{\pi,p}(s, \cdot)$ as the payoff matrices. We extend the notion of RQE to Markov games as follows:

**Definition 2.4** (Stationary Markov RQE). A pair of Markov policies $\pi^* = (\pi_1^*, \pi_2^*)$ where $\pi_i^* : \mathcal{S} \to \Delta_{|\mathcal{A}_i|}$ is said to be an RQE of a two-player Markov game $\mathcal{MG}$ if for both $i \in \{1, 2\}$:

$$f_i(\pi_i^*, \pi_{-i}^*; s) \leq f_i(\pi_i, \pi_{-i}^*; s), \forall s \in \mathcal{S}, \pi_i : \mathcal{S} \to \Delta_{|\mathcal{A}_i|}. \quad (12)$$

Definition 2.4 is a simpler version of RQE than that in Zhang & Mazumdar (2025) which additionally considered risk-aversion against the potential stochasticity of the environment. Despite this simplification, we expect most of our results could be easily translated to their notion. We also note that unlike the widely studied CE or CCE, RQE policies are Markovian and can be executed independently.

To characterize the computation of RQE in discounted Markov games, assume that the stage game given a $Q$ function pair $\mathbf{Q}$ has a unique RQE, we define the risk-averse quantal-response Bellman operators as follows:

**Definition 2.5.** Given a two-player discounted Markov game $\mathcal{MG}$, risk-aversion penalty functions $D_i(\cdot, \cdot)$ and regularizers $\nu_i(\cdot)$ where $i \in \{1, 2\}$, the risk-averse quantal-response Bellman optimality operator $\mathcal{T}$ maps a $Q$ function pair $\mathbf{Q} = (Q_1, Q_2)$ where $Q_i : \mathcal{S} \times \mathcal{A} \to \mathbb{R}$ to another $Q$ function pair $\mathcal{T}\mathbf{Q}$ in the same function space, defined elementwise as:

$$(\mathcal{T}\mathbf{Q})_i(s, \mathbf{a}) = -r_i(s, \mathbf{a}) + \gamma \mathbb{E}_{s' \sim P(\cdot|s, \mathbf{a})}[\text{RQE}_i(-\mathbf{Q}(s', \cdot))] \quad (13)$$

here we view $-\mathbf{Q}(s',\cdot)$ as a pair of payoff matrices in some normal form game with action space $\mathcal{A} = \mathcal{A}_1 \times \mathcal{A}_2$. Similarly, for a joint policy profile $z = (\pi, p)$, we define the risk-averse quantal-response Bellman evaluation operator with respect to $z$ as:

$$(\mathcal{T}_z \mathbf{Q})_i(s, \mathbf{a}) = -r_i(s, \mathbf{a}) + \gamma \mathbb{E}_{s' \sim P(\cdot|s,\mathbf{a})}[\epsilon_i \nu_i(\pi_i; s') +$$
$$\pi_i^T(\cdot|s')Q_i(s',\cdot)p_i(\cdot|s') - D_i(p_i, \pi_{-i}; s')/\tau_i] \quad (14)$$

also elementwise for $i \in \{1, 2\}$.

When context is clear, we will simply use "Bellman optimality operator" (or simply "Bellman operator") and "Bellman evaluation operator" to refer to the operators defined in (13) and (14) respectively. Notice that the Bellman optimality operator is adapted from Definition 4.1 in Zhang & Mazumdar (2025) through removing the risk-aversion to the environment, and the Bellman evaluation operator can be viewed as evaluating the $Q$ function (9) corresponding to $z$.

The intuition behind the definition of Bellman operators is that at state $s$, when deciding the first action, each agent is faced with a *"stage game"* as a normal form game with payoff matrices $-\mathbf{Q}(s,\cdot)$, computed by adding up the immediate reward plus a discounted RQE value it can get by applying its current policy, as the regularized risk-averse version of that originally proposed in the Nash Q-learning algorithm by Hu & Wellman (2003). It can be shown (details in Appendix E) that the following property connecting the Bellman operator and the RQE of Markov games holds:

**Proposition 2.6.** *Let $\mathbf{Q}^*$ be a fixed point of the Bellman optimality operator $\mathcal{T}$, the policy profile $\pi^* = (\pi_1^*, \pi_2^*)$ where $\pi_i^* : \mathcal{S} \to \Delta_{|\mathcal{A}_i|}$ given by:*

$$\pi_i^*(\cdot|s) = \arg\min_{\pi_i \in \Delta_{|\mathcal{A}_i|}} \max_{p_i \in \Delta_{|\mathcal{A}_{-i}|}} \pi_i^T Q_i^*(s,\cdot)p_i$$
$$- D_i(p_i, \pi_{-i}(\cdot|s))/\tau_i + \epsilon_i \nu_i(\pi_i) \quad (15)$$

*is an RQE of the Markov game. Additionally, let $\pi^* = (\pi_1^*, \pi_2^*)$ be an RQE of the Markov game, then for $p^* = (p_1^*, p_2^*)$ where $p_i^* = \arg\max_{p_i:\mathcal{S}\to\Delta_{|\mathcal{A}_{-i}|}} V_i^{\pi^*,p}(s), \forall s \in \mathcal{S}$, the associated $Q$ function $\mathbf{Q}^{\pi^*,p^*}$ is a fixed point of the Bellman optimality operator $\mathcal{T}$.*

Proposition 2.6 suggests that solving an RQE and its corresponding $Q$ functions can be reduced to finding a fixed point of the Bellman optimality operator $\mathcal{T}$. This, combined with the contraction property of $\mathcal{T}$ shown in Proposition 4.2, gives us a practical way of solving RQE in Markov games.

We note that the first statement in Proposition 2.6 is stated in Proposition 4.3 (under stronger assumptions) of Zhang & Mazumdar (2025) but not formally proved. Proposition 2.6 generalizes that result by stating that the policy $\pi^*$ is an RQE if and only if its associated $\mathbf{Q}$ function is a fixed point of the Bellman optimality operator $\mathcal{T}$.

## 3. Results for Normal-form Games

In this section, we provide our results for normal-form games. We first provide a result suggesting uniqueness and Lipschitz continuity (with respect to payoff matrices) of RQE under $\lambda$-monotonicity (Definition 2.3) of the 4-player game, and then provide conditions for the game to be $\lambda$-monotone for some $\lambda$. The results presented in this section generalize those in Zhang & Mazumdar (2025).

Recall the 4-player game view and the objective functions (6), let $z = (\pi, p) = (\pi_1, \pi_2, p_1, p_2)$ denote the joint strategy of all players (2 original players and 2 adversaries), the gradient operator of the 4-player game is:

$$F(z) = \begin{bmatrix} \nabla_{\pi_1} J_1 \\ \nabla_{\pi_2} J_2 \\ \nabla_{p_1} \bar{J}_1 \\ \nabla_{p_2} \bar{J}_2 \end{bmatrix} = \begin{bmatrix} -R_1 p_1 + \epsilon_1 \nabla \nu_1(\pi_1) \\ -R_2 p_2 + \epsilon_2 \nabla \nu_2(\pi_2) \\ R_1^T \pi_1 + \frac{1}{\tau_1} \nabla_p D_1(p_1, \pi_2) \\ R_2^T \pi_2 + \frac{1}{\tau_2} \nabla_p D_2(p_2, \pi_1) \end{bmatrix} \quad (16)$$

Our first result, stated in Theorem 3.1, captures uniqueness and Lipschitz continuity of RQE with respect to the payoff matrices, whose proof can be found in Appendix F.1.

**Theorem 3.1.** *Suppose $\lambda > 0$ can be written as $\lambda = (\lambda_1 \mathbf{1}_{|\mathcal{A}_1|}, \lambda_2 \mathbf{1}_{|\mathcal{A}_2|}, \lambda_1 \mathbf{1}_{|\mathcal{A}_2|}, \lambda_2 \mathbf{1}_{|\mathcal{A}_1|})^T$, we have:*

1. *If the 4-player game (6) is $\lambda$-strictly monotone, the RQE of the original two-player game is unique.*

2. *If the 4-player game is $(\mu, \lambda)$-strongly monotone, for two different pairs of payoff matrices $\mathbf{R}$ and $\mathbf{R}'$, their corresponding Nash equilibria $z^* = (\pi_1^*, \pi_2^*, p_1^*, p_2^*)$ and $z^\dagger = (\pi_1^\dagger, \pi_2^\dagger, p_1^\dagger, p_2^\dagger)$ satisfy:*

$$\|z^* - z^\dagger\|_2 \leq \frac{2\|\lambda\|_\infty \left(\sqrt{|\mathcal{A}_1|} + \sqrt{|\mathcal{A}_2|}\right)}{\mu} \|\mathbf{R} - \mathbf{R}'\|_{\max}.$$

*As a result, the corresponding RQEs of the original 2-player game satisfy:*

$$\|\pi^* - \pi^\dagger\|_2 \leq \frac{2\|\lambda\|_\infty \left(\sqrt{|\mathcal{A}_1|} + \sqrt{|\mathcal{A}_2|}\right)}{\mu} \|\mathbf{R} - \mathbf{R}'\|_{\max}.$$

Theorem 3.1 generalizes Proposition 3.2 and Theorem 3.3 in Zhang & Mazumdar (2025) through introducing a weight vector $\lambda$ that weighs the components of gradient operator $F$ for player $i$ by $\lambda_i$. When taking $\lambda_1 = \lambda_2 = 1$, we recover the strong monotonicity condition used by Zhang & Mazumdar (2025) that controls the distance between $z^*$ and $z^\dagger$ in $L_2$ norm.

Now we give the conditions for the game to be $\lambda$-strictly / $(\mu, \lambda)$-strongly monotone:

**Theorem 3.2.** *We have the following regarding the monotonicity conditions for the 4-player game:*

1. Let $M_i(\lambda, z)$ denote the following matrix:

$$\begin{bmatrix} 2\lambda_i \epsilon_i \nabla^2 \nu_i(z) & \frac{\lambda_{-i}}{\tau_{-i}} \nabla^2_{p\pi} D_{-i}(z) \\ \frac{\lambda_{-i}}{\tau_{-i}} \nabla^2_{p\pi} D_{-i}(z) & 2\frac{\lambda_{-i}}{\tau_{-i}} \nabla^2_p D_{-i}(z) \end{bmatrix},$$

then the game (6) is $\lambda$-strictly monotone if for all $z$, $M_i(\lambda, z) \succeq 0, i \in \{1, 2\}$, and is $(\mu, \lambda)$-strongly monotone if and only if $\forall z, M_i(\lambda, z) \succeq 2\mu I, i \in \{1, 2\}$.

2. If $\nu_i(\cdot)$ are log-barrier functions and $D_i(\cdot, \cdot)$ are KL-divergences, then the game is $(\mu, \lambda)$-strongly monotone for some $\mu, \lambda > 0$ as long as $16\epsilon_1\epsilon_2\tau_1\tau_2 > 1$.

3. If $\nu_i(\cdot)$ are negative entropy and $D_i(\cdot, \cdot)$ are reverse KL-divergences, then the game is $\lambda$-strictly monotone as long as $16\epsilon_1\epsilon_2\tau_1\tau_2 > 1$.

The proof is deferred to Appendix F.2. As stated in Theorem 3.2, whether the 4-player game is monotone or not does not depend on the payoff matrices of the game, but only depends on the Hessians of the regularizers $\nu_i, D_i$. More specifically, if the regularizers are taken to be KL/log-barrier or reverse KL/negative entropy, the condition for monotonicity simplifies to $16\epsilon_1\epsilon_2\tau_1\tau_2 > 1$. Notice that when the regularizers are reverse KL/negative entropy, the game can only be strictly monotone (but not strongly monotone) because the Hessian of reverse KL may not be strongly convex even in the interior of the simplex. When the regularizer pairs are either KL/log-barrier or reverse KL/negative entropy, we compare the regions where the uniqueness of RQE is guaranteed given by Theorem 3.2 to that in Mazumdar et al. (2025) and Zhang & Mazumdar (2025) in Figure 2:

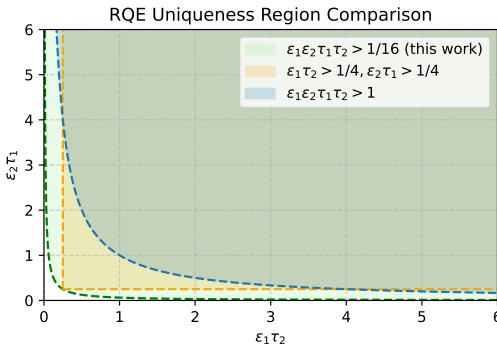

RQE Uniqueness Region Comparison

Legend:
- $\epsilon_1\epsilon_2\tau_1\tau_2 > 1/16$ (this work)
- $\epsilon_1\tau_2 > 1/4, \epsilon_2\tau_1 > 1/4$
- $\epsilon_1\epsilon_2\tau_1\tau_2 > 1$

*Figure 2.* RQE uniqueness region for KL/log-barrier or reverse KL/negative entropy. Green captures the region indicated by Theorems 3.1 and 3.2. Orange captures that in Zhang & Mazumdar (2025) and blue captures that in Mazumdar et al. (2025).

As shown in Figure 2, if we fix $\epsilon_{-i}$ and $\tau_{-i}$, the minimum $\tau_i$ scales inversely proportionally in $\epsilon_i$. This can be seen as a smooth interpolation between two computationally tractable cases: $\tau_i \to \infty$, where the optimal policy $\pi_i$ is decoupled with $\pi_{-i}$ as $D_i(p_i, \pi_{-i})/\tau_i \to 0$, and $\epsilon_i \to \infty$,

where the term $\epsilon_i\nu_i(\pi_i)$ dominates and therefore $\pi_i^*$ is fixed. Compared with prior work, Theorems 3.1 and 3.2 require strictly weaker conditions on the levels of risk-aversion and bounded rationality, as Mazumdar et al. (2025) essentially takes a *social convexity* (Even-dar et al., 2009) approach, which is stronger than the $\lambda$-monotonicity that we use in Definition 2.3. The conditions provided in Zhang & Mazumdar (2025) didn't introduce $\lambda$ and are equivalent to setting $\lambda = 1$ in our setting.

# 4. Results for Markov Games and MARL

In this section we shift our focus to the harder problem of discounted general-sum Markov games. We first prove the contraction property of the Bellman operator, and then use it to design a provably convergent coupled iteration rule with faster actor and slower critic. Finally, we propose an actor-critic algorithm that provably learns the RQE in finite sample through interaction with the environment. Throughout this section, we assume:

**Assumption 4.1.** Regarding the 4-player stage game (11), there exists $\lambda = (\lambda_1 \mathbf{1}_{|\mathcal{A}_1|}, \lambda_2 \mathbf{1}_{|\mathcal{A}_2|}, \lambda_1 \mathbf{1}_{|\mathcal{A}_2|}, \lambda_2 \mathbf{1}_{|\mathcal{A}_1|})^T > 0$ for it to be $(\mu, \lambda)$-strongly monotone for every $s \in \mathcal{S}$.

Notice that the monotonicity of a stage game depends *only on the level of risk-aversion and bounded rationality* of the players, not on the property of the original MG. Additionally, since the condition in Mazumdar et al. (2025) is stronger than ours, Assumption 4.1 is realistic, and it captures real-world human behaviors as shown in Mazumdar et al. (2025).

## 4.1. Contraction of Bellman Operator

In this section we adapt and generalize Theorem 4.2 in Zhang & Mazumdar (2025) to our setting as follows, whose detailed version and proof can be found in Appendix G.1:

**Proposition 4.2.** *Under Assumption 4.1, when either of the following cases hold: (i) The regularizers $D_i(\cdot, \cdot)$ are $L_D$-Lipschitz metrics that satisfy triangle inequality; (ii) If $D_i(\cdot, \cdot)$ are KL-divergence and $\nu_i(\cdot)$ are log-barrier functions, and the $Q$ functions are bounded, if $\gamma$ is smaller than some threshold, there exists $\gamma_0 < 1$ such that the Bellman optimality operator $\mathcal{T}$ is a $\gamma_0$-contraction mapping.*

Proposition 4.2 generalizes Theorem 4.2 in Zhang & Mazumdar (2025) in two ways: First, it assumes $(\mu, \lambda)$-strong monotonicity, while the assumption is stronger in Zhang & Mazumdar (2025), requiring $\lambda = 1$. Second, it covers the case for KL/log-barrier regularizer pair, while Theorem 4.2 in Zhang & Mazumdar (2025) only works when the regularizer $D_i(\cdot, \cdot)$ are Lipschitz metrics.

## 4.2. Coupled Iteration Rule for Markov Games

In principle, given the contraction property of the Bellman operator $\mathcal{T}$, if we start from some bounded $Q$ function $\mathbf{Q}_0$ and iteratively apply $\mathcal{T}$ to it, or more generally conduct the value iteration rule $\mathbf{Q}_{t+1} = (1 - \alpha_t)\mathbf{Q}_t + \alpha_t\mathcal{T}\mathbf{Q}_t$ as stated in Corollary 4.4 in Zhang & Mazumdar (2025) for some step size sequence $\{\alpha_t\}_{t=0}^T > 0$, we know that $\mathbf{Q}_t$ converges to the unique fixed point $\mathbf{Q}^*$ of $\mathcal{T}$ and as suggested by Proposition 2.6, the corresponding policies $\pi^*$ obtained by (15) is an RQE of the Markov game. However, applying $\mathcal{T}$ requires an oracle of $\mathrm{RQE}_i(\cdot)$ that computes the RQE value of agent $i$ given some $Q$ function $\mathbf{Q}$. This prevents us from directly conducting value iteration, and we have to design another practical algorithm that converges to RQE without directly computing $\mathrm{RQE}_i(\cdot)$. To design that algorithm, we make a smoothness assumption on the regularizers:

**Assumption 4.3.** The regularizers $D_i(\cdot, \cdot)$ and $\nu_i(\cdot)$ are both $S$-smooth functions for some $S > 0$.

Assumption 4.3 is a standard assumption in optimization theory and RL. When the regularizers are KL/log-barrier, although they are not smooth on the entire simplex, we can modify the projection step in the $z$ update in (17) to project onto a subset of $\mathcal{Z}$ where policies are uniformly lower-bounded, on which both KL and log-barrier are smooth.

Under Assumptions 4.1 and 4.3, if we have a fixed $\mathbf{Q}$ function, iteratively applying preconditioned GD $z_{t+1} = \mathrm{Proj}_{\mathcal{Z}}(z_t - \eta\Lambda F(z_t))$ on the joint policy $z$ of all players, it holds that $z$ converges to the unique Nash equilibrium of the 4-player game, which by Proposition 2.2 has its $\pi$ component converging to the RQE of the two-player game, corresponding to the stage game given $\mathbf{Q}$. This can be seen as an approximation of $\mathrm{RQE}_i(\cdot)$ with some error.

In light of this rationale, we propose a coupled iteration process regarding the 4-player game with two original agents and two adversaries, that provably converges to the RQE of the Markov game as follows:

$$
\begin{aligned}
z_{t+1}(\cdot|s) &\leftarrow \mathrm{Proj}_{\mathcal{Z}}\left(z_t(\cdot|s) - \beta_t\Lambda F(z_t; -\mathbf{Q}_t)(s)\right), \forall s; \\
\mathbf{Q}_{t+1} &\leftarrow (1 - \alpha_t)\mathbf{Q}_t + \alpha_t\mathcal{T}_{z_{t+1}}\mathbf{Q}_t.
\end{aligned}
\tag{17}
$$

where we initialize $\mathbf{Q}_0 = 0$ and $z_0$ to be uniform policies. Here $\alpha_t, \beta_t$ are step sizes (with $\alpha_t \ll \beta_t$) and $\mathrm{Proj}_{\mathcal{Z}}$ is the projection operator that projects each component $(\pi_{1,t}, \pi_{2,t}, p_{1,t}, p_{2,t})$ onto their respective simplexes. In each iterate, we first conduct a projected preconditioned GD step to obtain $z_{t+1}$ using the current gradient operator $F(z_t; -\mathbf{Q}_t)$, followed by a soft $Q$-update to drive $Q_{t+1}$ closer to the true $Q$ function of $z_{t+1}$. Iteration rule (17) can be interpreted as follows: If we conduct many $z$ gradient steps before one $Q$-iteration step, it recovers the risk-averse version of Nash $Q$-Learning (Hu & Wellman, 2003) $\mathbf{Q}_{t+1} \leftarrow (1 - \alpha_t)\mathbf{Q}_t + \alpha_t\mathcal{T}\mathbf{Q}_t$ with an approximation oracle

of RQE instead of Nash equilibrium. If we perform many $Q$-iteration steps before one $z$ gradient step, it recovers policy gradient methods.

The convergence guarantee for the iteration rule (17) is stated in Theorem 4.4, whose detailed version and proof can be found in Appendix G.2.

**Theorem 4.4.** *Under Assumptions 4.1 and 4.3 and assume $\mathcal{T}$ is a $\gamma_0$-contraction mapping, let $z^*$ be the Nash equilibrium of the 4-player game, if the step sizes satisfy $\alpha_t \ll \beta_t \ll 1$, the iteration rule (17) satisfies:*

1. *For constant step size $\alpha_t = \alpha, \beta_t = \beta$, there exists a constant $D_1$ such that $\|z_t(\cdot|s) - z^*(\cdot|s)\|_2 \leq D_1\left(1 - \frac{1-\gamma_0}{2}\alpha\right)^t, \forall s \in \mathcal{S}$.*

2. *For diminishing step size $\alpha_t = \frac{\alpha}{t+h}, \beta_t = \frac{\beta}{t+h}$, there exists a constant $D_2$ such that $\|z_t(\cdot|s) - z^*(\cdot|s)\|_2 \leq D_2\left(\frac{h}{h+t+1}\right)^{\frac{1-\gamma_0}{2}\alpha}, \forall s \in \mathcal{S}$.*

*As a result, the $\pi$ component of $z$ converges to the RQE of the original 2-player game at the same rate.*

Theorem 4.4 suggests that when we use constant step sizes, the iterates of (17) converges to the RQE at a linear rate, and if we use diminishing step sizes $\mathcal{O}(1/t)$, the convergence rate becomes sublinear. Crucially, in contrast to standard policy gradient methods, Theorem 4.4 requires the policy step size $\beta_t$ be much larger than the $Q$ function step size $\alpha_t$ in order to use the property that $\mathcal{T}$ is a contraction mapping, which is satisfied under either case in Proposition 4.2.

## 4.3. Convergent Actor-Critic through Risk-aversion

In the previous part we proved that under $(\mu, \lambda)$-strong monotonicity assumption of the 4-player game, the iteration rule (17) provably converges to the RQE of the Markov game. Following this, we design an actor-critic style MARL algorithm (Algorithm 1 in Appendix C) that learns the $Q$ function through interacting with the environment and provably converge to the RQE of the game. The main difference between Algorithm 1 and (17) is that it conducts $Q$ iteration with stochastic approximation through samples instead of directly applying $\mathcal{T}_{z_{t+1}}$ which requires knowing the transition matrix. Thanks to its actor-critic nature, Algorithm 1 supports both *on-policy* and *off-policy* training, where the only difference is that for on-policy, we use current policies $\pi$ to sample transition data, while for off-policy, transitions are sampled using a fixed reference policy $\pi^r$. To guarantee the Markov game can be sufficiently explored, we make Assumptions 4.5 and 4.6, both being common assumptions in modern stochastic approximation and RL literature:

**Assumption 4.5.** For all joint policy $\pi$, the underlying Markov chain induced by transition kernel $P_\pi$ is irreducible

and uniformly geometrically ergodic. That is, let $P_\pi^t(s, \cdot)$ denote the state distribution at time step $t$ with initial state being $s$, there exists a unique state distribution $\mu_\pi(\cdot)$ uniformly lower-bounded by $\underline{\mu}$, such that $\|P_\pi^t(s, \cdot) - \mu_\pi(\cdot)\|_{\mathrm{TV}} \leq C\rho^t, \forall s \in \mathcal{S}$ for some constants $C < \infty$ and $\rho \in (0, 1)$.

**Assumption 4.6.** For all $t, s \in \mathcal{S}$ and $a \in \mathcal{A}$, the policies used for sampling are uniformly lower bounded by $\underline{\pi}$.

We now present Theorem 4.7, a finite-sample convergence result for Algorithm 1 as follows:

**Theorem 4.7.** *Under Assumptions 4.1, 4.3, 4.5 and 4.6, assume $\mathcal{T}$ is a $\gamma_0$-contraction mapping, let $z^*$ be the Nash equilibrium of the risk-adjusted 4-player game (and correspondingly $\pi^*$ the RQE of the original game), if the step sizes satisfy $\alpha_t \ll \beta_t \ll 1$, then for both on- and off-policy variants, let $\underline{d} = \underline{\mu}\underline{\pi}$, the iterates of Algorithm 1 satisfy:*

1. *For constant step size $\alpha_t = \alpha, \beta_t = \beta$, there exist constants $D_3, D_4$ such that for all $s \in \mathcal{S}$:*

$$\mathbb{E}[\|z_t(\cdot|s) - z^*(\cdot|s)\|_2^2] \leq (1 - \frac{(1-\gamma_0)}{2}\underline{d}\alpha)^t D_3 + \alpha D_4.$$

2. *For diminishing step size $\alpha_t = \frac{\alpha}{t+h}, \beta_t = \frac{\beta}{t+h}$, there exists constant $D_5$ such that for all $s \in \mathcal{S}$:*

$$\mathbb{E}[\|z_t(\cdot|s) - z^*(\cdot|s)\|_2^2] \leq D_5 \left(\frac{h+1}{h+t}\right)^{\frac{(1-\gamma_0)}{2}\underline{d}\alpha}.$$

A detailed version of Theorem 4.7 and its proof can be found in Appendix H. The proof involves constructing and solving a novel coupled Lyapunov drift inequality for both the policies $z_t$ and the $Q$ functions $\mathbf{Q}_t$. The main technical challenge is that, instead of relying on the policy gradient step to provide a negative drift (which no longer holds in multi-agent game-theoretic setting), we rely on the contraction of Bellman operator $\mathcal{T}$ to yield a negative drift, which is only possible with faster actor and slower critic, compared to the usual slower actor and faster critic regime used in existing analyses of actor-critic algorithms.

## 5. Experiments

In this section we conduct numerical experiments to showcase the effectiveness of risk-aversion in normal-form games and MARL. We first provide Algorithm 2, a scalable implementation of Algorithm 1 that employs policy/Q networks and a replay buffer in Appendix C.1. To emphasize how risk-aversion helps training compared to risk-neutral, we compare the risk-averse version of Algorithm 2 against its risk-neutral version in our MARL experiments. We note that although some of our experiment configurations do not satisfy the RQE monotonicity condition stated in Theorem 3.2, adding a mild level of risk-aversion already contributes to

better convergence behaviors empirically, indicating the effectiveness of our risk-averse actor-critic framework even beyond theoretical implications.

### 5.1. Inspection Game Experiment

Our first experiment is a simple normal-form *Inspection Game*, whose payoff matrices are specified by $R_1 = \begin{bmatrix} 0 & 5 \\ 3 & 3 \end{bmatrix}; R_2 = \begin{bmatrix} -3 & -5 \\ 0 & 3 \end{bmatrix}$. Here player 1 is the *Inspectee*, and player 2 is the *Inspector*. The inspector decides whether to audit (left column) or not (right column), and the inspectee chooses whether to defect (top row) or to comply (bottom row). When the inspector inspects, it will enjoy a high utility if the inspectee defects, but will have low utility if the inspectee complies. For the inspectee, when choosing to defect, getting inspected will incur a penalty of $-5$, but if not inspected it will get a high utility of 3. We plot the learning dynamics for gradient descent when applying $z_{t+1} = \mathrm{Proj}_{\mathcal{Z}}(z_t - \eta F(z_t; \mathbf{R}))$ for different risk-aversion levels $\tau$ in Figure 3. We can see that without regularization,

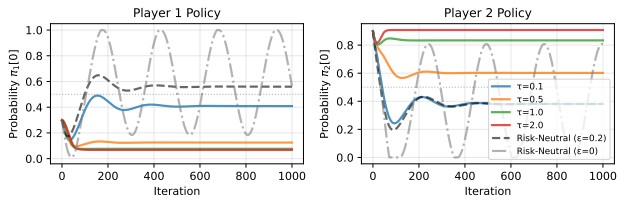

*Figure 3.* GD dynamics for different $\tau$ with KL and log-barrier risks, $\epsilon_i = 0.2$ and $\tau_1 = \tau_2$.

the risk-neutral gradient descent fails to converge, and when we fix $\epsilon = 0.2$, larger $\tau$ implies faster convergence. Additionally, the RQE is shown in Figure 3 as the policies to which gradient descent converge. For larger $\tau$ (more risk-averse), the inspector will less likely choose inspect, and the inspectee will more likely choose comply, leading to a lower utility variance for both players. This shows that our framework indeed induces more risk-averse agent behaviors as $\tau$ gets larger.

### 5.2. Gridworld Cooperation Game Experiment

Our second experiment considers an MARL gridworld environment with two agents where each step agents can choose between *cooperation*, which leads to a medium reward each step and *defection*, which when the other agent cooperates, gets a high reward, but gets no reward when the other agent also defects. We train both agents and compare the training reward curves (moving average of 100 episodes) for risk-averse and risk-neutral training (each for 10 independent runs) in Figure 4. We can see that risk-averse training curves are much more consistent and converge much faster, while risk-neutral curves are inconsistent across different runs and may never converge. To demonstrate that this stability stems

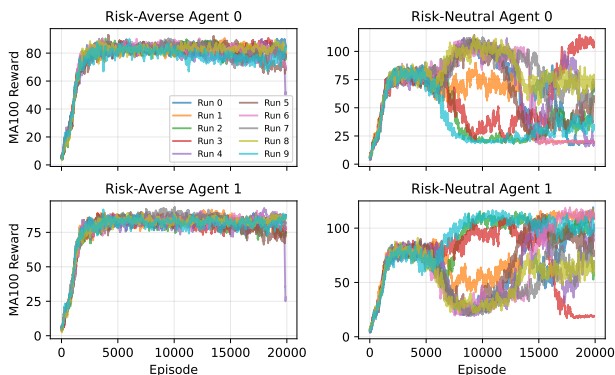

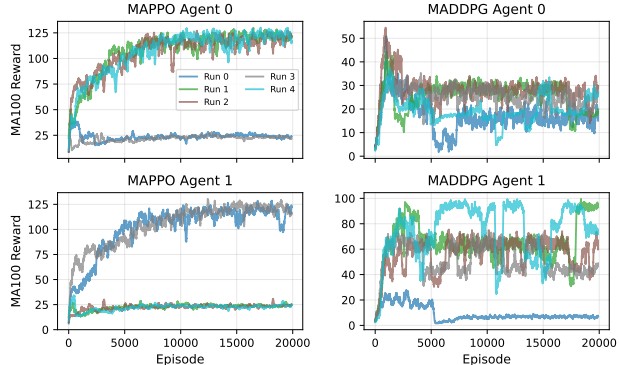

*Figure 4.* MA100 reward curves of gridworld cooperation game for 10 risk-averse and 10 risk-neutral training runs.

from risk-aversion rather than other algorithmic choices, we present the training curves (5 independent runs each) for MAPPO (Yu et al., 2022) and MADDPG (Lowe et al., 2017) as baselines in Figure 5, which shows that neither algorithm provides stable training curves as our risk-averse actor-critic algorithm.

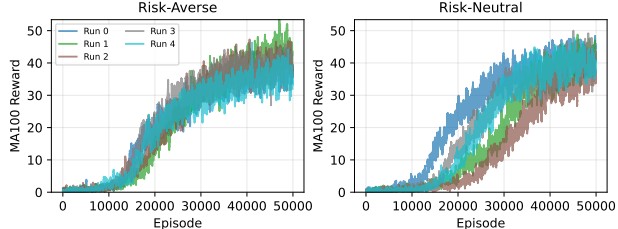

*Figure 6.* MA100 reward curves of Simple Tag fixing good agents for 5 risk-averse and 5 risk-neutral training runs.

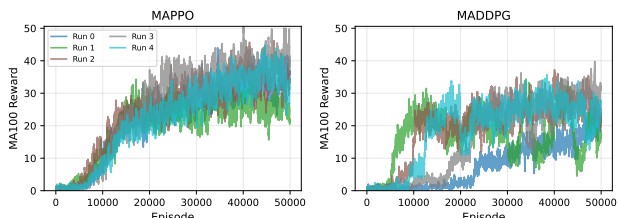

*Figure 7.* MA100 reward curves of Simple Tag fixing good agents of MAPPO and MADDPG for 5 independent training runs.

We also list the final reward statistics (mean $\pm$ standard deviation) across 5 runs in Table 1.

*Table 1.* Performance comparison of different algorithms.

|  | Risk-averse AC | MAPPO | MADDPG |
| --- | --- | --- | --- |
| Reward | 36.74$\pm$2.65 | 35.50$\pm$4.91 | 30.20$\pm$9.53 |

We can see that our risk-averse actor-critic achieves simultaneously higher reward and lower variance than both MAPPO and MADDPG. This illustrates the effectiveness of risk-aversion even in completing cooperative tasks. Details for MPE simple tag experiment are provided in Appendix D.2.

## 6. Conclusion

In this work, we have addressed the long-standing challenge of designing provably convergent MARL algorithms for discounted general-sum Markov games through shifting the objective from computationally intractable, risk-neutral Nash equilibria to the framework of RQE. Our theoretical analysis demonstrates how risk-aversion effectively regularizes and smooths the MARL optimization landscape, while our experiments verify how risk-aversion changes agent behavior and stabilizes learning. We believe our framework will serve as a foundation for future research in both theoretical and empirical fields, including further improvement in sample efficiency, adaptation of our results to linear function approximation and softmax policy parameterization, and designing independent learning algorithms that do not explicitly require opponent policies. Additionally, extending the RQE framework and the tractability result to extensive form games and even imperfect information games are interesting future directions.

*Figure 5.* MA100 reward curves of gridworld cooperation game of MAPPO and MADDPG for 5 independent training runs.

Details of the environment and algorithm parameters for gridworld cooperation games can be found in Appendix D.1.

### 5.3. MPE Simple Tag Experiment

To illustrate the effectiveness of risk-aversion in a broader class of games, we conduct an experiment on the Simple Tag environment with fixed good agents of Multi Particle Environments (MPE) (Lowe et al., 2017). We adopt an MPE environment with 3 agents (1 good agent and 2 adversaries) where the good agent policy is fixed (trained from a previous joint 3-agent training run). When an adversary hits the good agent, both adversaries receive a positive reward. We plot the reward curves for adversaries under 5 independent runs for risk-averse and risk-neutral respectively in Figure 6. We can see that running risk-averse training induces more consistent training curves while having a similar final performance. Similarly, we also provide the training curves for MAPPO and MADDPG baselines in Figure 7.

## Acknowledgements

EM acknowledges support from NSF Award 2240110.

## Impact Statement

This paper advances the theoretical understanding and algorithmic development of multi-agent reinforcement learning and game-theoretic solution concepts. Our work focuses on equilibrium computation under risk aversion and bounded rationality, and proposes provably convergent algorithms. The results are mathematical in nature and evaluated in controlled experimental environments; we do not deploy agents in real-world systems nor optimize for human behavior. We note that multi-agent learning frameworks have potential applications in socio-technical systems involving strategic interactions (e.g., markets, cybersecurity, or automated negotiation). While such domains could carry societal implications depending on the deployment context, the contributions of this work are methodological and do not in themselves present foreseeable direct ethical risks. We therefore believe that the broader societal impact of this paper aligns with that of advancing the field of machine learning more broadly.

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

# A. Notations and Basic Definitions

The notations used in our analysis is summarized in Table 2. We also clarify the definition of Lipschitz continuity, smoothness and contraction mapping below:

**Definition A.1** ($L$-Lipschitz Continuity). A function $f : \mathcal{X} \to \mathcal{Y}$ is said to be $L$-Lipschitz continuous with respect to the norm $\|\cdot\|$ if there exists a constant $L \geq 0$ such that for all $x, y \in \mathcal{X}$,

$$\|f(x) - f(y)\| \leq L\|x - y\|. \tag{18}$$

By default, we use $L_2$ norm if not specifically mentioned.

**Definition A.2** ($S$-Smoothness). A differentiable function $f : \mathcal{X} \to \mathbb{R}$ is said to be $S$-smooth with respect to the norm $\|\cdot\|$ if its gradient $\nabla f$ is $S$-Lipschitz continuous. That is, for all $x, y \in \mathcal{X}$,

$$\|\nabla f(x) - \nabla f(y)\|_* \leq S\|x - y\|, \tag{19}$$

where $S \geq 0$ is the smoothness constant and $\|\cdot\|_*$ denotes the dual norm induced by $\|\cdot\|$. By default, we use $L_2$ norm (whose dual norm is itself) if not specifically mentioned.

**Definition A.3** (Contraction Mapping). Let $(\mathcal{X}, d)$ be a complete metric space. A mapping $T : \mathcal{X} \to \mathcal{X}$ is called a $\gamma$-contraction mapping if there exists a constant $\gamma \in [0, 1)$ such that for all $x, y \in \mathcal{X}$,

$$d(T(x), T(y)) \leq \gamma d(x, y). \tag{20}$$

In a normed vector space where $d(x, y) = \|x - y\|$, this condition satisfies $\|T(x) - T(y)\| \leq \gamma\|x - y\|$. By default, regarding Bellman operators $\mathcal{T}, \mathcal{T}_z$ operating on **Q** functions, we use the max-norm $\|\cdot\|$ if not specifically mentioned.

*Table 2.* Summary of Notations

| Notation | Description |
|----------|-------------|
| *General Notation* | |
| $\mathbf{x} = (x_1, \ldots, x_n)$ | Tuple constructed by concatenating components for each player (e.g., $\mathbf{R} := (R_1, R_2)$). |
| $i, -i$ | Indices for the player $i$ and all other players $-i$. |
| $\|\cdot\|_1, \|\cdot\|_2, \|\cdot\|_\infty$ | $L_1, L_2$, and $L_\infty$ norms, respectively. |
| $\|A\|_{\max}$ | Max-norm of a matrix $A$, defined as $\max_{i,j} A_{ij}$. |
| $\langle \cdot, \cdot \rangle$ | Euclidean inner product. |
| $\langle x, y \rangle_\lambda$ | $\lambda$-weighted inner product $\sum_i \lambda_i x_i y_i$ for $\lambda > 0$. |
| $\|x\|_\lambda$ | $\lambda$-weighted norm $\sqrt{\langle x, x \rangle_\lambda}$. |
| $\mathrm{sp}(R)$ | Span of a matrix $R$: $\max_{i,j} R_{ij} - \min_{i,j} R_{ij}$. |
| $\mathrm{sp}(\mathbf{R})$ | Maximum span of a tuple of matrices: $\max_{k \in [n]} \mathrm{sp}(R_k)$. |
| $\|\mathbf{R}\|_{\max}$ | Maximum max-norm across a tuple: $\max_{k \in [n]} \|R_k\|_{\max}$. |
| $\mathrm{Proj}_{\mathcal{K}}(\cdot)$ | Euclidean projection operator onto a convex set $\mathcal{K}$. |
| *Game & Strategy Parameters* | |
| $\pi_i, p_i$ | Mixed strategy (policy) of agent $i$ and adversary $i$. |
| $\pi, p$ | Unsubscripted symbols represent joint policies for agents ($\pi$) and adversaries ($p$). |
| $z = (\pi, p)$ | Joint strategy profile combining primal agents and adversaries. |
| $\mathcal{A}_i, \Delta_{|\mathcal{A}_i|}$ | Action space and the probability simplex over actions for player $i$. |
| $\tau_i$ | Risk-aversion parameter for player $i$. |
| $\epsilon_i$ | Bounded rationality parameter for player $i$. |
| $R_i, \mathbf{R}$ | Payoff matrix for player $i$ and the tuple of matrices $(R_1, \ldots, R_n)$. |
| $\gamma$ | Discount factor for the Markov game. |
| *Functions & Operators* | |
| $f_i(\pi; \mathbf{R})$ | Global objective function usage in the original 2-player game (e.g., $f_i(\pi_i, \pi_{-i}; R_i)$). |
| $J_i(\pi, p; \mathbf{R})$ | Global objective function usage in the 4-player game. |
| $D_i(p_i, \pi_{-i})$ | Penalty function regularizing the adversary (e.g., KL/reverse KL). |
| $\nu_i(\pi_i)$ | Regularization function for the agent (e.g., Entropy/log-barrier). |
| $\mathcal{T}, \mathcal{T}_z$ | Risk-averse Bellman Optimality and Evaluation operators. |
| $\alpha_t, \beta_t$ | Step sizes for Critic (Q-function) and Actor (Policy) updates. |

# B. Related Work

In this section we provide a detailed discussion on related work. Our work primarily considers the solution concept of RQE originally proposed by Mazumdar et al. (2025), where they proved that all CCEs of the 4-player game have their $\pi$ components being an RQE of the original 2-player game, and provided an extension of the solution concept to finite-horizon Markov games. Working further on RQE, Zhang & Mazumdar (2025) studied the case where the 4-player game is monotone, and proved the uniqueness and Lipschitz continuity of RQE with respect to the payoff matrices. They also considered discounted Markov games and proved the contraction of Bellman operator under the same monotonicity condition. However, the condition provided in (Mazumdar et al., 2025) doesn't match that in (Zhang & Mazumdar, 2025) (and neither includes the other). To reconcile these disparate conditions, we introduce a generalized class of $\lambda$-monotone games and provide a condition that strictly includes both conditions. We additionally provide a practical algorithm that naturally fits into the Actor-Critic framework, not relying on the RQE oracle, which neither of the above works provides.

Our work is situated at the intersection of algorithm design for MARL, learning in games, risk-aversion, robustness and bounded rationality decision-making and stochastic approximation (especially for convergence analysis of Actor-Critic algorithms). We list the related work for each field in the following paragraphs.

**MARL algorithms and approaches.**  Various distinctive MARL algorithms have been proposed and empirically tested, among which MAPPO (Yu et al., 2022) and QMIX (Rashid et al., 2020) are the most empirically successful for fully cooperative environments. For the environments where agents are not fully cooperative, MADDPG (Lowe et al., 2017), MAAC (Iqbal & Sha, 2019) and even Individual PPO (Schulman et al., 2017) have been tested to have good empirical performance (Rudolph et al., 2026). Focusing on the strategic side of MARL, several techniques have been proposed, including *opponent shaping* (learning with opponent-learning awareness) (Foerster et al., 2018; Lu et al., 2022), *Theory of Mind* (Sclar et al., 2022) and *Rationality-preserving* Policy optimization (Lauffer et al., 2026). Empirical studies of risk-averse MARL have also been extensively conducted (Eriksson et al., 2022; Ganesh et al., 2019; Qiu et al., 2021; Shen et al., 2023). While these methods perform well in practice, they generally lack theoretical convergence guarantees, especially in *general-sum* environments.

**Learning in normal-form and Markov games.**  Parallel to empirical advances, the theoretical foundations of learning in games—both normal-form and Markov games (MGs)—have also seen significant development. In normal form games, although Nash equilibria are proven to be computationally intractable for general-sum games (Daskalakis et al., 2009), prior work developed learning algorithms like *fictitious play* (Robinson, 1951), MWU (Freund & Schapire, 1997) and OMD/OGDA (Daskalakis et al., 2018; Wei et al., 2021) that provably converge to Nash in zero-sum games, or to *Coarse Correlated equilibria* (CCEs) in general-sum games. In Markov games, Littman (1994) first formalized the notion of Markov games (discounted) and proposed minimax-Q learning that provably converges in 2-player zero-sum MGs. Bai & Jin (2020) provided the first provably sample-efficient self-play algorithm for finite-horizon zero-sum MGs achieving $\mathcal{O}(\sqrt{T})$ regret. For finite-horizon general-sum MGs, Jin et al. (2024) proposed a V-learning framework that provably learns its CCE in polynomial complexity with respect to the number of agents, yet the CCE is typically not a Markov policy, and requires joint randomness to execute. Prior work has also assumed access to *equilibrium oracles* to solve Markov Games. Hu & Wellman (2003) established Nash Q-learning, which extends Q-learning in single-agent RL to multi-agent RL through solving the stage game at each state. Liu et al. (2021) refined the algorithm for better sample-complexity through the V-learning framework, and Zehfroosh & Tanner (2022) combined the ideas of Nash Q-learning and delayed Q-learning and built a new algorithm for PAC MARL.

For general-sum games, in addition to learning CCEs, there are also works that try to learn Nash equilibrium for games with additional structure. Monderer & Shapley (1996) introduced *potential games*, where a single global function tracks the improvement of any agent's unilateral move, where Nash equilibria are learnable, and Fox et al. (2022) later generalized this idea to Markov potential games. Rosen (1965) introduced *monotone games*, where gradient dynamics converges to the unique Nash equilibrium, and many algorithms are designed and proven to have better rates of convergence (Cai & Zheng, 2023), or to be robust to noisy gradient steps (Mertikopoulos & Zhou, 2019). More recently, Even-dar et al. (2009) explored *socially convex* games where the (weighted) sum of player utilities is convex. There is also a line of works exploring the effectiveness of *regularization* for learning in games (Mertikopoulos & Sandholm, 2016; Giannou et al., 2021; Sokota et al., 2023; Cen et al., 2024). However, as shown by Mertikopoulos et al. (2018), regularization itself doesn't provide convergence guarantees even for zero-sum games without additional structure.

Despite various attempts on learning in normal-form and Markov games, no existing algorithm provide provable guarantee

for the most natural infinite-horizon general-sum Markov games, as is provided in our work.

**Risk-aversion, robustness and bounded-rationality in decision-making.** The solution concept of RQE naturally unifies three paradigms that have gained significant traction in recent years: behavioral robustness, risk-aversion, and bounded rationality in decision-making. In behavioral economics, risk-aversion (Gollier, 2001; Goeree & Offerman, 2002; Goeree et al., 2003) and bounded rationality (McKelvey & Palfrey, 1992; 1995; 1998) in human decision-making has been extensively studied, showing that the solution concept of Nash equilibrium does not necessarily capture real-world human decision-making behaviors, where both risk-aversion and bounded rationality are important aspects. In reinforcement learning, robustness and risk-aversion (proven to be equivalent in (Zhang et al., 2024)) has also been studied to tackle stochasticity and uncertainty in the environment (Mihatsch & Neuneier, 2002; Shen et al., 2014). Several more recent works have focused on the theoretical foundations of risk-sensitive MARL (Gao et al., 2021; Slumbers et al., 2023; Wang et al., 2024; Yekkehkhany et al., 2020), yet most of their results still rely on the game to be structured itself. Recent work (Lanzetti et al., 2025) considered an equilibrium concept of *strategically robust equilibrium* sharing similar expression to RQE but a different motivation of robustness. Contrary to risk-aversion, *risk-seeking* has also been studied recently in MARL by Zhang et al. (2025). There are different formulations of risk-aversion used in the works above, among which our work mainly considers a class of *convex risk measures* proposed by Föllmer & Schied (2002), where they proposed a *dual representation theorem* connecting risk-aversion to regularization in agent behaviors.

Among all these works regarding risk-aversion, our work differentiates itself in two aspects: (i) We mainly consider *strategic* risk-aversion, where agents are risk-averse against the behaviors of other agents rather than the environment; (ii) Our analysis do not rely on the payoff structure of the original game, but only on the level of risk-aversion and bounded rationality.

**Stochastic approximation and Actor-Critic algorithms.** Our theoretical analysis is largely based on the *stochastic approximation* (SA) framework introduced by (Robbins & Monro, 1951) that has served as a fundamental tool for analyzing the convergence of Q-learning (Tsitsiklis, 1994; Chen et al., 2022b) and TD learning (Srikant & Ying, 2019; Chen et al., 2024a). Following a *two-timescale* SA framework (Borkar, 1997), the seminal work of (Konda & Tsitsiklis, 1999) provided the first rigorous convergence proofs, and more recently, different variants of Actor-Critic algorithms (direct parameterization or linear function approximation, policy gradient or natural policy gradient) have been analyzed in different ways (asymptotic or finite-sample analysis) (Dalal et al., 2018; Zhang et al., 2020; Wu et al., 2020; Chen et al., 2022a). Despite the prevalence of two-timescale analyses that require the stepsizes decaying in different asymptotic rates, single-timescale analyses assuming a constant stepsize ratio have also been studied in single-agent settings (Fu et al., 2020; Chen & Zhao, 2023; Kumar et al., 2026). The single-timescale analyses are generally more complex as they require solving a coupled recursion system between the actor and the critic. SA has also been used to analyze learning in games. For two-player zero-sum MGs, Chen et al. (2024b) proved convergence of two-timescale Q-learning with function approximation, and later on provided an independent learning algorithm that enjoys last-iterate convergence in (Chen et al., 2024c).

While the analysis of Actor-Critic algorithms under the SA framework has been extensively studied, nearly all works are for single agent RL and uses a faster critic and a slower actor, whose final convergence relies on analyzing the policy gradient dynamics, with one exception of the so-called Critic-Actor framework proposed by Panda & Bhatnagar (2025) (yet their convergence analysis still relies on policy gradient instead of contraction). In comparison, In the MARL problem that we consider, policy gradient no longer yields negative drift, making all previous techniques invalid. To circumvent this, we adopt a "reverse" step size order with a slower critic with a faster actor in order to utilize the contraction property of the Bellman operator in our convergence proof.

## C. Algorithm: Multi-agent Risk-averse Actor-Critic

In this section we present our algorithm of Multi-agent Risk-averse Actor-Critic, as discussed in Section 4.3.

---

**Algorithm 1** Multi-agent Risk-averse Actor-Critic for $n$ agents

---

1: **Input:** Step sizes $\{\alpha_t\}_{t=0}^T, \{\beta_t\}_{t=0}^T$, agent regularizers $\nu_i(\cdot), D_i(\cdot,\cdot)$, bounded rationality level $\epsilon_i$, risk-aversion level $\tau_i$, weight-vector $\lambda$, number of episodes $T$ and number of samples $K$ per update. {Off-policy: Behavior policies $\{\pi_i^r\}_{i=1}^n$.}

2: Initialize the environment and receive initial states $s_0$.

3: Initialize $\pi_{i,0}, p_{i,0}$ to be uniform policies and $Q_{i,0} = 0$ for all $i \in \{1, 2, \ldots, n\}$.

4: **for** episode $t = 0, 1, 2, \ldots, T - 1$ **do**

5:     **for** agent $i = 1, 2, \ldots n$ and all $s \in \mathcal{S}$ **do**

6:         Update policy

$$\pi_{i,t+1}(\cdot|s) \leftarrow \text{Proj}_{\Delta_{|\mathcal{A}_i|}} \left( \pi_{i,t}(\cdot|s) - \beta_t \lambda_i [Q_{i,t}(s, \cdot) p_{i,t}(\cdot|s) + \epsilon_i \nabla \nu_i(\pi_{i,t}; s)] \right) \tag{21}$$

7:         Update adversary

$$p_{i,t+1}(\cdot|s) \leftarrow \text{Proj}_{\Delta_{|\mathcal{A}_{-i}|}} \left( p_{i,t}(\cdot|s) - \beta_t \lambda_i [-Q_{i,t}^T(s, \cdot) \pi_{i,t}(\cdot|s) + \frac{1}{\tau_i} \nabla_{p_i} D_i(p_i, \pi_{-i}; s)] \right) \tag{22}$$

8:     **end for**

9:     **for** timestep $k = 0, 1, 2, \ldots, K - 1$ **do**

10:         **for** agent $i = 1, 2, \ldots n$ **do**

11:             Sample $a_{i,k} \sim \pi_{i,t}(\cdot|s_k)$. {Off-policy: $a_{i,k} \sim \pi_i^r(\cdot|s_k)$}

12:         **end for**

13:         Play $\mathbf{a}_k = (a_{i,k})_{i=1}^n$, receive $\mathbf{r}_k = (r_{i,k})_{i=1}^n$ and observe $s_{k+1}$.

14:         **for** agent $i = 1, 2, \ldots n$ **do**

15:             Construct target:

$$\begin{aligned} \hat{q}_{i,k} = & - r_{i,k} + \gamma \pi_{i,t+1}^T(\cdot|s_{k+1}) Q_{i,t}(s_{k+1}, \cdot) p_{i,t+1}(\cdot|s_{k+1}) \\ & - \gamma \left( \frac{1}{\tau_i} D_i(p_{i,t+1}, \pi_{-i,t+1}; s_{k+1}) - \epsilon_i \nu_i(\pi_{i,t+1}; s_{k+1}) \right) \end{aligned} \tag{23}$$

16:         **end for**

17:     **end for**

18:     **for** agent $i = 1, 2, \ldots n$ **do**

19:         Compute update $\hat{\delta}_i$:

$$\hat{\delta}_i(s, \mathbf{a}) = \frac{1}{K} \sum_{k=0}^{K-1} (\hat{q}_{i,k} - Q_{i,t}(s_k, \mathbf{a}_k)) \mathbf{1}[(s, \mathbf{a}) = (s_k, \mathbf{a}_k)] \tag{24}$$

20:         Update $Q_{i,t+1}(s, \mathbf{a}) \leftarrow Q_{i,t}(s, \mathbf{a}) + \alpha_t \hat{\delta}_i(s, \mathbf{a})$ for all $s \in \mathcal{S}, \mathbf{a} \in \mathcal{A}$.

21:     **end for**

22:     Set $s_0 \leftarrow s_K$ for the next episode.

23: **end for**

24: **Output:** Policies $\pi_{i,T}, p_{i,T}$.

---

### C.1. Practical Implementation of Algorithm 1

Although Algorithm 1 is already a self-contained MARL algorithm, it doesn't necessarily perform well on practical environments. Therefore, we provide an adapted version that incorporates several usual implementation tricks, most of which being used on existing actor-critic algorithms like SAC (Haarnoja et al., 2018).

The first adaptation we make is to use neural networks to parametrize actor and critic. For actor, we use $\theta_i$ to denote the policy parameter of agent $i$, and $\bar{\theta}_i$ to denote the policy parameter of adversary $i$. For critic, we adopt the double-Q trick (Hasselt, 2010) and target networks (Mnih, 2013) (notice that target networks matches the $Q$ update in line 20 of

Algorithm 1). For agent $i$, we use $\phi_i^1, \phi_i^2$ denote two Q networks and $\phi_i^{\text{targ},1}, \phi_i^{\text{targ},2}$ denote the target networks.

The procedure of sampling new transitions only for one update is not sample efficient. Additionally, for large (or continuous) state spaces $\mathcal{S}$ we cannot afford updating policy for all states $\mathcal{S}$. Therefore, we introduce a replay buffer $\mathcal{D}$, from which we sample a batch $B \sim \mathcal{D}$ and optimize the following loss functions for actors:

$$
\begin{aligned}
\mathcal{L}(\theta_i; B) =& \frac{1}{|B|} \sum_{s \in B} \left( \pi_{\theta_i}(\cdot|s)^T \max\{Q_{\phi_i^1}, Q_{\phi_i^2}\}(s, \cdot) p_{\bar{\theta}_i}(\cdot|s) + \epsilon_i \nu_i(\pi_{\theta_i}; s) \right); \\
\mathcal{L}(\bar{\theta}_i; B) =& \frac{1}{|B|} \sum_{s \in B} \left( -\pi_{\theta_i}(\cdot|s)^T \max\{Q_{\phi_i^1}, Q_{\phi_i^2}\}(s, \cdot) p_{\bar{\theta}_i}(\cdot|s) + \frac{1}{\tau_i} D_i(p_{\bar{\theta}_i}, \pi_{\theta_{-i}}; s) \right).
\end{aligned}
\tag{25}
$$

Notice that minimizing (25) can be carried out using any built-in optimizer from various python libraries like `torch`, and when the batch $B$ precisely consists of every state $s \in \mathcal{S}$ once, using preconditioned gradient descent to optimize (25) is equivalent to (21) and (22). Additionally, when the action space is large or continuous, (25) can be approximated using samples from the *current* policies $\pi_{\theta_i}$ and $p_{\bar{\theta}_i}$ using similar tricks as in SAC (Haarnoja et al., 2018).

For critic update, we construct a loss function as follows:

$$
\mathcal{L}(\phi_i^k; B) = \frac{1}{|B|} \sum_{(s, \mathbf{a}, \mathbf{r}, s') \in B} (\hat{q}_i^{\text{targ}} - Q_{\phi_i^k}(s, \mathbf{a}))^2, k \in \{1, 2\}.
\tag{26}
$$

where $\hat{q}_i^{\text{targ}}$ is computed through (23) using target networks for each transition $(s, \mathbf{a}, r, s') \in B$ in the following way:

$$
\hat{q}_i^{\text{targ}} = -r_i + \gamma \left( \pi_{\theta_i}^T(\cdot|s') \max\{Q_{\phi_i^{\text{targ},1}}, Q_{\phi_i^{\text{targ},2}}\}(s', \cdot) p_{\bar{\theta}_i}(\cdot|s') - \frac{1}{\tau_i} D_i(p_{\bar{\theta}_i}, \pi_{\theta_{-i}}; s') + \epsilon_i \nu_i(\pi_{\theta_i}; s') \right)
\tag{27}
$$

Similarly, the critic loss (26) can also be optimized using built-in optimizers, and can be estimated using samples for large/continuous action spaces. After conducting an actor update step, we conduct a soft update for the target networks

$$
\phi_i^{\text{targ},k} \leftarrow (1 - \alpha_t) \phi_i^{\text{targ},k} + \alpha_t \phi_i^k, k \in \{1, 2\}.
\tag{28}
$$

To summarize, we present our practical adaptation of Algorithm 1 in Algorithm 2.

---

**Algorithm 2** Practical Multi-agent Risk-averse Actor-Critic for $n$ agents.

1: **Input:** Regularizers $\nu_i(\cdot), D_i(\cdot, \cdot)$, bounded rationality level $\epsilon_i$, risk-aversion level $\tau_i$. Initial network parameters $\theta_i, \bar{\theta}_i, \phi_i^1, \phi_i^2, \phi_i^{\text{targ},1}, \phi_i^{\text{targ},2}$, empty replay buffer $\mathcal{D}$.
2: **repeat**
3:     Observe state $s$ and sample joint action $\mathbf{a}$ for all agents with either on-policy or off-policy.
4:     Execute $\mathbf{a}$, observe reward $\mathbf{r}$ and next state $s'$.
5:     Store $(s, \mathbf{a}, \mathbf{r}, s')$ in replay buffer $\mathcal{D}$.
6:     **if** update now **then**
7:         Sample a batch $B$ from $\mathcal{D}$.
8:         Update actors and adversaries with (25) for all agents.
9:         Update critics with (26) and (27) for all agents.
10:        Update target networks with (28) for all agents.
11:     **end if**
12: **until** convergence

---

Notice that there exists a risk-neutral version of Algorithm 2 that does not maintain adversaries, which we use as benchmarks in our experiments. This can be done by replacing (25) by:

$$
\mathcal{L}(\theta_i; B) = \frac{1}{|B|} \sum_{s \in B} \left( \pi_{\theta_i}(\cdot|s)^T \max\{Q_{\phi_i^1}, Q_{\phi_i^2}\}(s, \cdot) \pi_{\theta_{-i}}(\cdot|s) + \epsilon_i \nu_i(\pi_{\theta_i}; s) \right),
\tag{29}
$$

and replace (27) by:

$$
\hat{q}_i^{\text{targ}} = -r_i + \gamma \pi_{\theta_i}^T(\cdot|s') \max\{Q_{\phi_i^{\text{targ},1}}, Q_{\phi_i^{\text{targ},2}}\}(s', \cdot) \pi_{\theta_{-i}}(\cdot|s') + \gamma \epsilon_i \nu_i(\pi_{\theta_i}; s').
\tag{30}
$$

*Remark* C.1. Since we only maintain one shared critic network for each agent-adversary pair, the only additional computational complexity of the risk-averse version of Algorithm 2 and its risk-neutral version is to maintain and update the adversary actors $\bar{\theta}_i$. Additionally, thanks to the off-policy nature of actor-critic algorithms, we don't have to maintain separate environments for each agent-adversary pair, but only need to sample from one single environment consisting of "real" agents. This means the statistical complexity of the risk-averse algorithm stays the same as the risk-neutral version.

## D. Experiment Details

In this section we provide details for our experiments.

### D.1. Details for Gridworld Cooperation Game

Here we provide details for our gridworld cooperation game experiment. We first specify our environment setup for gridworld cooperation game, and then provide the table of hyperparameters used in our experiments, followed by a presentation of detailed training data for each individual training run.

#### D.1.1. ENVIRONMENT SPECIFICATION

The gridworld cooperation game is a fully-observable MARL environment with 2 agents on a $5 \times 5$ gridworld shown in Figure 8.

*Figure 8.* Gridworld Layout. Agent 0 and agent 1 are shown in blue and red dots on the upper-left corner. The defection zones are painted in blue (for agent 0) and red (for agent 1)

In the gridworld cooperation game, both agents are initialized at the upper-left corner of the grid. There are three special cells. The upper-right corner (painted in blue) and lower-left corner (painted in red) are two *defection zones* for agent 0 and agent 1 respectively. The lower-right corner (painted in grey) is the *cooperation zone*. All other cells that are left blank in Figure 8 are called the *blank area*. At each timestep, if an agent is in the blank area or at the other agent's defection zone, it is not given any reward. If it is in the cooperation zone, it is given a reward of $0.5$ if the other agent is in the defection zone, a reward 1 if the other agent is in the blank area and a reward 2 if the other agent is also in the cooperation zone. If the agent is in its defection zone (indicated by the same color, it gets a reward of 3 if the other agent is in the cooperation zone, and gets 0 otherwise.

At each time step, each agent has 5 actions, whether to move to one direction, or to stay in the same cell. If the agent chooses an *infeasible* action (for example, choosing "up" when at the top row of the grid), it will randomly move to a feasible direction (including staying). When an agent gets into its defection zone, it is forced to stay in its defection zone until the end of the game. When an agent is in the cooperation zone, it is forced to stay in the cooperation zone with a probability of $0.7$ for each step.

For the gridworld cooperation game, the social welfare (sum of rewards) is maximized when both agents choose to cooperate. However, when the other agent chooses to cooperate, choosing to defect always yields a higher reward (3 compared to 2). Therefore, on a high-level, when agents are risk-neutral, there are two symmetric Nash equilibria (defect, cooperate) and (cooperate, defect) in this game. This pair of symmetric equilibria makes risk-neutral learning extremely unstable. In comparison, with risk-averse training, both agents expect the other agent to be "adversarial", so when it chooses to defect, the other agent won't cooperate and would lead to a zero reward for it. This leads to both agents choosing to cooperate and therefore achieves a higher social welfare.

### D.1.2. HYPERPARAMETER TABLES

We present the hyperparameters used for training in Table 3.

*Table 3.* Hyperparameters used for gridworld cooperation game experiments.

| Category | Hyperparameter | Value |
| --- | --- | --- |
| **Common** | Optimizer | Adam |
| | Actor Learning Rate | $5 \times 10^{-4}$ |
| | Critic Learning Rate | $5 \times 10^{-4}$ |
| | Discount Factor ($\gamma$) | 0.99 |
| | Batch Size | 256 |
| | Network Type (both Actor and Critic) | MLP |
| | Hidden Layers | 2 |
| | Hidden Units per Layer | 128 |
| | Activation Function | ReLU |
| **Risk-Aversion** | Risk-Aversion Level $\tau_1, \tau_2$ | 5.0, 5.0 |
| | Bounded-rationality $\epsilon_1, \epsilon_2$ | 0.2, 0.2 |
| | Regularizer Type | KL / Negative entropy |
| | Target Network Update ($\alpha$) | 0.002 |
| **Environment** | Horizon | 50 steps |
| | Training Episodes | 20000 |
| | Moving Average Window | 100 episodes |

For simplicity, we have fixed the risk-aversion level $\tau_1, \tau_2$ and bounded rationality level $\epsilon_1, \epsilon_2$ to be the same for both agents. We choose the regularizer type to be KL and negative entropy instead of KL and log-barrier to relieve the instability induced by the unboundedness of log-barrier functions.

### D.1.3. DETAILED TRAINING DATA

We present the detailed training and evaluation curves and the social welfare comparison for each training run as follows:

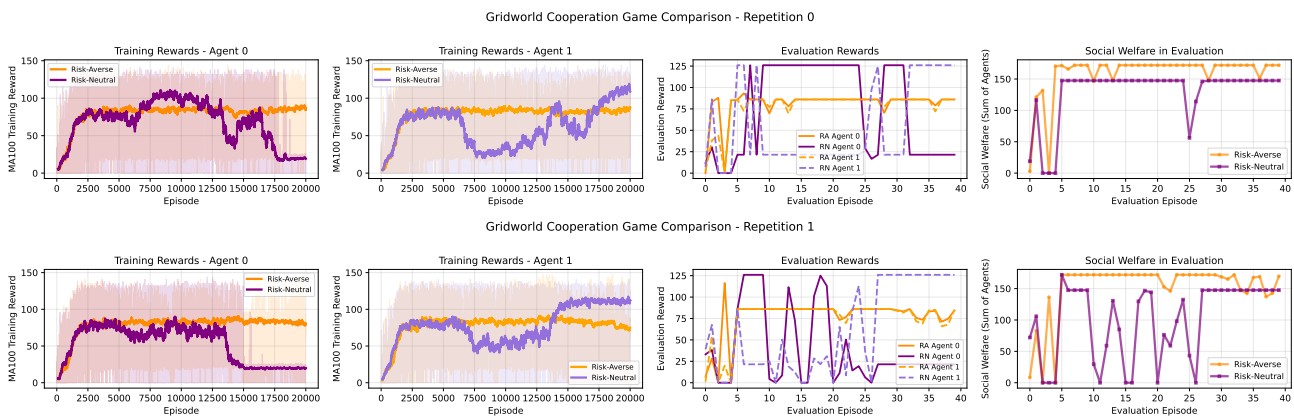

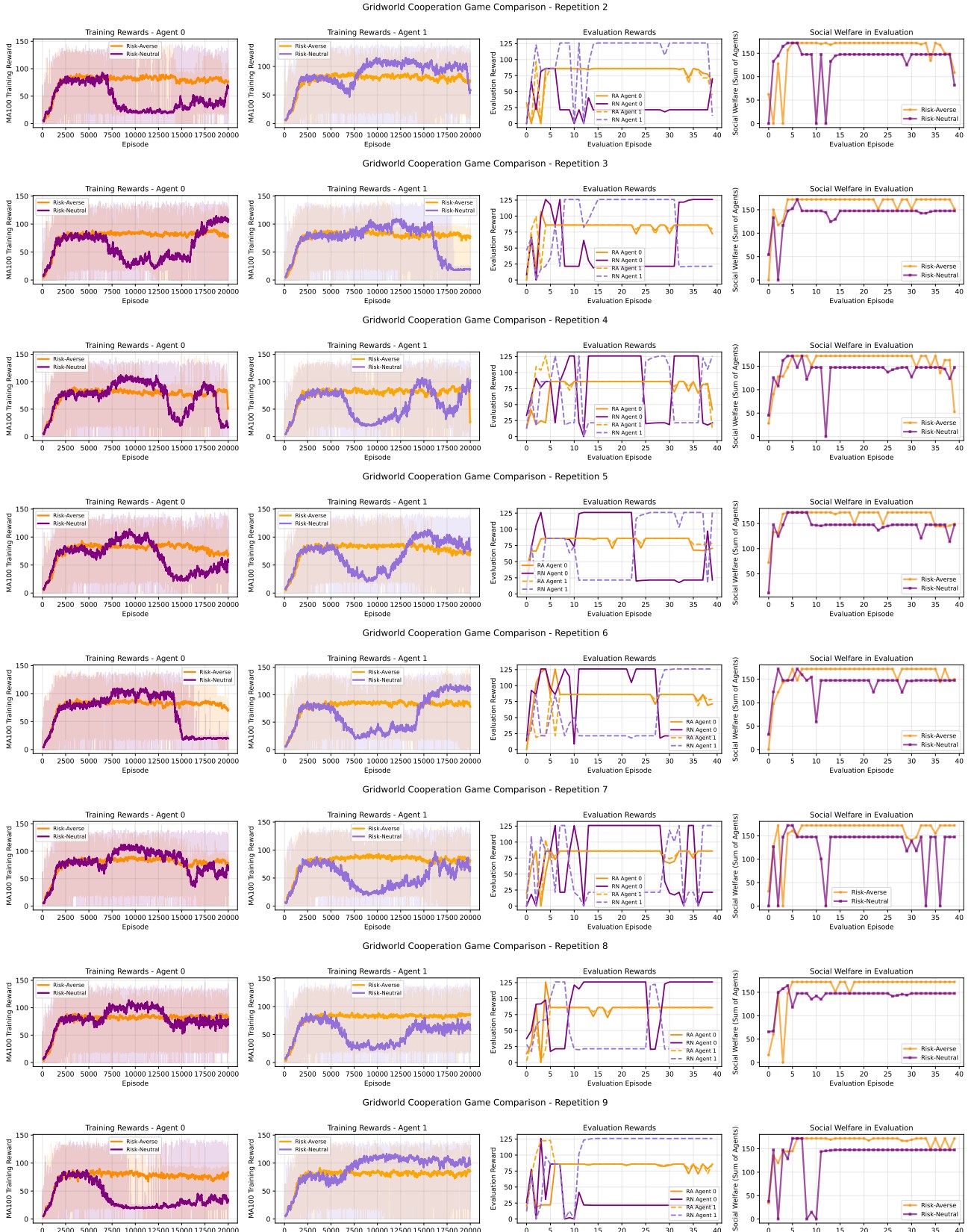

*Figure 9.* Detailed gridworld cooperation game comparison across 10 independent training runs.

## D.2. Details for MPE Simple Tag

In this section we present the details of our MPE simple tag experiment, similar to that for the gridworld cooperation game.

### D.2.1. ENVIRONMENT SPECIFICATION

Our experiment is based on the Simple Tag environment provided by Multi Particle Environments (MPE) (Lowe et al., 2017). We adopt a 3-agent version of the Simple Tag environment where there are 1 good agent and 2 adversaries, shown in Figure 10.

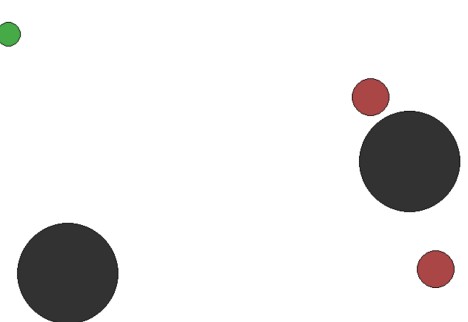

*Figure 10.* Layout of MPE2 Simple Tag environment with 3 agents. Green circle indicates the good agent, red circles indicate the adversaries, and black circles indicate the obstacles.

The MPE Simple Tag environment is a predator-prey environment. The good agent (green circle) is faster and receive a negative reward $-10$ for being hit by adversaries. Adversaries (red circle) are slower and are rewarded 10 for hitting good agents. Obstacles (large black circles) block the ways of agents. The agents are also penalized for leaving the dedicated area.

In order to create a fully-cooperative environment for MARL based on Simple Tag, we first jointly train three agents using Algorithm 2. After that, we fix the policy of the good agent and turn this environment into a fully cooperative game between two adversaries.

### D.2.2. HYPERPARAMETER TABLES

We provide the hyperparameters used in jointly training 3 agents, and training two adversaries to cooperate in Table 4 and Table 5 respectively. In each training, we use the default discrete state and action space for MPE. Compared with 3-player training, in 2-player training we modify the $\epsilon$ values to be $0.02$ to allow more higher performance in adversary policies (otherwise the adversary policies will be too random to chase the already well-trained good agent). We also increase the number of episodes from 20000 to 50000 for the training rewards to converge.

### D.2.3. DETAILED TRAINING DATA

We present the detailed training curve for each run in Figure 11.

*Table 4.* Hyperparameters used for MPE Simple Tag experiments.

| Category | Hyperparameter | Value |
|---|---|---|
| **Common** | Optimizer | Adam |
| | Actor Learning Rate | $5 \times 10^{-4}$ |
| | Critic Learning Rate | $5 \times 10^{-4}$ |
| | Discount Factor ($\gamma$) | 0.99 |
| | Batch Size | 256 |
| | Network Type (both Actor and Critic) | MLP |
| | Hidden Layers | 2 |
| | Hidden Units per Layer | 128 |
| | Activation Function | ReLU |
| **Risk-Aversion** | Risk-Aversion Level $\tau_1, \tau_2$ | Risk-neutral |
| | Bounded-rationality $\epsilon_1, \epsilon_2$ | $0.2, 0.2$ |
| | Regularizer Type | Negative entropy |
| | Target Network Update ($\alpha$) | 0.002 |
| **Environment** | Horizon | 50 steps |
| | Training Episodes | 20000 |
| | Moving Average Window | 100 episodes |

*Table 5.* Hyperparameters used for MPE Simple Tag experiments.

| Category | Hyperparameter | Value |
|---|---|---|
| **Common** | Optimizer | Adam |
| | Actor Learning Rate | $5 \times 10^{-4}$ |
| | Critic Learning Rate | $5 \times 10^{-4}$ |
| | Discount Factor ($\gamma$) | 0.99 |
| | Batch Size | 256 |
| | Network Type (both Actor and Critic) | MLP |
| | Hidden Layers | 2 |
| | Hidden Units per Layer | 128 |
| | Activation Function | ReLU |
| **Risk-Aversion** | Risk-Aversion Level $\tau_1, \tau_2$ | $1.0, 1.0$ |
| | Bounded-rationality $\epsilon_1, \epsilon_2$ | $0.02, 0.02$ |
| | Regularizer Type | KL / Negative entropy |
| | Target Network Update ($\alpha$) | 0.002 |
| **Environment** | Horizon | 50 steps |
| | Training Episodes | 50000 |
| | Moving Average Window | 100 episodes |

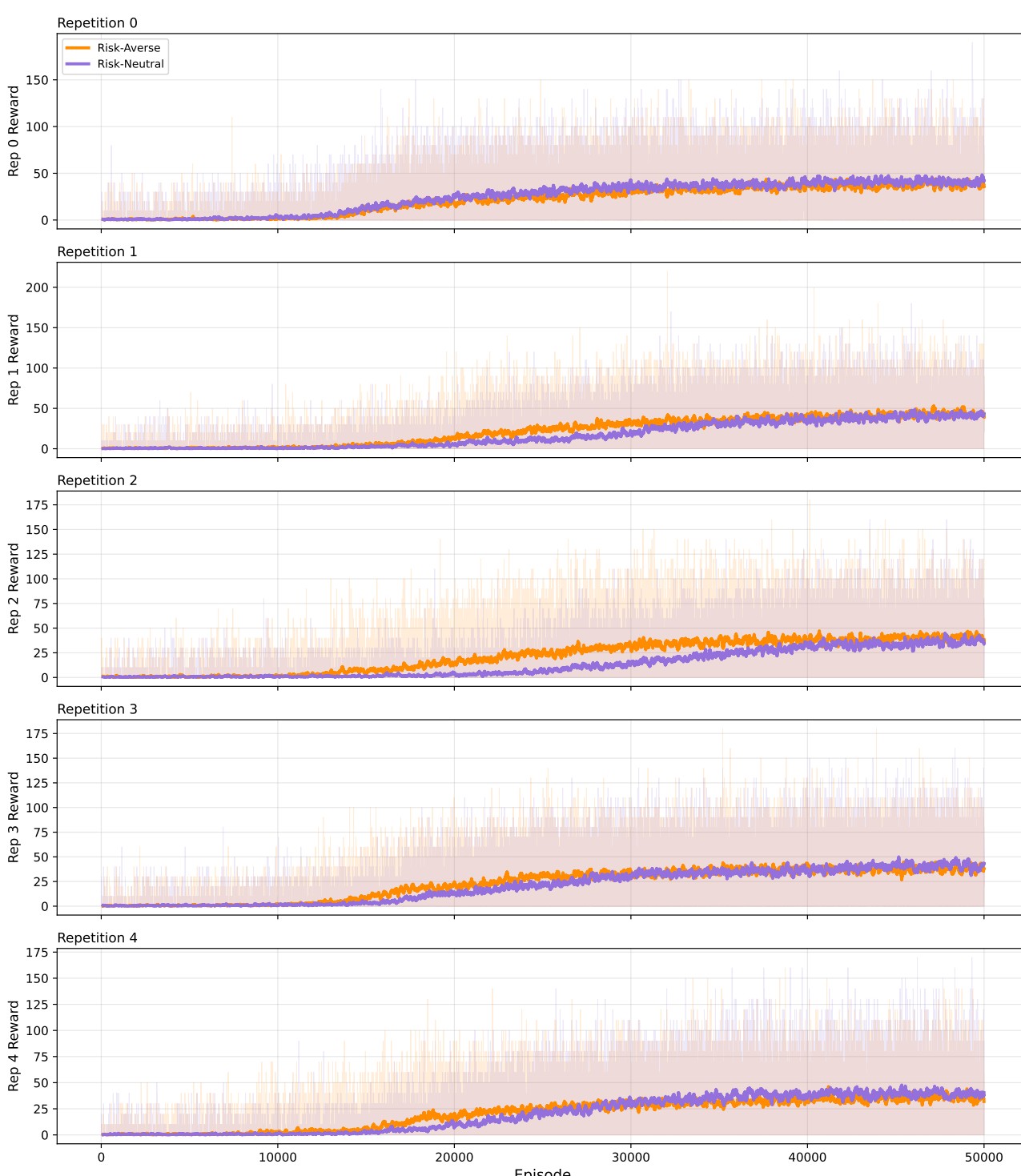

*Figure 11.* Detailed MPE Simple Tag comparison across 5 independent training runs.

# E. Proof of Proposition 2.6

In order to prove Proposition 2.6, we first construct an auxiliary system for our analysis. Similar to normal form games, each controlling $p_i, i \in \{1, 2\}$ and maximizes $V_i^{\pi,p}(s)$ for all $s \in \mathcal{S}$. This makes the game a 4-player Markov game.

In addition to the Bellman optimality operator $\mathcal{T}$ and Bellman evaluation operator $\mathcal{T}_z$ for some $z = (\pi, p)$, we define $\mathcal{T}_{*,\pi_{-i},p}$ and $\mathcal{T}_{\pi,*}$ (since $Q_i$ does not depend on $p_{-i}$, we drop the notational dependence as well) to be the Bellman optimality operators acting on $Q_i$ with respect to only $\pi_i$ or $p_i$ as follows:

$$\mathcal{T}_{*,\pi_{-i},p}Q_i(s, \mathbf{a}) = -r_i(s, \mathbf{a}) + \gamma \min_{\pi_i} \mathbb{E}_{s' \sim P(\cdot|s,\mathbf{a})}[\pi_i^T(\cdot|s')Q_i(s', \cdot)p_i(\cdot|s') - \frac{1}{\tau_i}D_i(p_i, \pi_{-i}; s') + \epsilon_i \nu_i(\pi_i; s')], \quad (31)$$

$$\mathcal{T}_{\pi,*}Q_i(s, \mathbf{a}) = -r_i(s, \mathbf{a}) + \gamma \max_{p_i} \mathbb{E}_{s' \sim P(\cdot|s,\mathbf{a})}[\pi_i^T(\cdot|s')Q_i(s', \cdot)p_i(\cdot|s') - \frac{1}{\tau_i}D_i(p_i, \pi_{-i}; s') + \epsilon_i \nu_i(\pi_i; s')]. \quad (32)$$

We can view $\mathcal{T}_{*,\pi_{-i},p}$ as optimizing $\pi$ assuming $\pi_{-i}, p$ being fixed, corresponding to each player optimizing its own policy assuming fixed opponents' policies.

To utilize the partial zero-sum structure between each player and its adversary in the 4-player game, for fixed $\pi_{-i}$, we define $\mathcal{T}_{\pi_{-i}}$ to be the operator mapping to the minimax solution w.r.t. $Q_i$, as a result of co-optimization between player $i$ and its adversary as follows:

$$\mathcal{T}_{\pi_{-i}}Q_i(s, \mathbf{a}) = -r_i(s, \mathbf{a}) + \gamma \min_{\pi_i} \max_{p_i} \mathbb{E}_{s' \sim P(\cdot|s,\mathbf{a})}[\pi_i^T(\cdot|s')Q_i(s', \cdot)p_i(\cdot|s') - \frac{1}{\tau_i}D_i(p_i, \pi_{-i}; s') + \epsilon_i \nu_i(\pi_i; s')]. \quad (33)$$

We have the following contraction property for all these three operators, and the coordinate-wise monotonicity for $\mathcal{T}_{*,\pi_{-i},p}$ and $\mathcal{T}_{\pi,*}$:

**Lemma E.1.** *The Bellman operators $\mathcal{T}_{*,\pi_{-i},p}, \mathcal{T}_{\pi,*}$ and $\mathcal{T}_{\pi_{-i}}$ satisfy $\|\mathcal{T}'Q_i - \mathcal{T}'Q_i'\|_{\max} \leq \gamma\|Q_i - Q_i'\|_{\max}$, for all $Q_i, Q_i' : \mathcal{S} \times \mathcal{A} \to \mathbb{R}$, where $\mathcal{T}' \in \{\mathcal{T}_{*,\pi_{-i},p}, \mathcal{T}_{\pi,*}, \mathcal{T}_{\pi_{-i}}\}$. Additionally, when $Q_i \geq Q_i'$, we have $\mathcal{T}_{*,\pi_{-i},p}Q_i \geq \mathcal{T}_{*,\pi_{-i},p}Q_i'$ and $\mathcal{T}_{\pi,*}Q_i \geq \mathcal{T}_{\pi,*}Q_i'$.*

*Proof.* For $\mathcal{T}_{*,\pi_{-i},p}$ and $\mathcal{T}_{\pi,*}$, the proofs of contraction and coordinate-wise monotonicity are standard using the same argument as that in single agent RL. We now prove the contraction of $\mathcal{T}_{\pi_{-i}}$ following a similar outline as in (Shapley, 1953). Let $\pi_i^\dagger(\cdot|s') = \arg\min_{\pi_i(\cdot|s')} \max_{p_i(\cdot|s')} \pi_i^T(\cdot|s')Q_i'(s', \cdot)p_i(\cdot|s') - \frac{1}{\tau_i}D_i(p_i, \pi_{-i}; s') + \epsilon_i \nu_i(\pi_i; s')$ and $p_i^\dagger(\cdot|s) = \arg\max_{p_i(\cdot|s')}(\pi_i^\dagger)^T(\cdot|s')Q_i(s', \cdot)p_i(\cdot|s') - \frac{1}{\tau_i}D_i(p_i, \pi_{-i}; s') + \epsilon_i \nu_i(\pi_i^\dagger; s')$

$$\|\mathcal{T}_{\pi_{-i}}Q_i - \mathcal{T}_{\pi_{-i}}Q_i'\|_{\max}$$
$$= \|\gamma \min_{\pi_i} \max_{p_i} \mathbb{E}_{s' \sim P(\cdot|s,\mathbf{a})}[\pi_i^T(\cdot|s')Q_i(s', \cdot)p_i(\cdot|s') - \frac{1}{\tau_i}D_i(p_i, \pi_{-i}; s') + \epsilon_i \nu_i(\pi_i; s')]$$
$$- \gamma \min_{\pi_i} \max_{p_i} \mathbb{E}_{s' \sim P(\cdot|s,\mathbf{a})}[\pi_i^T(\cdot|s')Q_i'(s', \cdot)p_i(\cdot|s') - \frac{1}{\tau_i}D_i(p_i, \pi_{-i}; s') + \epsilon_i \nu_i(\pi_i; s')]\|_{\max}$$
$$\overset{(i)}{=} \gamma \|\mathbb{E}_{s' \sim P(\cdot|s,\mathbf{a})}[\min_{\pi_i(\cdot|s')} \max_{p_i(\cdot|s')} \{\pi_i^T(\cdot|s')Q_i(s', \cdot)p_i(\cdot|s') - \frac{1}{\tau_i}D_i(p_i, \pi_{-i}; s') + \epsilon_i \nu_i(\pi_i; s')\} \quad (34)$$
$$- \min_{\pi_i'(\cdot|s')} \max_{p_i'(\cdot|s')} \{\pi_i'^T(\cdot|s')Q_i'(s', \cdot)p_i'(\cdot|s') - \frac{1}{\tau_i}D_i(p_i', \pi_{-i}; s') + \epsilon_i \nu_i(\pi_i'; s')\}]\|_{\max}$$
$$\overset{(ii)}{\leq} \gamma \|\mathbb{E}_{s' \sim P(\cdot|s,\mathbf{a})}[(\pi_i^\dagger)^T(\cdot|s')(Q_i - Q_i')(s', \cdot)p_i^\dagger(\cdot|s')]\|_{\max}$$
$$\leq \gamma \|Q_i - Q_i'\|_{\max}$$

where (i) holds because $\pi_i, p_i$ are defined per state $s' \in \mathcal{S}$, and (ii) holds by:

$$\min_{\pi_i(\cdot|s')} \max_{p_i(\cdot|s')} \{\pi_i^T(\cdot|s')Q_i(s', \cdot)p_i(\cdot|s') - \frac{1}{\tau_i}D_i(p_i, \pi_{-i}; s') + \epsilon_i \nu_i(\pi_i; s')\}$$
$$\leq \max_{p_i(\cdot|s')} \{(\pi_i^\dagger)^T(\cdot|s')Q_i(s', \cdot)p_i(\cdot|s') - \frac{1}{\tau_i}D_i(p_i, \pi_{-i}; s') + \epsilon_i \nu_i(\pi_i^\dagger; s')\} \quad (35)$$
$$= (\pi_i^\dagger)^T(\cdot|s')Q_i(s', \cdot)p_i^\dagger(\cdot|s') - \frac{1}{\tau_i}D_i(p_i^\dagger, \pi_{-i}; s') + \epsilon_i \nu_i(\pi_i^\dagger; s'),$$

and similarly,

$$\min_{\pi_i(\cdot|s')} \max_{p_i(\cdot|s')} \{\pi_i^T(\cdot|s')Q_i'(s',\cdot)p_i(\cdot|s') - \frac{1}{\tau_i}D_i(p_i,\pi_{-i};s') + \epsilon_i\nu_i(\pi_i;s')\}$$

$$= \max_{p_i(\cdot|s')} \{(\pi_i^\dagger)^T(\cdot|s')Q_i'(s',\cdot)p_i(\cdot|s') - \frac{1}{\tau_i}D_i(p_i,\pi_{-i};s') + \epsilon_i\nu_i(\pi_i^\dagger;s')\} \tag{36}$$

$$\geq (\pi_i^\dagger)^T(\cdot|s')Q_i'(s',\cdot)p_i^\dagger(\cdot|s') - \frac{1}{\tau_i}D_i(p_i^\dagger,\pi_{-i};s') + \epsilon_i\nu_i(\pi_i^\dagger;s').$$

This completes the proof of the contraction property. □

Given the contraction property, we proceed to prove the minimax theorem between player $i$ and its adversary in Markov games:

**Lemma E.2.** *Given fixed $\pi_{-i}$, the following holds for all $s \in \mathcal{S}$:*

$$\min_{\pi_i} \max_{p_i} V_i^{\pi_i,\pi_{-i},p_i}(s) = \max_{p_i} \min_{\pi_i} V_i^{\pi_i,\pi_{-i},p_i}(s) \tag{37}$$

*Proof.* This proof is also analogous to that given in (Shapley, 1953). By contraction property of $\mathcal{T}_{\pi_{-i}}$ and Banach's fixed point theorem, there exists a unique fixed point $Q_i^{*,\pi_{-i}}$ of $\mathcal{T}_{\pi_{-i}}$. Let $V_i^{*,\pi_{-i}}$ be the corresponding value function:

$$V_i^{*,\pi_{-i}}(s) = \min_{\pi_i} \max_{p_i} \pi_i(\cdot|s)^T Q_i^{*,\pi_{-i}}(s,\cdot)p_i(\cdot|s) - \frac{1}{\tau_i}D_i(p_i,\pi_{-i};s) + \epsilon_i\nu_i(\pi_i;s), \tag{38}$$

and let $\pi_i^*, p_i^*$ be the corresponding minimax solution, we have that $V_i^{\pi_i^*,\pi_{-i},p_i^*} = V_i^{*,\pi_{-i}}$ because:

$$\mathcal{T}_{\pi_i^*,\pi_{-i},p_i^*}Q_i^{*,\pi_{-i}}(s,\mathbf{a})$$

$$= -r_i(s,\mathbf{a}) + \gamma\mathbb{E}_{s'\sim P(\cdot|s,\mathbf{a})}[(\pi_i^*)^T(\cdot|s')Q_i^{*,\pi_{-i}}(s',\cdot)p_i^*(\cdot|s') - \frac{1}{\tau_i}D_i(p_i^*,\pi_{-i};s') + \epsilon_i\nu_i(\pi_i^*;s')]$$

$$= -r_i(s,\mathbf{a}) + \gamma\mathbb{E}_{s'\sim P(\cdot|s,\mathbf{a})}[\min_{\pi_i}\max_{p_i}\pi_i^T(\cdot|s')Q_i^{*,\pi_{-i}}(s',\cdot)p_i(\cdot|s') - \frac{1}{\tau_i}D_i(p_i,\pi_{-i};s') + \epsilon_i\nu_i(\pi_i;s')] \tag{39}$$

$$= \mathcal{T}_{\pi_{-i}}Q_i^{*,\pi_{-i}}(s,\mathbf{a})$$

$$= Q_i^{*,\pi_{-i}}(s,\mathbf{a})$$

so $Q_i^{*,\pi_{-i}} = Q_i^{\pi_i^*,\pi_{-i},p_i^*}$ because the unique fixed point of $\mathcal{T}_{\pi_i^*,\pi_{-i},p_i^*}$ is $Q_i^{\pi_i^*,\pi_{-i},p_i^*}$ (by a similar contraction statement on $\mathcal{T}_{\pi_i^*,\pi_{-i},p_i^*}$).

Similarly, let $\pi_i^\dagger, p_i^\dagger$ be the maximin solution to $\max_{p_i}\min_{\pi_i}V_i^{\pi_i,\pi_{-i},p_i}(s)$, by similar rationale we know that $Q^{\pi_i^\dagger,\pi_{-i},p_i^\dagger}$ is a unique fixed point of the maximin Bellman operator $\mathcal{T}_{\pi_{-i}}'$ defined by:

$$\mathcal{T}_{\pi_{-i}}'Q_i(s,\mathbf{a}) = -r_i(s,\mathbf{a}) + \gamma\max_{p_i}\min_{\pi_i}\mathbb{E}_{s'\sim P(\cdot|s,\mathbf{a})}[\pi_i^T(\cdot|s')Q_i(s',\cdot)p_i(\cdot|s') - \frac{1}{\tau_i}D_i(p_i,\pi_{-i};s') + \epsilon_i\nu_i(\pi_i;s')]. \tag{40}$$

However, for arbitrary $Q_i$ function (not as a function of $\pi_i$ and $p_i$) that can be viewed as a matrix, the minimax theorem for normal-form game holds, such that

$$\max_{p_i}\min_{\pi_i}\mathbb{E}_{s'\sim P(\cdot|s,\mathbf{a})}[\pi_i^T(\cdot|s')Q_i(s',\cdot)p_i(\cdot|s') - \frac{1}{\tau_i}D_i(p_i,\pi_{-i};s') + \epsilon_i\nu_i(\pi_i;s')]$$

$$= \min_{\pi_i}\max_{p_i}\mathbb{E}_{s'\sim P(\cdot|s,\mathbf{a})}[\pi_i^T(\cdot|s')Q_i(s',\cdot)p_i(\cdot|s') - \frac{1}{\tau_i}D_i(p_i,\pi_{-i};s') + \epsilon_i\nu_i(\pi_i;s')] \tag{41}$$

so that $\mathcal{T}_{\pi_{-i}}$ and $\mathcal{T}_{\pi_{-i}}'$ are indeed identical and share the same fixed point. Therefore, we claim that:

$$\min_{\pi_i}\max_{p_i}V_i^{\pi_i,\pi_{-i},p_i}(s) = V_i^{\pi_i^*,\pi_{-i},p_i^*} = V_i^{*,\pi_{-i}} = V_i^{\pi_i^\dagger,\pi_{-i},p_i^\dagger} = \max_{p_i}\min_{\pi_i}V_i^{\pi_i,\pi_{-i},p_i}(s) \tag{42}$$

□

Having established the minimax theorem for Markov games, we now prove that similar to the normal-form case, that a Nash equilibrium of the 4-player Markov game corresponds to an RQE of the two-player game, and vice versa:

**Lemma E.3.** *Let $(\pi^*, p^*)$ be a Nash equilibrium of the 4-player Markov game, we have that $\pi^*$ is an RQE of the original two-player Markov game. Furthermore, if $\pi^*$ is an RQE of the two-player Markov game, then $(\pi^*, p^*)$ is a Nash equilibrium of the 4-player Markov game where $p_i^* = \arg\max_{p_i : \mathcal{S} \to \Delta_{|\mathcal{A}_{-i}|}} V_i^{\pi^*, p_i}(s)$ for all $s \in \mathcal{S}$.*

*Proof.* The first statement can be proved using a similar argument as in the proof of Proposition 2.2. Let $(\pi^*, p^*)$ be a Nash equilibrium of the 4-player Markov game, by definition we have for all $\pi_i : \mathcal{S} \to \Delta_{|\mathcal{A}_i|}$ and $s \in \mathcal{S}$:

$$V_i^{\pi^*, p^*}(s) \leq V_i^{\pi_i, \pi^*_{-i}, p^*}(s), \tag{43}$$

and for all $p_i : \mathcal{S} \to \Delta_{|\mathcal{A}_{-i}|}$:

$$V_i^{\pi^*, p_i, p^*_{-i}}(s) \leq V_i^{\pi^*, p^*}(s). \tag{44}$$

Therefore, we have:

$$\begin{aligned}
f_i(\pi_i^*, \pi_{-i}^*; s) &= \max_{p_i : \mathcal{S} \to \Delta_{|\mathcal{A}_{-i}|}} V_i^{\pi^*, p}(s) = V_i^{\pi^*, p^*}(s) \\
&\leq V_i^{\pi_i, \pi^*_{-i}, p^*}(s) \leq \max_{p_i' : \mathcal{S} \to \Delta_{|\mathcal{A}_{-i}|}} V_i^{\pi_i, \pi^*_{-i}, p_i'}(s) \\
&= f_i(\pi_i, \pi^*_{-i}; s).
\end{aligned} \tag{45}$$

For the second statement, let $\pi^*$ be an RQE of the two-player Markov game, by the minimax theorem in Markov games, we have a similar argument as in Proposition 2.2 that

$$\min_{\pi_i : \mathcal{S} \to \Delta_{|\mathcal{A}_i|}} \max_{p_i : \mathcal{S} \to \Delta_{|\mathcal{A}_{-i}|}} V_i^{\pi_i, \pi^*_{-i}, p_i}(s) = \min_{\pi_i : \mathcal{S} \to \Delta_{|\mathcal{A}_i|}} V_i^{\pi_i, \pi^*_{-i}, p_i^*}(s), \tag{46}$$

so that for all $s \in \mathcal{S}$,

$$V_i^{\pi^*, p^*}(s) = \min_{\pi_i : \mathcal{S} \to \Delta_{|\mathcal{A}_i|}} \max_{p_i : \mathcal{S} \to \Delta_{|\mathcal{A}_{-i}|}} V_i^{\pi_i, \pi^*_{-i}, p_i}(s) = \min_{\pi_i : \mathcal{S} \to \Delta_{|\mathcal{A}_i|}} V_i^{\pi_i, \pi^*_{-i}, p_i^*}(s) \leq V_i^{\pi_i', \pi^*_{-i}, p_i^*}(s) \tag{47}$$

for all $\pi_i' : \mathcal{S} \to \Delta_{|\mathcal{A}_i|}$, which completes the proof. $\square$

With the auxiliary lemmas stated, we proceed to prove Proposition 2.6.

We first prove the first part. Let $\mathbf{Q}^*$ be a fixed point of the Bellman optimality operator and $z^* = (\pi^*, p^*)$ denote the minimax policies, we first prove that $\mathbf{Q}^* = \mathbf{Q}^{z^*}$. By the definition (13) of $\mathcal{T}$ we have that for all $i \in \{1, 2\}$ and $(s, \mathbf{a}) \in \mathcal{S} \times \mathcal{A}$:

$$\begin{aligned}
&Q_i^*(s, \mathbf{a}) \\
=&(\mathcal{T}\mathbf{Q}^*)_i(s, \mathbf{a}) \\
=&-r_i(s, \mathbf{a}) + \gamma \mathbb{E}_{s' \sim P(\cdot|s, \mathbf{a})}[\text{RQE}_i(-\mathbf{Q}^*(s', \cdot))] \\
=&-r_i(s, \mathbf{a}) + \gamma \mathbb{E}_{s' \sim P(\cdot|s, \mathbf{a})}\left[(\pi_i^*)^T(\cdot|s')Q_i^*(s', \cdot)p_i^*(\cdot|s') - \frac{1}{\tau_i}D_i(p_i^*, \pi^*_{-i}; s') + \epsilon_i \nu_i(\pi_i^*; s')\right] \\
=&-r_i(s, \mathbf{a}) + \gamma \mathbb{E}_{s' \sim P(\cdot|s, \mathbf{a})}\left[J_i\left(\pi_i^*(\cdot|s'), \pi^*_{-i}(\cdot|s'), p_i^*(\cdot|s'); -\mathbf{Q}^*(s', \cdot)\right)\right]
\end{aligned} \tag{48}$$

We claim that $\mathbf{Q}^* = \mathbf{Q}^{z^*}$ because we have the following for all $i \in [N], s \in \mathcal{S}$ and $\mathbf{a} \in \mathcal{A}$:

$$\begin{aligned}
&Q_i^*(s, \mathbf{a}) - Q_i^{z^*}(s, \mathbf{a}) \\
=&\gamma \mathbb{E}_{s' \sim P(\cdot|s, \mathbf{a})}\left[J_i\left(\pi_i^*(\cdot|s'), \pi^*_{-i}(\cdot|s'), p_i^*(\cdot|s'); -\mathbf{Q}^*(s', \cdot)\right) - J_i\left(\pi_i^*(\cdot|s'), \pi^*_{-i}(\cdot|s'), p_i^*(\cdot|s'); -\mathbf{Q}^{z^*}(s', \cdot)\right)\right] \\
=&\gamma \mathbb{E}_{s' \sim P(\cdot|s, \mathbf{a})}[(\pi_i^*)^T(\cdot|s')(Q_i^* - Q_i^{z^*})(s', \cdot)p_i^*(\cdot|s')] \\
\leq&\gamma \|\mathbf{Q}^* - \mathbf{Q}^{z^*}\|_{\max}.
\end{aligned} \tag{49}$$

Since this inequality holds for all $i, s \in \mathcal{S}$ and $\mathbf{a} \in \mathcal{A}$, we know that $\|\mathbf{Q}^* - \mathbf{Q}^{z^*}\|_{\max} = 0$.

Therefore, we have that for all policy $\pi_i$ and all states $s \in \mathcal{S}$,

$$
\begin{aligned}
f_i(\pi_i, \pi_{-i}^*; s) &= \max_{p_i} V_i^{\pi_i, \pi_{-i}^*, P}(s) \\
&= \max_{p_i} \pi_i(\cdot|s)^T Q_i^{\pi_i, \pi_{-i}^*, P}(s, \cdot) p_i(\cdot|s) - \frac{1}{\tau_i} D_i(p_i, \pi_{-i}^*; s) + \epsilon_i \nu_i(\pi_i; s) \\
&\overset{(i)}{\geq} \pi_i(\cdot|s)^T Q_i^{\pi_i, \pi_{-i}^*, p^*}(s, \cdot) p_i^*(\cdot|s) - \frac{1}{\tau_i} D_i(p_i^*, \pi_{-i}^*; s) + \epsilon_i \nu_i(\pi_i; s) \\
&\overset{(ii)}{\geq} (\pi_i^*)^T(\cdot|s) Q_i^*(s, \cdot) p_i^*(\cdot|s) - \frac{1}{\tau_i} D_i(p_i^*, \pi_{-i}^*; s) + \epsilon_i \nu_i(\pi_i^*; s) \\
&= \text{RQE}_i(-\mathbf{Q}^*(s, \cdot)) \\
&= \text{RQE}_i(-\mathbf{Q}^{z^*}(s, \cdot))
\end{aligned}
\tag{50}
$$

where (i) holds because we have taken maximum over $p_i$. Now we prove that (ii) holds. Notice that now $\pi_{-i}^*$ and $p^*$ are fixed, the problem becomes a single-agent MDP problem where the only policy to maximize is $\pi_i$. Recall that $\mathcal{T}_{*, \pi_{-i}^*, p^*}$ is the Bellman optimality operator acting on $Q_i$ with respect to only $\pi_i$ as follows:

$$
\mathcal{T}_{*, \pi_{-i}^*, p^*} Q_i(s, \mathbf{a}) = -r_i(s, \mathbf{a}) + \gamma \min_{\pi_i} \mathbb{E}_{s' \sim P(\cdot|s, \mathbf{a})} [\pi_i^T(\cdot|s') Q_i(s', \cdot) p_i^*(\cdot|s') - \frac{1}{\tau_i} D_i(p_i^*, \pi_{-i}^*; s') + \epsilon_i \nu_i(\pi_i; s')] \tag{51}
$$

By construction of $\pi_i^*$ we know that $\mathbf{Q}^*$ is also a fixed point of $\mathcal{T}_{*, \pi_{-i}^*, p^*}$ as well. By standard result in single-agent MDPs and $\pi_i^*$ is the corresponding optimal policy, we have:

$$
V_i^{\pi_i, \pi_{-i}^*, p^*}(s) \geq V_i^{\pi_i^*, \pi_{-i}^*, p^*}(s), \forall s \in \mathcal{S}, \pi_i : \mathcal{S} \rightarrow \Delta_{|\mathcal{A}_i|} \tag{52}
$$

which immediately implies (ii) and completes the proof of the first part.

Now we proceed to prove the second part. Let $\pi^*$ be an RQE of the two-player Markov game, we know by Lemma E.3 that $(\pi^*, p^*)$ is a Nash equilibrium of the 4-player Markov game. By Lemma 10 in (Hu & Wellman, 2003), we know that $(\pi^*, p^*)$ is also a Nash equilibrium of the stage game characterized by $\mathbf{Q}^{\pi^*, p^*}(s, \cdot)$ for all $s \in \mathcal{S}$. Therefore, by Proposition 2.2 we know that $\pi^*$ is an RQE of the two-player stage , which suggests that

$$
V_i^{\pi^*, p^*}(s) = \text{RQE}_i(-\mathbf{Q}^{\pi^*, p^*}(s, \cdot)) \tag{53}
$$

This leads to the final result:

$$
\begin{aligned}
(\mathcal{T} \mathbf{Q}^{\pi^*, p^*})_i(s, \mathbf{a}) &= -r_i(s, \mathbf{a}) + \gamma \mathbb{E}_{s' \sim P(\cdot|s, \mathbf{a})} [\text{RQE}_i(-\mathbf{Q}^{\pi^*, p^*}(s', \cdot))] \\
&= -r_i(s, \mathbf{a}) + \gamma \mathbb{E}_{s' \sim P(\cdot|s, \mathbf{a})} [V_i^{\pi^*, p^*}(s')] \\
&= Q_i^{\pi^*, p^*}(s, \mathbf{a}).
\end{aligned}
\tag{54}
$$

which completes the proof of Proposition 2.6.

# F. Proofs and Additional Details for Section 3

## F.1. Proof of Theorem 3.1

We follow a similar approach as in (Zhang & Mazumdar, 2025).

**Proof of uniqueness.** Consider the KKT conditions for the objective functions (6a) and (6b) over the simplex of each player given $\mathbf{R}$, which must be satisfied at equilibrium point $z^*$. For the original player we have:

$$
\begin{aligned}
-R_i p_i^* + \epsilon_i \nabla \nu_i(\pi_i^*) - \lambda(\pi_i) + \mu(\pi_i) 1 &= 0; \\
\pi_i^* \in \Delta_{|\mathcal{A}_i|}; \lambda(\pi_i) \geq 0; \mu(\pi_i) \in \mathbb{R}; \lambda(\pi_i)^T \pi_i^* &= 0.
\end{aligned}
\tag{55}
$$

where $i \in \{1, 2\}$, $\lambda(\pi_i)$ and $\mu(\pi_i)$ are Lagrange multipliers with respect to the simplex constraint, and $\lambda(\pi_i)^T \pi_i^* = 0$ denotes complementary slackness. For the adversaries we have:

$$
\begin{aligned}
R_i^T \pi_i^* + \frac{1}{\tau_i} \nabla_p D_i(p_i^*, \pi_{-i}^*) - \lambda(p_i) + \mu(p_i) 1 &= 0; \\
p_i^* \in \Delta_{|\mathcal{A}_{-i}|}; \lambda(p_i) \geq 0; \mu(p_i) \in \mathbb{R}; \lambda(p_i)^T p_i^* &= 0.
\end{aligned}
\tag{56}
$$

where similarly $i \in \{1, 2\}$, $\lambda(p_i), \mu(p_i)$ are Lagrange multipliers and $\lambda(p_i)^T p_i^* = 0$ denotes complementary slackness. We can combine (55) and (56) in a more compact form:

$$F(z^*; \mathbf{R}) = \begin{bmatrix} \lambda(\pi_1) - \mu(\pi_1)1 \\ \lambda(\pi_2) - \mu(\pi_2)1 \\ \lambda(p_1) - \mu(p_1)1 \\ \lambda(p_2) - \mu(p_2)1 \end{bmatrix}. \tag{57}$$

Therefore, for arbitrary $z \in \mathcal{Z}$ we have:

$$
\begin{aligned}
&\langle z^* - z, F(z^*; \mathbf{R}) \rangle_\lambda \\
&\overset{(i)}{=} \lambda_1 \langle \pi_1^* - \pi_1, \lambda(\pi_1) - \mu(\pi_1)1 \rangle + \lambda_2 \langle \pi_2^* - \pi_2, \lambda(\pi_2) - \mu(\pi_2)1 \rangle \\
&\quad + \lambda_1 \langle p_1^* - p_1, \lambda(p_1) - \mu(p_1)1 \rangle + \lambda_2 \langle p_2^* - p_2, \lambda(p_2) - \mu(p_2)1 \rangle \\
&\overset{(ii)}{=} \lambda_1 \langle \pi_1^* - \pi_1, \lambda(\pi_1) \rangle + \lambda_2 \langle \pi_2^* - \pi_2, \lambda(\pi_2) \rangle + \lambda_1 \langle p_1^* - p_1, \lambda(p_1) \rangle + \lambda_2 \langle p_2^* - p_2, \lambda(p_2) \rangle \\
&\overset{(iii)}{=} -\lambda_1 \langle \pi_1, \lambda(\pi_1) \rangle - \lambda_2 \langle \pi_2, \lambda(\pi_2) \rangle - \lambda_1 \langle p_1, \lambda(p_1) \rangle - \lambda_2 \langle p_2, \lambda(p_2) \rangle \\
&\overset{(iv)}{\leq} 0
\end{aligned}
\tag{58}
$$

where (i) uses the definition of $\lambda$-weighted inner product, (ii) holds because $\pi_i \in \Delta_{|\mathcal{A}_i|}, p_i \in \Delta_{|\mathcal{A}_{-i}|}$ so that $\langle \pi_i^* - \pi_i, 1 \rangle = 0$ and $\langle p_i^* - p_i, 1 \rangle = 0$, (iii) holds by complementary slackness and (iv) holds because $\lambda_i, \pi_i, p_i, \lambda(\pi_i), \lambda(p_i) > 0$.

If $z_1^*$ and $z_2^*$ are two Nash equilibria of the 4-player game with respect to $\mathbf{R}$, we must have $\langle z_1 - z_2, F(z_1; \mathbf{R}) \rangle_\lambda \leq 0$ and $\langle z_2 - z_1, F(z_2; \mathbf{R}) \rangle_\lambda \leq 0$. Adding these two inequalities up we have $\langle z_1 - z_2, F(z_1; \mathbf{R}) - F(z_2; \mathbf{R}) \rangle_\lambda \leq 0$, combining $\lambda$-strict monotonicity of $F(\cdot; \mathbf{R})$ we have $z_1 = z_2$, indicating the uniqueness of the Nash equilibrium.

**Proof of Lipschitz Continuity.** In order to prove the Lipschitz continuity of RQE with respect to payoff matrices, let $\mathbf{R}$ and $\mathbf{R}'$ be two pairs of payoff matrices, notice that (58) implies:

$$\langle z^* - z^\dagger, F(z^*; \mathbf{R}) \rangle_\lambda \leq 0; \quad \langle z^\dagger - z^*, F(z^\dagger; \mathbf{R}') \rangle_\lambda \leq 0. \tag{59}$$

adding these two inequalities up, we have:

$$\langle z^* - z^\dagger, F(z^*; \mathbf{R}) - F(z^\dagger; \mathbf{R}') \rangle_\lambda \leq 0. \tag{60}$$

We split the left hand side into two difference terms and get:

$$\langle z^* - z^\dagger, F(z^*; \mathbf{R}) - F(z^*; \mathbf{R}') \rangle_\lambda + \langle z^* - z^\dagger, F(z^*; \mathbf{R}') - F(z^\dagger; \mathbf{R}') \rangle_\lambda \leq 0. \tag{61}$$

For the first term, we can use Cauchy-Schwarz inequality for $\lambda$-weighted norm:

$$
\begin{aligned}
&\langle z^* - z^\dagger, F(z^*; \mathbf{R}) - F(z^*; \mathbf{R}') \rangle_\lambda \\
&\geq - \|\lambda\|_\infty \|z^* - z^\dagger\|_2 \|F(z^*; \mathbf{R}) - F(z^*; \mathbf{R}')\|_2
\end{aligned}
\tag{62}
$$

For the second term, $(\mu, \lambda)$-strong monotonicity yields:

$$\langle z^* - z^\dagger, F(z^*; \mathbf{R}') - F(z^\dagger; \mathbf{R}') \rangle_\lambda \geq \mu \|z^* - z^\dagger\|_2^2. \tag{63}$$

Therefore, combining the bounds above, we have:

$$\mu \|z^* - z^\dagger\|_2^2 \leq \|\lambda\|_\infty \|z^* - z^\dagger\|_2 \|F(z^*; \mathbf{R}) - F(z^*; \mathbf{R}')\|_2, \tag{64}$$

canceling out $\|z^* - z^\dagger\|_\lambda$ and rearranging terms we get:

$$\|z^* - z^\dagger\|_2 \leq \|\lambda\|_\infty \|F(z^*; \mathbf{R}) - F(z^*; \mathbf{R}')\|_2 / \mu. \tag{65}$$

Finally, by expanding $F(z^*; \mathbf{R}) - F(z^*; \mathbf{R}')$, we obtain:

$$\|F(z^*; \mathbf{R}) - F(z^*; \mathbf{R}')\|_2 = \left\| \begin{bmatrix} -(R_1 - R_1')p_1^* \\ -(R_2 - R_2')p_2^* \\ (R_1 - R_1')^T \pi_1^* \\ (R_2 - R_2')^T \pi_2^* \end{bmatrix} \right\|_2 \le 2 \left( \sqrt{|\mathcal{A}_1|} + \sqrt{|\mathcal{A}_2|} \right) \|\mathbf{R} - \mathbf{R}'\|_{\max}, \tag{66}$$

and therefore,

$$\|z^* - z^\dagger\|_2 \le \frac{2\|\lambda\|_\infty \left( \sqrt{|\mathcal{A}_1|} + \sqrt{|\mathcal{A}_2|} \right)}{\mu} \|\mathbf{R} - \mathbf{R}'\|_{\max}, \tag{67}$$

which completes the proof.

### F.2. Proof of Theorem 3.2

**Part 1: General condition.** By Lemma I.1, the monotonicity of the 4-player game (or equivalently, the monotonicity of the gradient operator $F$) can be characterized by the positivity of the operator $\Lambda \nabla F(z) + \nabla F(z)^T \Lambda$, where $\Lambda = \mathrm{Diag}(\lambda)$ is a diagonal matrix with entries identical to $\lambda$. Recall the expression (16) of the gradient operator $F$, we can write the Jacobian of $F$ as:

$$\nabla F(z) = \begin{bmatrix} \epsilon_1 \nabla^2 \nu_1 & 0 & -R_1 & 0 \\ 0 & \epsilon_2 \nabla^2 \nu_2 & 0 & -R_2 \\ R_1^T & \frac{1}{\tau_1}\nabla^2_{p\pi}D_1 & \frac{1}{\tau_1}\nabla^2_p D_1 & 0 \\ \frac{1}{\tau_2}\nabla^2_{p\pi}D_2 & R_2^T & 0 & \frac{1}{\tau_2}\nabla^2_p D_2 \end{bmatrix}, \tag{68}$$

and therefore, $\Lambda \nabla F(z)$ equals:

$$\Lambda \nabla F(z) = \begin{bmatrix} \lambda_1 \epsilon_1 \nabla^2 \nu_1 & 0 & -\lambda_1 R_1 & 0 \\ 0 & \lambda_2 \epsilon_2 \nabla^2 \nu_2 & 0 & -\lambda_2 R_2 \\ \lambda_1 R_1^T & \frac{\lambda_1}{\tau_1}\nabla^2_{p\pi}D_1 & \frac{\lambda_1}{\tau_1}\nabla^2_p D_1 & 0 \\ \frac{\lambda_2}{\tau_2}\nabla^2_{p\pi}D_2 & \lambda_2 R_2^T & 0 & \frac{\lambda_2}{\tau_2}\nabla^2_p D_2 \end{bmatrix}. \tag{69}$$

Similarly, we have:

$$\nabla F(z)^T \Lambda = \begin{bmatrix} \lambda_1 \epsilon_1 \nabla^2 \nu_1 & 0 & \lambda_1 R_1 & \frac{\lambda_2}{\tau_2}\nabla^2_{p\pi}D_2 \\ 0 & \lambda_2 \epsilon_2 \nabla^2 \nu_2 & \frac{\lambda_1}{\tau_1}\nabla^2_{p\pi}D_1 & \lambda_2 R_2 \\ -\lambda_1 R_1^T & 0 & \frac{\lambda_1}{\tau_1}\nabla^2_p D_1 & 0 \\ 0 & -\lambda_2 R_2^T & 0 & \frac{\lambda_2}{\tau_2}\nabla^2_p D_2 \end{bmatrix} \tag{70}$$

Combining the two expressions above, we obtain:

$$\Lambda \nabla F(z) + \nabla F(z)^T \Lambda = \begin{bmatrix} 2\lambda_1 \epsilon_1 \nabla^2 \nu_1 & 0 & 0 & \frac{\lambda_2}{\tau_2}\nabla^2_{p\pi}D_2 \\ 0 & 2\lambda_2 \epsilon_2 \nabla^2 \nu_2 & \frac{\lambda_1}{\tau_1}\nabla^2_{p\pi}D_1 & 0 \\ 0 & \frac{\lambda_1}{\tau_1}\nabla^2_{p\pi}D_1 & 2\frac{\lambda_1}{\tau_1}\nabla^2_p D_1 & 0 \\ \frac{\lambda_2}{\tau_2}\nabla^2_{p\pi}D_2 & 0 & 0 & 2\frac{\lambda_2}{\tau_2}\nabla^2_p D_2 \end{bmatrix}. \tag{71}$$

As a result, the condition stated in Lemma I.1 that $\Lambda \nabla F(z) + \nabla F(z)^T \Lambda \succeq 2\mu I$ becomes:

$$\begin{bmatrix} 2\lambda_1 \epsilon_1 \nabla^2 \nu_1 - 2\mu I & 0 & 0 & \frac{\lambda_2}{\tau_2}\nabla^2_{p\pi}D_2 \\ 0 & 2\lambda_2 \epsilon_2 \nabla^2 \nu_2 - 2\mu I & \frac{\lambda_1}{\tau_1}\nabla^2_{p\pi}D_1 & 0 \\ 0 & \frac{\lambda_1}{\tau_1}\nabla^2_{p\pi}D_1 & 2\frac{\lambda_1}{\tau_1}\nabla^2_p D_1 - 2\mu I & 0 \\ \frac{\lambda_2}{\tau_2}\nabla^2_{p\pi}D_2 & 0 & 0 & 2\frac{\lambda_2}{\tau_2}\nabla^2_p D_2 - 2\mu I \end{bmatrix} \succeq 0 \tag{72}$$

By rearranging rows and columns 2 and 4, we turn the matrix into block-diagonal form where positivity condition still holds:

$$\begin{bmatrix} 2\lambda_1 \epsilon_1 \nabla^2 \nu_1 - 2\mu I & \frac{\lambda_2}{\tau_2}\nabla^2_{p\pi}D_2 & 0 & 0 \\ \frac{\lambda_2}{\tau_2}\nabla^2_{p\pi}D_2 & 2\frac{\lambda_2}{\tau_2}\nabla^2_p D_2 - 2\mu I & 0 & 0 \\ 0 & 0 & 2\lambda_2 \epsilon_2 \nabla^2 \nu_2 - 2\mu I & \frac{\lambda_1}{\tau_1}\nabla^2_{p\pi}D_1 \\ 0 & 0 & \frac{\lambda_1}{\tau_1}\nabla^2_{p\pi}D_1 & 2\frac{\lambda_1}{\tau_1}\nabla^2_p D_1 - 2\mu I \end{bmatrix} \succeq 0, \tag{73}$$

this simplifies to the positivity of the two diagonal blocks:

$$\begin{bmatrix} 2\lambda_1\epsilon_1\nabla^2\nu_1 - 2\mu I & \frac{\lambda_2}{\tau_2}\nabla^2_{p\pi}D_2 \\ \frac{\lambda_2}{\tau_2}\nabla^2_{p\pi}D_2 & 2\frac{\lambda_2}{\tau_2}\nabla^2_p D_2 - 2\mu I \end{bmatrix} \succeq 0; \quad \begin{bmatrix} 2\lambda_2\epsilon_2\nabla^2\nu_2 - 2\mu I & \frac{\lambda_1}{\tau_1}\nabla^2_{p\pi}D_1 \\ \frac{\lambda_1}{\tau_1}\nabla^2_{p\pi}D_1 & 2\frac{\lambda_1}{\tau_1}\nabla^2_p D_1 - 2\mu I \end{bmatrix} \succeq 0. \tag{74}$$

This concludes the proof of part 1 in Theorem 3.2.

*More discussion:* We can see that a necessary condition for these two matrices to be positive semidefinite is:

$$\nabla^2\nu_i \succeq \frac{\mu}{\epsilon_i\lambda_i}I; \quad \nabla^2_p D_i \succeq \frac{\mu\tau_i}{\lambda_i}I, \quad i \in \{1,2\}. \tag{75}$$

Notice that we further have:

$$\begin{bmatrix} 2\lambda_i\epsilon_i\nabla^2\nu_i - 2\mu I & \frac{\lambda_{-i}}{\tau_{-i}}\nabla^2_{p\pi}D_{-i} \\ \frac{\lambda_{-i}}{\tau_{-i}}\nabla^2_{p\pi}D_{-i} & 2\frac{\lambda_{-i}}{\tau_{-i}}\nabla^2_p D_{-i} - 2\mu I \end{bmatrix}$$
$$= \frac{\lambda_{-i}}{\tau_{-i}}\begin{bmatrix} \nabla^2_\pi D_{-i} & \nabla^2_{p\pi}D_{-i} \\ \nabla^2_{p\pi}D_{-i} & \nabla^2_p D_{-i} \end{bmatrix} + \begin{bmatrix} 2\lambda_i\epsilon_i\nabla^2\nu_i - \frac{\lambda_{-i}}{\tau_{-i}}\nabla^2_\pi D_{-i} - 2\mu I & 0 \\ 0 & \frac{\lambda_{-i}}{\tau_{-i}}\nabla^2_p D_{-i} - 2\mu I \end{bmatrix}, \tag{76}$$

where the first term is always positive by joint convexity of $D_1$. Therefore, a sufficient condition for this matrix to be positive semidefinite is:

$$\frac{\lambda_{-i}}{\tau_{-i}}\nabla^2_p D_{-i} - 2\mu I \succeq 0; \quad 2\lambda_i\epsilon_i\nabla^2\nu_i - \frac{\lambda_{-i}}{\tau_{-i}}\nabla^2_\pi D_{-i} - 2\mu I \succeq 0, \quad i \in \{1,2\}. \tag{77}$$

If there exists some $\mu > 0$ such that the conditions above are satisfied, we have:

$$2\lambda_i\epsilon_i\nabla^2\nu_i - \frac{\lambda_{-i}}{\tau_{-i}}\nabla^2_\pi D_{-i} \succeq 0. \tag{78}$$

**Part 1: Log-barrier and KL** If $\nu_i(\cdot)$ are log-barrier function and $D_i(\cdot,\cdot)$ are KL divergence, we have that:

$$\nabla^2\nu_i(\pi_i) = \text{diag}(\pi_i)^{-2}; \nabla^2_{\pi_{-i}}D_i(p_i,\pi_{-i}) = \text{diag}(p_i)\text{diag}(\pi_{-i})^{-2};$$
$$\nabla^2_{p_i}D_i(p_i,\pi_{-i}) = \text{diag}(p_i)^{-1}; \nabla^2_{p_i,\pi_{-i}}D_i(p_i,\pi_{-i}) = -\text{diag}(\pi_{-i})^{-1}. \tag{79}$$

Therefore, the original condition:

$$\begin{bmatrix} 2\lambda_i\epsilon_i\nabla^2\nu_i & \frac{\lambda_{-i}}{\tau_{-i}}\nabla^2_{p\pi}D^T_{-i} \\ \frac{\lambda_{-i}}{\tau_{-i}}\nabla^2_{p\pi}D_{-i} & 2\frac{\lambda_{-i}}{\tau_{-i}}\nabla^2_p D_{-i} \end{bmatrix} \succeq 2\mu I \tag{80}$$

can be turned using Schur complement into:

$$\frac{\lambda_{-i}}{\tau_{-i}}\text{diag}(p_i)^{-1} - \mu I \succeq \frac{\lambda^2_{-i}}{4\tau^2_{-i}}\text{diag}(\pi_i)^{-1}\left(\lambda_i\epsilon_i\text{diag}(\pi_i)^{-2} - \mu I\right)^{-1}\text{diag}(\pi_i)^{-1} \tag{81}$$

this can be further rearranged into:

$$\left(\frac{\lambda_{-i}}{\tau_{-i}}\text{diag}(p_i)^{-1} - \mu I\right)\left(\lambda_i\epsilon_i\text{diag}(\pi_i)^{-2} - \mu I\right) \succeq \frac{\lambda^2_{-i}}{4\tau^2_{-i}}\text{diag}(\pi_i)^{-2}. \tag{82}$$

Solving the inequality above, we obtain the precise condition of:

$$\mu I \preceq \frac{\frac{\lambda_{-i}}{\tau_{-i}}\text{diag}(p_i)^{-1} + \lambda_i\epsilon_i\text{diag}(\pi_i)^{-2} - \sqrt{(\frac{\lambda_{-i}}{\tau_{-i}}\text{diag}(p_i)^{-1} + \lambda_i\epsilon_i\text{diag}(\pi_i)^{-2})^2 - \frac{\lambda_{-i}}{\tau_{-i}}\text{diag}(\pi_i)^{-2}(4\lambda_i\epsilon_i\text{diag}(p_i)^{-1} - \frac{\lambda_{-i}}{\tau_{-i}})}}{2} \tag{83}$$

We can see that to guarantee the right hand side of (83) to be positive, it suffices to have (taking $\mu \to 0$):

$$\lambda_i\epsilon_i\text{diag}(p_i)^{-1} \succ \frac{\lambda_{-i}}{4\tau_{-i}}I, \tag{84}$$

and since $\text{diag}(p_i) \prec I$, the condition can be simplified as:

$$4\epsilon_i\tau_{-i} \geq \frac{\lambda_{-i}}{\lambda_i}. \tag{85}$$

If we only require there exists *some* $\lambda$ satisfying this condition, we would only need:

$$4\epsilon_1\tau_2 \geq \frac{\lambda_2}{\lambda_1} \geq \frac{1}{4\epsilon_2\tau_1}, \tag{86}$$

that is, $16\epsilon_1\epsilon_2\tau_1\tau_2 > 1$. To obtain an exact lower bound on $\mu$, notice that we have $A - \sqrt{A^2 - \epsilon} > \frac{\epsilon}{2A}$, the right hand side of (83) can be lower bounded by:

$$\frac{\frac{\lambda_{-i}}{\tau_{-i}}\text{diag}(\pi_i)^{-2}(4\lambda_i\epsilon_i\text{diag}(p_i)^{-1} - \frac{\lambda_{-i}}{\tau_{-i}})}{4\left(\frac{\lambda_{-i}}{\tau_{-i}}\text{diag}(p_i)^{-1} + \lambda_i\epsilon_i\text{diag}(\pi_i)^{-2}\right)} = \frac{\frac{\lambda_{-i}}{\tau_{-i}}\left(4\lambda_i\epsilon_i - \frac{\lambda_{-i}}{\tau_{-i}}\text{diag}(p_i)\right)}{4\left(\frac{\lambda_{-i}}{\tau_{-i}}\text{diag}(\pi_i)^2 + \lambda_i\epsilon_i\text{diag}(p_i)\right)} \geq \frac{\frac{\lambda_{-i}}{\tau_{-i}}\left(4\lambda_i\epsilon_i - \frac{\lambda_{-i}}{\tau_{-i}}\right)}{4\left(\frac{\lambda_{-i}}{\tau_{-i}} + \lambda_i\epsilon_i\right)} \tag{87}$$

**Part 2: Negative entropy and reverse KL.** If $\nu_i(\cdot)$ are negative entropy and $D_i(\cdot, \cdot)$ are reverse KL divergence, we have that:

$$\nabla^2\nu_i(\pi_i) = \text{diag}(\pi_i)^{-1}; \nabla^2_{\pi_{-i}}D_i(p_i, \pi_{-i}) = \text{diag}(\pi_{-i})^{-1};$$

$$\nabla^2_{p_i}D_i(p_i, \pi_{-i}) = \text{diag}(\pi_{-i})\text{diag}(p_i)^{-2}; \nabla^2_{p_i, \pi_{-i}}D_i(p_i, \pi_{-i}) = -\text{diag}(p_i)^{-1}. \tag{88}$$

Following a similar argument we have:

$$\left(\frac{\lambda_{-i}}{\tau_{-i}}\text{diag}(\pi_i)\text{diag}(p_i)^{-2} - \mu I\right)\left(\lambda_i\epsilon_i\text{diag}(\pi_i)^{-1} - \mu I\right) \succeq \frac{\lambda^2_{-i}}{4\tau^2_{-i}}\text{diag}(p_i)^{-2} \tag{89}$$

so we get the same strict monotonicity condition $4\epsilon_i\tau_{-i} > \frac{\lambda_{-i}}{\lambda_i}$. Therefore, as long as $16\epsilon_1\epsilon_2\tau_1\tau_2 > 1$, there exists $\lambda$ such that the game is strictly monotone. However, we can see that there is no uniform $\mu$ such that the game is $(\mu, \lambda)$-strongly monotone, as it requires

$$\frac{\lambda_{-i}}{\tau_{-i}}\text{diag}(\pi_i) - \mu\text{diag}(p_i)^2 \succeq 0 \tag{90}$$

which cannot hold on the entire simplex.

# G. Proofs for Section 4

## G.1. Detailed Statement and Proof of Proposition 4.2

We now provide a more detailed version of Proposition 4.2 as follows:

**Proposition G.1** (Proposition 4.2, detailed). *If the 4-player stage game specified by* (11) *is $\mu$-strongly monotone for every state $s \in \mathcal{S}$ (notice that whether a stage game is monotone does not depend on the Q functions but only the regularizers), then we have the following:*

1. *If the regularizers $D_i(\cdot, \cdot)$ are $L_D$-Lipschitz metrics that satisfy triangle inequality, and the discount factor $\gamma$ satisfies:*

$$\gamma \leq \frac{\tau_{\min}\mu}{\tau_{\min}\mu + 2\|\lambda\|_\infty L_D(\sqrt{|\mathcal{A}_1|} + \sqrt{|\mathcal{A}_2|})}, \tag{91}$$

   *where $\tau_{\min} = \min\{\tau_1, \tau_2\}$, then the Bellman optimality operator $\mathcal{T}$ is a $\gamma_0$-contraction mapping for $\gamma_0 = \gamma\left(1 + \frac{2\|\lambda\|_\infty L_D\left(\sqrt{|\mathcal{A}_1|} + \sqrt{|\mathcal{A}_2|}\right)}{\tau_{\min}\mu}\right)$.*

2. *If $D_i(\cdot, \cdot)$ are KL-divergence and $\nu_i(\cdot)$ are log-barrier functions, given*

$$\gamma_0 = \gamma\left(1 + 2\|\lambda\|_\infty\left(|\mathcal{A}_{\max}| + \frac{\max\{\text{sp}(\mathbf{Q}), \text{sp}(\mathbf{Q}')\}}{\epsilon_{\min}}\right)\frac{\left(\sqrt{|\mathcal{A}_1|} + \sqrt{|\mathcal{A}_2|}\right)}{\tau^2_{\min}\mu}\right) < 1, \tag{92}$$

   *where $\epsilon_{\min} = \min\{\epsilon_1, \epsilon_2\}$ and $|\mathcal{A}_{\max}| = \max\{|\mathcal{A}_1|, |\mathcal{A}_2|\}$, then the Bellman optimality operator $\mathcal{T}$ is a $\gamma_0$-contraction mapping.*

*Proof.* For two $Q$ function pairs $\mathbf{Q}$ and $\mathbf{Q}'$, we have:

$$(\mathcal{T}\mathbf{Q} - \mathcal{T}\mathbf{Q}')_i(s, \mathbf{a}) \leq \gamma \mathbb{E}_{s' \sim P(\cdot|s,\mathbf{a})} \left[ \mathrm{RQE}_i(\mathbf{Q}'(s', \cdot)) - \mathrm{RQE}_i(\mathbf{Q}(s', \cdot)) \right] \tag{93}$$

For two different payoff matrices $\mathbf{R}$ and $\mathbf{R}'$, $(\pi^*, p^*)$ and $(\pi^\dagger, p^\dagger)$ are the RQEs w.r.t. $\mathbf{R}$ and $\mathbf{R}'$ respectively, we have the following bound for the RQE difference term:

$$
\begin{aligned}
&\mathrm{RQE}_i(\mathbf{R}) - \mathrm{RQE}_i(\mathbf{R}') \\
&= -(\pi_i^*)^T R_i p_i^* - \frac{1}{\tau_i} D_i(p_i^*, \pi_{-i}^*) + \epsilon_i \nu_i(\pi_i^*) + (\pi_i^\dagger)^T R_i' p_i^\dagger + \frac{1}{\tau_i} D_i(p_i^\dagger, \pi_{-i}^\dagger) - \epsilon_i \nu_i(\pi_i^\dagger) \\
&\leq -(\pi_i^\dagger)^T R_i p_i^* - \frac{1}{\tau_i} D_i(p_i^*, \pi_{-i}^*) + \epsilon_i \nu_i(\pi_i^\dagger) + (\pi_i^\dagger)^T R_i' p_i^\dagger + \frac{1}{\tau_i} D_i(p_i^\dagger, \pi_{-i}^\dagger) - \epsilon_i \nu_i(\pi_i^\dagger) \\
&= (\pi_i^\dagger)^T (R_i' p_i^\dagger - R_i p_i^*) - \frac{1}{\tau_i} D_i(p_i^*, \pi_{-i}^*) + \frac{1}{\tau_i} D_i(p_i^\dagger, \pi_{-i}^\dagger) \\
&\leq (\pi_i^\dagger)^T (R_i' - R_i) p_i^* - \frac{1}{\tau_i} D_i(p_i^*, \pi_{-i}^*) + \frac{1}{\tau_i} D_i(p_i^*, \pi_{-i}^\dagger)
\end{aligned}
$$

with the inequalities follow from Definition 2.1.

**Proof for Part 1.** We first consider the case that $D_i$ are $L_D$-Lipschitz and satisfies triangle inequality, by Theorem 3.1 we have:

$$
\begin{aligned}
&\mathrm{RQE}_i(\mathbf{R}) - \mathrm{RQE}_i(\mathbf{R}') \\
&\leq (\pi_i^\dagger)^T (R_i' - R_i) p_i^* + \frac{1}{\tau_i} D_i(\pi_{-i}^*, \pi_{-i}^\dagger) \\
&\leq (\pi_i^\dagger)^T (R_i' - R_i) p_i^* + \frac{L_D}{\tau_i} \|\pi_{-i}^* - \pi_{-i}^\dagger\|_2 \\
&\leq \left( 1 + \frac{2\|\lambda\|_\infty L_D \left( \sqrt{|\mathcal{A}_1|} + \sqrt{|\mathcal{A}_2|} \right)}{\tau_i \mu} \right) \|\mathbf{R} - \mathbf{R}'\|_{\max}.
\end{aligned}
\tag{94}
$$

Now we take max-norm with respect to all possible $(s, \mathbf{a})$ pairs in (93) and obtain:

$$\|(\mathcal{T}\mathbf{Q} - \mathcal{T}\mathbf{Q}')_i\|_{\max} \leq \gamma \left( 1 + \frac{2\|\lambda\|_\infty L_D \left( \sqrt{|\mathcal{A}_1|} + \sqrt{|\mathcal{A}_2|} \right)}{\tau_i \mu} \right) \|\mathbf{Q} - \mathbf{Q}'\|_{\max} \tag{95}$$

since this upper bound holds for both $i \in \{1, 2\}$, the left hand side can be simply rewritten as $\|\mathcal{T}\mathbf{Q} - \mathcal{T}\mathbf{Q}'\|_{\max}$, therefore $\mathcal{T}$ is a contraction when

$$\gamma \left( 1 + \frac{2\|\lambda\|_\infty L_D \left( \sqrt{|\mathcal{A}_1|} + \sqrt{|\mathcal{A}_2|} \right)}{\tau_i \mu} \right) \leq 1. \tag{96}$$

**Proof for Part 2.** For the case where $D_i$ are KL-divergence and $\nu_i$ are log-barrier functions, from Lemma I.2 we obtain that $\pi^*, \pi^\dagger$ are bounded below by $\underline{\pi} = \frac{\epsilon_i}{\epsilon_i|\mathcal{A}_i| + \max\{\mathrm{sp}(\mathbf{R}), \mathrm{sp}(\mathbf{R}')\}}$, and therefore:

$$
\begin{aligned}
&D_i(p_i^*, \pi_{-i}^*) - D_i(p_i^*, \pi_{-i}^\dagger) \\
&= (p_i^*)^T (\log \pi_{-i}^\dagger - \log \pi_{-i}^*) \\
&\leq \|p_i^*\|_2 \| \frac{\pi_{-i}^\dagger - \pi_{-i}^*}{\underline{\pi}} \|_2 \\
&\leq \frac{1}{\underline{\pi}} \|\pi_{-i}^\dagger - \pi_{-i}^*\|_2 \\
&= \left( |\mathcal{A}_i| + \frac{\max\{\mathrm{sp}(\mathbf{R}), \mathrm{sp}(\mathbf{R}')\}}{\epsilon_i} \right) \|\pi_{-i}^\dagger - \pi_{-i}^*\|_2,
\end{aligned}
\tag{97}
$$

this gives the result that:

$$\mathrm{RQE}_i(\mathbf{R}) - \mathrm{RQE}_i(\mathbf{R}')$$

$$\leq (\pi_i^\dagger)^T (R_i' - R_i) p_i^* - \frac{1}{\tau_i} D_i(p_i^*, \pi_{-i}^*) + \frac{1}{\tau_i} D_i(p_i^*, \pi_{-i}^\dagger)$$

$$\leq (\pi_i^\dagger)^T (R_i' - R_i) p_i^* + \left( \frac{|\mathcal{A}_i|}{\tau_i} + \frac{\max\{\mathrm{sp}(\mathbf{R}), \mathrm{sp}(\mathbf{R}')\}}{\epsilon_i \tau_i} \right) \|\pi_{-i}^\dagger - \pi_{-i}^*\|_2$$

$$\leq \|\mathbf{R} - \mathbf{R}'\|_{\max} + \|\lambda\|_\infty \left( \frac{|\mathcal{A}_i|}{\tau_i} + \frac{\max\{\mathrm{sp}(\mathbf{R}), \mathrm{sp}(\mathbf{R}')\}}{\epsilon_i \tau_i} \right) \frac{2\left(\sqrt{|\mathcal{A}_1|} + \sqrt{|\mathcal{A}_2|}\right)}{\tau_{\min}\mu} \|\mathbf{R} - \mathbf{R}'\|_{\max}$$

$$\leq \left( 1 + 2\|\lambda\|_\infty \left( |\mathcal{A}_i| + \frac{\max\{\mathrm{sp}(\mathbf{R}), \mathrm{sp}(\mathbf{R}')\}}{\epsilon_{\min}} \right) \frac{\left(\sqrt{|\mathcal{A}_1|} + \sqrt{|\mathcal{A}_2|}\right)}{\tau_{\min}^2\mu} \right) \|\mathbf{R} - \mathbf{R}'\|_{\max},$$

(98)

so that

$$\|\mathcal{T}\mathbf{Q} - \mathcal{T}\mathbf{Q}'\|_{\max} \leq \gamma \left( 1 + 2\|\lambda\|_\infty \left( |\mathcal{A}_{\max}| + \frac{\max\{\mathrm{sp}(\mathbf{Q}), \mathrm{sp}(\mathbf{Q}')\}}{\epsilon_{\min}} \right) \frac{\left(\sqrt{|\mathcal{A}_1|} + \sqrt{|\mathcal{A}_2|}\right)}{\tau_{\min}^2\mu} \right) \|\mathbf{Q} - \mathbf{Q}'\|_{\max}. \quad (99)$$

which completes the proof. $\qquad\square$

### G.2. Detailed Statement and Proof of Theorem 4.4

We first provide a detailed statement of Theorem 4.4 as follows:

**Theorem G.2** (Theorem 4.4, detailed). *If there exists $\gamma_0$ such that the Bellman optimality operator $\mathcal{T}$ is a contraction mapping, and additionally $D_i(\cdot, \cdot)$ are $L_D$-Lipschitz in either argument when the other is fixed, $\nu_i(\cdot)$ are $L_\nu$-Lipschitz with respect to its input, and the gradient operator of the modified 4-player game is $(\mu, \lambda)$-strongly monotone and $L_F$-Lipschitz for every state $s \in \mathcal{S}$, let*

$$C_\mathcal{T} = \gamma \left( \frac{\sqrt{|\mathcal{A}_{\max}|}}{2(1-\gamma)} \left( 1 + \gamma \left( \frac{2\sqrt{2}L_D}{\tau_{\min}} + \sqrt{2}\epsilon_{\max}L_\nu \right) \right) + \frac{L_D}{\tau_{\min}} + \max\{\frac{L_D}{\tau_{\min}}, \epsilon_{\max}L_\nu\} \right) \quad (100)$$

*then* (17) *with stepsize choice satisfying the following:*

$$\beta_t \leq \min\{\frac{1}{\mu}, \frac{\mu}{L_F^2}\};$$

$$\frac{\beta_t}{\alpha_t} \geq \left( (1-\gamma_0) + 2L_{RQE}C_\mathcal{T} \left( 1 - \gamma_0 + 2\frac{1+\gamma_0}{1-\gamma_0} \right) \right), \quad (101)$$

*where $L_{RQE}$ is the Lipschitz continuity constant of RQE with respect to the payoff matrix in normal form games as indicated by Theorem 3.1, then the sequence of $z_t$ converges to the RQE $z^*$ of the Markov game and the Q function $\mathbf{Q}_t$ converges to the corresponding $\mathbf{Q}^*$ to $z^*$ at the following rates:*

*1. If we use constant step sizes $\alpha_t = \alpha, \beta_t = \beta$, then $\forall s \in \mathcal{S}$:*

$$\|z_t(\cdot|s) - z^*(\cdot|s)\|_2 \leq \left( 1 - \frac{\tilde{\alpha}}{2} \right)^t \left( 1 + \frac{L_{RQE}}{C_2} \right) \left( \left( \sqrt{|\mathcal{A}_1|} + \sqrt{|\mathcal{A}_2|} \right) + C_2 Q_{\max} \right);$$

$$\|\mathbf{Q}_t - \mathbf{Q}^*\|_{\max} \leq \left( 1 - \frac{\tilde{\alpha}}{2} \right)^t \left( \frac{1}{C_2} \left( \sqrt{|\mathcal{A}_1|} + \sqrt{|\mathcal{A}_2|} \right) + Q_{\max} \right). \quad (102)$$

*where $\tilde{\alpha} = (1-\gamma_0)\alpha$, $C_2 = \frac{1+\gamma_0}{1-\gamma_0}L_{RQE}$, and*

$$Q_{\max} = \frac{1}{1-\gamma} + \frac{\gamma}{1-\gamma} \left( \frac{1}{\tau_{\min}}(D_{\min} + 2\sqrt{2}L_D) + \epsilon_{\max}\left( \nu_{\min} + \sqrt{2}L_\nu \right) \right). \quad (103)$$

2. *If we use diminishing step sizes $\alpha_t = \frac{\alpha}{t+h}$ and $\beta_t = \frac{\beta}{t+h}$, then $\forall s \in \mathcal{S}$:*

$$\|z_t(\cdot|s) - z^*(\cdot|s)\|_2 \leq \left(1 + \frac{L_{RQE}}{C_2}\right)\left(\left(\sqrt{|\mathcal{A}_1|} + \sqrt{|\mathcal{A}_2|}\right) + C_2 Q_{\max}\right)\left(\frac{h}{h+t+1}\right)^{\frac{\tilde{\alpha}}{2}};$$

$$\|\mathbf{Q}_t - \mathbf{Q}^*\|_{\max} \leq \left(\frac{1}{C_2}\left(\sqrt{|\mathcal{A}_1|} + \sqrt{|\mathcal{A}_2|}\right) + Q_{\max}\right)\left(\frac{h}{h+t+1}\right)^{\frac{\tilde{\alpha}}{2}}.$$

(104)

*where $\tilde{\alpha}, C_2$ and $Q_{\max}$ the same as above.*

Although the conditions of Theorem G.2 look stronger than Theorem 4.4, the condition that $D_i(\cdot, \cdot)$ and $\nu_i(\cdot)$ are both smooth indeed implies the conditions in Theorem G.2. This is because the domains of $D_i$ and $\nu_i$ are both compact sets, where smoothness implies they are Lipschitz continuous, and Lemma I.4 implies the Lipschitz continuity of $F$ as well.

*Proof of Theorem G.2.* For notational simplicity, we use $F_t(\cdot)$ to denote $F(\cdot; -\mathbf{Q}_t)$. Let $z_t^*(\cdot|s)$ denote the state-wise RQE induced by $\mathbf{Q}_t(s, \cdot)$. We first prove that the $Q$ iterations are bounded throughout:

**Lemma G.3.** *If $D_i(\cdot, \cdot)$ are $L_D$-Lipschitz and $\nu_i(\cdot)$ are $L_\nu$-Lipschitz, then we have:*

$$\mathrm{sp}(\mathbf{Q}_t) \leq Q_{span}, \quad \forall t. \quad \mathrm{sp}(\mathbf{Q}^*) \leq Q_{span}.$$

(105)

*where*

$$Q_{span} = \frac{1}{1-\gamma}\left(1 + \gamma\left(\frac{2\sqrt{2}L_D}{\tau_{\min}} + \sqrt{2}\epsilon_{\max}L_\nu\right)\right).$$

(106)

*Additionally, we have:*

$$\|\mathbf{Q}_t\|_{\max} \leq Q_{\max}, \quad \forall t. \quad \|\mathbf{Q}^*\|_{\max} \leq Q_{\max}.$$

(107)

*where*

$$Q_{\max} = \frac{1}{1-\gamma} + \frac{\gamma}{1-\gamma}\left(\frac{1}{\tau_{\min}}(D_{\min} + 2\sqrt{2}L_D) + \epsilon_{\max}\left(\nu_{\min} + \sqrt{2}L_\nu\right)\right).$$

(108)

*Proof.* We provide proofs for span and max-norm respectively.

**Result for span of $Q$.** By the update rule

$$\mathbf{Q}_{t+1} = (1-\alpha_t)\mathbf{Q}_t + \alpha_t \mathcal{T}_{z_{t+1}}\mathbf{Q}_t$$

(109)

we have that

$$\mathrm{sp}(\mathbf{Q}_{t+1}) \leq (1-\alpha_t)\mathrm{sp}(\mathbf{Q}_t) + \alpha_t \mathrm{sp}(\mathcal{T}_{z_{t+1}}\mathbf{Q}_t)$$

(110)

for the latter term, recall the definition (14) of $\mathcal{T}_{z_{t+1}}$, we have:

$$\mathrm{sp}(\mathcal{T}_{z_{t+1}}\mathbf{Q}_t)$$

$$= \max_{i,s}\left(\max_{\mathbf{a}}(\mathcal{T}_{z_{t+1}}\mathbf{Q}_t)_i(s, \mathbf{a}) - \min_{\mathbf{a}'}(\mathcal{T}_{z_{t+1}}\mathbf{Q}_t)_i(s, \mathbf{a}')\right)$$

$$\leq \max_{i,s}\left(\max_{\mathbf{a}}\{-r_i(s, \mathbf{a}) + \gamma\mathbb{E}_{s'\sim P(\cdot|s,\mathbf{a})}[\pi_{i,t+1}^T(\cdot|s')Q_{i,t}(s', \cdot)p_{i,t+1}(\cdot|s') - \frac{1}{\tau_i}D_i(p_{i,t+1}, \pi_{-i,t+1}; s') + \epsilon_i\nu_i(\pi_{i,t+1}; s')]\}\right.$$

$$\left. - \min_{\mathbf{a}'}\{-r_i(s, \mathbf{a}') + \gamma\mathbb{E}_{s'\sim P(\cdot|s,\mathbf{a}')}[\pi_{i,t+1}^T(\cdot|s')Q_{i,t}(s', \cdot)p_{i,t+1}(\cdot|s') - \frac{1}{\tau_i}D_i(p_{i,t+1}, \pi_{-i,t+1}; s') + \epsilon_i\nu_i(\pi_{i,t+1}; s')]\}\right)$$

$$\leq 1 + \gamma\max_i\left(\max_{s,\mathbf{a}}Q_{i,t}(s, \mathbf{a}) - \min_{s,\mathbf{a}}Q_{i,t}(s, \mathbf{a})\right) + \gamma\max_i\frac{1}{\tau_i}\left(\max_{p,\pi}D_i(p, \pi) - \min_{p,\pi}D_i(p, \pi)\right)$$

$$+ \gamma\max_i\epsilon_i\left(\max_\pi\nu_i(\pi) - \min_\pi\nu_i(\pi)\right)$$

(111)

Since $\nu_i(\cdot)$ are $L_\nu$-Lipschitz and $\pi \in \Delta$, we have:

$$\max_\pi \nu_i(\pi) - \min_\pi \nu_i(\pi) \leq \sqrt{2}L_\nu, \tag{112}$$

and similarly,

$$\max_{p,\pi} D_i(p,\pi) - \min_{p,\pi} D_i(p,\pi) \leq 2\sqrt{2}L_D \tag{113}$$

therefore, we have:

$$\mathrm{sp}(\mathcal{T}_{z_{t+1}}\mathbf{Q}_t) \leq 1 + \gamma\mathrm{sp}(\mathbf{Q}_t) + \gamma(\frac{2\sqrt{2}L_D}{\tau_{\min}} + \sqrt{2}\epsilon_{\max}L_\nu) \tag{114}$$

Therefore, we have:

$$\begin{aligned}
\mathrm{sp}(\mathbf{Q}_{t+1}) \leq &(1-\alpha_t)\mathrm{sp}(\mathbf{Q}_t) + \alpha_t\mathrm{sp}(\mathcal{T}_{z_{t+1}}\mathbf{Q}_t) \\
\leq &(1-(1-\gamma)\alpha_t)\mathrm{sp}(\mathbf{Q}_t) + \alpha_t\left(1 + \gamma(\frac{2\sqrt{2}L_D}{\tau_{\min}} + \sqrt{2}\epsilon_{\max}L_\nu)\right)
\end{aligned} \tag{115}$$

since $\mathbf{Q}_0 = 0$, using a simple induction argument we obtain:

$$\mathrm{sp}(\mathbf{Q}_t) \leq \frac{1}{1-\gamma}\left(1 + \gamma(\frac{2\sqrt{2}L_D}{\tau_{\min}} + \sqrt{2}\epsilon_{\max}L_\nu)\right), \quad \forall t. \tag{116}$$

For $\mathrm{sp}(\mathbf{Q}^*)$, notice that $\mathbf{Q}^* = \mathcal{T}_{z^*}\mathbf{Q}^*$ and the span bound holds for all $z_{t+1}$ gives the result.

**Result for max-norm of $Q$.** Similarly by the update rule, we only have to control $\|\mathcal{T}_{z_{t+1}}\mathbf{Q}_t\|_{\max}$. We have:

$$\begin{aligned}
&\|\mathcal{T}_{z_{t+1}}\mathbf{Q}_t\|_{\max} \\
=&\max_{i,s,\mathbf{a}}\{|-r_i(s,\mathbf{a}) + \gamma\mathbb{E}_{s'\sim P(\cdot|s,\mathbf{a})}[\pi_{i,t+1}^T(\cdot|s')Q_{i,t}(s',\cdot)p_{i,t+1}(\cdot|s') - \frac{1}{\tau_i}D_i(p_{i,t+1}, \pi_{-i,t+1}; s') + \epsilon_i\nu_i(\pi_{i,t+1}; s')]|\} \\
\leq&\gamma\left(1 + \|\mathbf{Q}_t\|_{\max} + \frac{1}{\tau_{\min}}(D_{\min} + 2\sqrt{2}L_D) + \epsilon_{\max}\left(\nu_{\min} + \sqrt{2}L_\nu\right)\right)
\end{aligned} \tag{117}$$

where we have used (112) and (113) in the inequality. Using the same reasoning as above, we have:

$$\|\mathbf{Q}_t\|_{\max} \leq \frac{1}{1-\gamma} + \frac{\gamma}{1-\gamma}\left(\frac{1}{\tau_{\min}}(D_{\min} + 2\sqrt{2}L_D) + \epsilon_{\max}\left(\nu_{\min} + \sqrt{2}L_\nu\right)\right), \tag{118}$$

and using the same argument again, we have:

$$\|\mathbf{Q}^*\|_{\max} \leq \frac{1}{1-\gamma} + \frac{\gamma}{1-\gamma}\left(\frac{1}{\tau_{\min}}(D_{\min} + 2\sqrt{2}L_D) + \epsilon_{\max}\left(\nu_{\min} + \sqrt{2}L_\nu\right)\right), \tag{119}$$

which completes the proof. $\square$

We now focus on the $z$ update. We have the following lemma on the update rule of $z_t$ for arbitrary $s \in \mathcal{S}$, where we drop the dependence on $s$ for simplicity:

**Lemma G.4.** *Consider the following update rule on $z$:*

$$z_{t+1} \leftarrow \mathrm{Proj}\left(z_t - \beta_t\Lambda F_t(z_t)\right), \tag{120}$$

*If $F_t$ is $L_F$-Lipschitz continuous and $(\mu, \lambda)$-strongly monotone in $z$, we have the following:*

$$\|z_{t+1} - z_t^*\|_2 \leq \sqrt{1 - 2\beta_t\mu + \beta_t^2 L_F^2}\|z_t - z_t^*\|_2; \tag{121}$$

$$\|z_{t+1} - z_{t+1}^*\|_2 \leq \sqrt{1 - 2\beta_t\mu + \beta_t^2 L_F^2}\|z_t - z_t^*\|_2 + \|z_{t+1}^* - z_t^*\|_2. \tag{122}$$

*Proof.* For $\|z_{t+1} - z_t^*\|_2$, we obtain by our update rule and Lemma I.3 that:

$$\|z_{t+1} - z_t^*\|_2 \leq \|\text{Proj}(z_t - \beta_t \Lambda F_t(z_t)) - \text{Proj}(z_t^* - \beta_t \Lambda F_t(z_t^*))\|_2$$
$$\leq \|z_t - z_t^* - \beta_t \Lambda (F_t(z_t) - F_t(z_t^*))\|_2 \tag{123}$$

Notice that we further have:

$$\|z_t - z_t^* - \beta_t \Lambda (F_t(z_t) - F_t(z_t^*))\|_2^2$$
$$= \|z_t - z_t^*\|_2^2 - 2\beta_t \langle z_t - z_t^*, F_t(z_t) - F_t(z_t^*) \rangle_\lambda + \beta_t^2 \|F_t(z_t) - F_t(z_t^*)\|_2^2$$
$$\leq \|z_t - z_t^*\|_2^2 - 2\beta_t \mu \|z_t - z_t^*\|_2^2 + \beta_t^2 L_F^2 \|z_t - z_t^*\|_2^2 \tag{124}$$
$$= (1 - 2\beta_t \mu + \beta_t^2 L_F^2) \|z_t - z_t^*\|_2^2$$

where we have used the $(\mu, \lambda)$-strong monotonicity of $F_t$ and $L_F$-Lipschitz continuity of $F_t$ in the inequality, we conclude that the first line of our results hold.

The second line holds by noticing that:

$$\|z_{t+1} - z_{t+1}^*\|_2 \leq \|z_{t+1} - z_t^*\|_2 + \|z_{t+1}^* - z_t^*\|_2$$
$$\leq \sqrt{1 - 2\beta_t \mu + \beta_t^2 L_F^2} \|z_t - z_t^*\|_2 + \|z_{t+1}^* - z_t^*\|_2, \tag{125}$$

which completes the proof of Lemma G.4. $\qquad\qquad\square$

Notice that $z_{t+1}^*$ is the state-wise RQE of $\mathbf{Q}_{t+1}$ and $z_t^*$ is the state-wise RQE of (the normal form) $\mathbf{Q}$, we have:

$$\|z_{t+1}^* - z_t^*\|_2 \leq L_{RQE} \|\mathbf{Q}_{t+1} - \mathbf{Q}_t\|_{\max}$$
$$= \alpha_t L_{RQE} \|\mathcal{T}_{z_{t+1}} \mathbf{Q}_t - \mathbf{Q}_t\|_{\max}$$
$$\leq \alpha_t L_{RQE} \left( \|\mathcal{T}_{z_{t+1}} \mathbf{Q}_t - \mathcal{T}\mathbf{Q}_t\|_{\max} + \|\mathcal{T}\mathbf{Q}_t - \mathbf{Q}_t\|_{\max} \right)$$
$$= \alpha_t L_{RQE} \left( \|\mathcal{T}_{z_{t+1}} \mathbf{Q}_t - \mathcal{T}_{z_t^*} \mathbf{Q}_t\|_{\max} + \|\mathcal{T}\mathbf{Q}_t - \mathbf{Q}_t\|_{\max} \right) \tag{126}$$
$$\leq \alpha_t L_{RQE} \left( \|\mathcal{T}_{z_{t+1}} \mathbf{Q}_t - \mathcal{T}_{z_t^*} \mathbf{Q}_t\|_{\max} + \|\mathcal{T}\mathbf{Q}_t - \mathcal{T}\mathbf{Q}^*\|_{\max} + \|\mathcal{T}\mathbf{Q}^* - \mathbf{Q}_t\|_{\max} \right)$$
$$\leq \alpha_t L_{RQE} \left( \|\mathcal{T}_{z_{t+1}} \mathbf{Q}_t - \mathcal{T}_{z_t^*} \mathbf{Q}_t\|_{\max} + \gamma_0 \|\mathbf{Q}_t - \mathbf{Q}^*\|_{\max} + \|\mathbf{Q}^* - \mathbf{Q}_t\|_{\max} \right)$$
$$= \alpha_t L_{RQE} \left( \|\mathcal{T}_{z_{t+1}} \mathbf{Q}_t - \mathcal{T}_{z_t^*} \mathbf{Q}_t\|_{\max} + (1 + \gamma_0) \|\mathbf{Q}_t - \mathbf{Q}^*\|_{\max} \right),$$

where $L_{RQE}$ is the Lipschitz continuity constant of RQE given by Theorem 3.1 with respect to the difference term $\|z_{t+1} - z_t\|_2$.

Now we consider the iteration step of $\mathbf{Q}$. Let $\mathbf{Q}^*$ be the $Q$ function corresponding to the RQE of the Markov game, we have:

$$\|\mathbf{Q}_{t+1} - \mathbf{Q}^*\|_{\max} \leq (1 - \alpha_t) \|\mathbf{Q}_t - \mathbf{Q}^*\|_{\max} + \alpha_t \|\mathcal{T}_{z_{t+1}} \mathbf{Q}_t - \mathcal{T}\mathbf{Q}_t\|_{\max} + \alpha_t \|\mathcal{T}\mathbf{Q}_t - \mathcal{T}\mathbf{Q}^*\|_{\max}$$
$$\leq (1 - (1 - \gamma_0)\alpha_t) \|\mathbf{Q}_t - \mathbf{Q}^*\|_{\max} + \alpha_t \|\mathcal{T}_{z_{t+1}} \mathbf{Q}_t - \mathcal{T}\mathbf{Q}_t\|_{\max} \tag{127}$$
$$= (1 - (1 - \gamma_0)\alpha_t) \|\mathbf{Q}_t - \mathbf{Q}^*\|_{\max} + \alpha_t \|\mathcal{T}_{z_{t+1}} \mathbf{Q}_t - \mathcal{T}_{z_t^*} \mathbf{Q}_t\|_{\max},$$

where we have used the $\gamma_0$-contraction property of $\mathcal{T}$ in the second inequality, and the fact that $z_t^*$ is the state-wise RQE of $\mathbf{Q}_t$. For the difference term $\|\mathcal{T}_{z_{t+1}} \mathbf{Q}_t - \mathcal{T}_{z_t^*} \mathbf{Q}_t\|_{\max}$, we have the following lemma:

**Lemma G.5.** *Given a set of $Q$ functions $\mathbf{Q}$, for arbitrary $z \in \Delta_{|\mathcal{A}_1|} \times \Delta_{|\mathcal{A}_2|} \times \Delta_{|\mathcal{A}_2|} \times \Delta_{|\mathcal{A}_1|}$, the following bound holds:*

$$\|\mathcal{T}_z \mathbf{Q} - \mathcal{T}\mathbf{Q}\|_{\max} \leq \gamma \left( \frac{\sqrt{|\mathcal{A}_{\max}|}}{2} \text{sp}(\mathbf{Q}) + \frac{L_D}{\tau_{\min}} + \max\{\frac{L_D}{\tau_{\min}}, \epsilon_{\max} L_\nu\}) \|z - z^*\|_2 \tag{128}$$

*where $z^*$ is the state-wise RQE of $\mathbf{Q}$ and $L_D, L_\nu$ are the maximum Lipschitz continuity constant of $D_i$ and $\nu_i$ respectively.*

*Proof.* Given state-action pair $(s, \mathbf{a})$, let $z = (\pi_1, \pi_2, p_1, p_2)$ and $z^* = (\pi_1^*, \pi_2^*, p_1^*, p_2^*)$, we have the following lower bound:

$$
\begin{aligned}
&(\mathcal{T}_z \mathbf{Q} - \mathcal{T}\mathbf{Q})_i(s, \mathbf{a}) \\
&= \gamma \mathbb{E}_{s' \sim P(\cdot|s,\mathbf{a})}[\pi_i^T Q_i(s', \cdot)p_i - \frac{1}{\tau_i}D_i(p_i, \pi_{-i}; s') + \epsilon_i \nu_i(\pi_i; s')] \\
&\quad - \gamma \mathbb{E}_{s' \sim P(\cdot|s,\mathbf{a})}[(\pi_i^*)^T Q_i(s', \cdot)p_i^* - \frac{1}{\tau_i}D_i(p_i^*, \pi_{-i}^*; s') + \epsilon_i \nu_i(\pi_i^*; s')] \\
&\geq \gamma \mathbb{E}_{s' \sim P(\cdot|s,\mathbf{a})}[\pi_i^T Q_i(s', \cdot)p_i - \frac{1}{\tau_i}D_i(p_i, \pi_{-i}; s') + \epsilon_i \nu_i(\pi_i; s')] \\
&\quad - \gamma \mathbb{E}_{s' \sim P(\cdot|s,\mathbf{a})}[\pi_i^T Q_i(s', \cdot)p_i^* - \frac{1}{\tau_i}D_i(p_i^*, \pi_{-i}^*; s') + \epsilon_i \nu_i(\pi_i; s')] \\
&= \gamma \mathbb{E}_{s' \sim P(\cdot|s,\mathbf{a})}[\pi_i^T Q_i(s', \cdot)(p_i - p_i^*) - \frac{1}{\tau_i}(D_i(p_i, \pi_{-i}; s') - D_i(p_i^*, \pi_{-i}^*; s'))] \\
&\geq -\gamma(\frac{\sqrt{|\mathcal{A}_{-i}|}}{2}\mathrm{sp}(Q_i) + \frac{2L_D}{\tau_i})\|z - z^*\|_2,
\end{aligned}
\tag{129}
$$

and a similar upper bound:

$$
\begin{aligned}
&(\mathcal{T}_z \mathbf{Q} - \mathcal{T}_{z^*}\mathbf{Q})_i(s, \mathbf{a}) \\
&= \gamma \mathbb{E}_{s' \sim P(\cdot|s,\mathbf{a})}[\pi_i^T Q_i(s', \cdot)p_i - \frac{1}{\tau_i}D_i(p_i, \pi_{-i}; s') + \epsilon_i \nu_i(\pi_i; s')] \\
&\quad - \gamma \mathbb{E}_{s' \sim P(\cdot|s,\mathbf{a})}[(\pi_i^*)^T Q_i(s', \cdot)p_i^* - \frac{1}{\tau_i}D_i(p_i^*, \pi_{-i}^*; s') + \epsilon_i \nu_i(\pi_i^*; s')] \\
&\leq \gamma \mathbb{E}_{s' \sim P(\cdot|s,\mathbf{a})}[\pi_i^T Q_i(s', \cdot)p_i - \frac{1}{\tau_i}D_i(p_i, \pi_{-i}; s') + \epsilon_i \nu_i(\pi_i; s')] \\
&\quad - \gamma \mathbb{E}_{s' \sim P(\cdot|s,\mathbf{a})}[(\pi_i^*)^T Q_i(s', \cdot)p_i - \frac{1}{\tau_i}D_i(p_i, \pi_{-i}^*; s') + \epsilon_i \nu_i(\pi_i^*; s')] \\
&= \gamma \mathbb{E}_{s' \sim P(\cdot|s,\mathbf{a})}[(\pi_i - \pi_i^*)^T Q_i(s', \cdot)p_i \\
&\quad - \frac{D_i(p_i, \pi_{-i}; s') - D_i(p_i, \pi_{-i}^*; s')}{\tau_i} + \epsilon_i(\nu_i(\pi_i; s') - \nu_i(\pi_i^*; s'))] \\
&\leq \gamma(\frac{\sqrt{|\mathcal{A}_i|}}{2}\mathrm{sp}(Q_i) + \frac{L_D}{\tau_i} + \epsilon_i L_\nu)\|z - z^*\|_2
\end{aligned}
\tag{130}
$$

taking max-norm on all possible $(s, \mathbf{a})$ pairs completes the proof. $\square$

Using Lemma G.5, we obtain:

$$
\begin{aligned}
\|\mathcal{T}_{z_{t+1}}\mathbf{Q}_t - \mathcal{T}_{z_t^*}\mathbf{Q}_t\|_{\max} &\leq \gamma(\frac{\sqrt{|\mathcal{A}_{\max}|}}{2}\mathrm{sp}(\mathbf{Q}_t) + \frac{L_D}{\tau_{\min}} + \max\{\frac{L_D}{\tau_{\min}}, \epsilon_{\max}L_\nu\})\|z_{t+1} - z_t^*\|_2 \\
&\leq \gamma(\frac{\sqrt{|\mathcal{A}_{\max}|}}{2}Q_{span} + \frac{L_D}{\tau_{\min}} + \max\{\frac{L_D}{\tau_{\min}}, \epsilon_{\max}L_\nu\})\|z_{t+1} - z_t^*\|_2
\end{aligned}
\tag{131}
$$

Let $C_{\mathcal{T}} = \gamma(\frac{\sqrt{|\mathcal{A}_{\max}|}}{2}Q_{span} + \frac{L_D}{\tau_{\min}} + \max\{\frac{L_D}{\tau_{\min}}, \epsilon_{\max}L_\nu\})$ denote this coefficient, our iteration dynamics can be written

as:

$$\|z_{t+1} - z_{t+1}^*\|_2$$

$$\leq \sqrt{1 - 2\beta_t\mu + \beta_t^2 L_F^2}\|z_t - z_t^*\|_2 + \alpha_t L_{RQE}\left(\|\mathcal{T}_{z_{t+1}}\mathbf{Q}_t - \mathcal{T}_{z_t^*}\mathbf{Q}_t\|_{\max} + (1 + \gamma_0)\|\mathbf{Q}_t - \mathbf{Q}^*\|_{\max}\right)$$

$$= \sqrt{1 - 2\beta_t\mu + \beta_t^2 L_F^2}\|z_t - z_t^*\|_2 + \alpha_t L_{RQE}\left(C_\mathcal{T}\|z_{t+1} - z_t^*\|_2 + (1 + \gamma_0)\|\mathbf{Q}_t - \mathbf{Q}^*\|_{\max}\right)$$

$$\leq \sqrt{1 - 2\beta_t\mu + \beta_t^2 L_F^2}\|z_t - z_t^*\|_2 \qquad (132)$$

$$+ \alpha_t L_{RQE}\left(C_\mathcal{T}\sqrt{1 - 2\beta_t\mu + \beta_t^2 L_F^2}\|z_t - z_t^*\|_2 + (1 + \gamma_0)\|\mathbf{Q}_t - \mathbf{Q}^*\|_{\max}\right)$$

$$= (1 + \alpha_t L_{RQE}C_\mathcal{T})\sqrt{1 - 2\beta_t\mu + \beta_t^2 L_F^2}\|z_t - z_t^*\|_2 + (1 + \gamma_0)\alpha_t L_{RQE}\|\mathbf{Q}_t - \mathbf{Q}^*\|_{\max}.$$

where we have used (124) in the second inequality, and

$$\|\mathbf{Q}_{t+1} - \mathbf{Q}^*\|_{\max} \leq (1 - (1 - \gamma_0)\alpha_t)\|\mathbf{Q}_t - \mathbf{Q}^*\|_{\max} + \alpha_t\|\mathcal{T}_{z_{t+1}}\mathbf{Q}_t - \mathcal{T}_{z_t^*}\mathbf{Q}_t\|_{\max}$$

$$\leq (1 - (1 - \gamma_0)\alpha_t)\|\mathbf{Q}_t - \mathbf{Q}^*\|_{\max} + \alpha_t C_\mathcal{T}\|z_{t+1} - z_t^*\|_2 \qquad (133)$$

$$= (1 - (1 - \gamma_0)\alpha_t)\|\mathbf{Q}_t - \mathbf{Q}^*\|_{\max} + \alpha_t C_\mathcal{T}\sqrt{1 - 2\beta_t\mu + \beta_t^2 L_F^2}\|z_t - z_t^*\|_2.$$

Notice that when $\beta_t \leq \min\{\frac{1}{\mu}, \frac{\mu}{L_F^2}\}$, we have:

$$\sqrt{1 - 2\beta_t\mu + \beta_t^2 L_F^2} \leq 1 - \beta_t\mu + \frac{\beta_t^2 L_F^2}{2} \leq 1 - \frac{\beta_t}{2}\mu \qquad (134)$$

Let $\tilde{\alpha}_t = (1 - \gamma_0)\alpha_t$ and $\tilde{\beta}_t = \frac{\mu}{2}\beta_t$, our recursion becomes:

$$\|z_{t+1} - z_{t+1}^*\|_2 \leq (1 + C_1\tilde{\alpha}_t)(1 - \tilde{\beta}_t)\|z_t - z_t^*\|_2 + C_2\tilde{\alpha}_t\|\mathbf{Q}_t - \mathbf{Q}^*\|_{\max} \qquad (135)$$

$$\|\mathbf{Q}_{t+1} - \mathbf{Q}^*\|_{\max} \leq (1 - \tilde{\alpha}_t)\|\mathbf{Q}_t - \mathbf{Q}^*\|_{\max} + C_3\tilde{\alpha}_t(1 - \tilde{\beta}_t)\|z_t - z_t^*\|_2 \qquad (136)$$

where $C_1 = L_{RQE}C_\mathcal{T}, C_2 = \frac{1+\gamma_0}{1-\gamma_0}L_{RQE}$ and $C_3 = \frac{C_\mathcal{T}}{1-\gamma_0}$. Let $v_t = \|\mathbf{Q}_t - \mathbf{Q}^*\|_{\max}$ and $u_t = \|z_t - z_t^*\|_2$, we can rewrite the recursion in vector form:

$$\begin{bmatrix} u_{t+1} \\ v_{t+1} \end{bmatrix} \leq \begin{bmatrix} (1 + C_1\tilde{\alpha}_t)(1 - \tilde{\beta}_t) & C_2\tilde{\alpha}_t \\ C_3\tilde{\alpha}_t(1 - \tilde{\beta}_t) & 1 - \tilde{\alpha}_t \end{bmatrix} \cdot \begin{bmatrix} u_t \\ v_t \end{bmatrix} \qquad (137)$$

we give different convergence results under different step size assumptions:

*Scenario 1:* Assume we are using constant step sizes $\tilde{\alpha}_t = \tilde{\alpha}, \tilde{\beta}_t = \tilde{\beta}, \tilde{\alpha} \ll \tilde{\beta}$. The recursion becomes:

$$\begin{bmatrix} u_{t+1} \\ v_{t+1} \end{bmatrix} \leq \begin{bmatrix} (1 + C_1\tilde{\alpha})(1 - \tilde{\beta}) & C_2\tilde{\alpha} \\ C_3\tilde{\alpha}(1 - \tilde{\beta}) & 1 - \tilde{\alpha} \end{bmatrix} \cdot \begin{bmatrix} u_t \\ v_t \end{bmatrix} \qquad (138)$$

Let $W_t = u_t + kv_t$ for some $k > 0$, we have:

$$W_{t+1} = u_{t+1} + kv_{t+1}$$

$$= \left((1 + C_1\tilde{\alpha})(1 - \tilde{\beta}) + kC_3\tilde{\alpha}(1 - \tilde{\beta})\right)u_t + (C_2\tilde{\alpha} + k(1 - \tilde{\alpha}))v_t$$

$$\leq \max\left\{(1 + C_1\tilde{\alpha})(1 - \tilde{\beta}) + kC_3\tilde{\alpha}(1 - \tilde{\beta}), \frac{C_2\tilde{\alpha}}{k} + (1 - \tilde{\alpha})\right\}(u_t + kv_t)$$

$$= \max\left\{(1 + (C_1 + kC_3)\tilde{\alpha})(1 - \tilde{\beta}), \frac{C_2\tilde{\alpha}}{k} + (1 - \tilde{\alpha})\right\}W_t \qquad (139)$$

$$\leq \max\left\{1 - \tilde{\beta} + (C_1 + kC_3)\tilde{\alpha}, 1 - (1 - \frac{C_2}{k})\tilde{\alpha}\right\}W_t$$

take $k = 2C_2$ and $\tilde{\beta} \geq \left(\frac{1}{2} + C_1 + 2C_2C_3\right)\tilde{\alpha}$, we have:

$$W_t \leq \left(1 - \frac{\tilde{\alpha}}{2}\right)^t W_0 \leq \left(1 - \frac{\tilde{\alpha}}{2}\right)^t \left(2\left(\sqrt{|\mathcal{A}_1|} + \sqrt{|\mathcal{A}_2|}\right) + k\|\mathbf{Q}^*\|_{\max}\right), \qquad (140)$$

and consequently,

$$\|z_t - z_t^*\|_2 \leq W_t \leq 2\left(1 - \frac{\tilde{\alpha}}{2}\right)^t \left(\left(\sqrt{|\mathcal{A}_1|} + \sqrt{|\mathcal{A}_2|}\right) + C_2 Q_{\max}\right); \tag{141}$$

$$\|\mathbf{Q}_t - \mathbf{Q}^*\|_{\max} \leq \frac{W_t}{k} \leq \left(1 - \frac{\tilde{\alpha}}{2}\right)^t \left(\frac{1}{C_2}\left(\sqrt{|\mathcal{A}_1|} + \sqrt{|\mathcal{A}_2|}\right) + Q_{\max}\right). \tag{142}$$

Let $z^*$ be the RQE of the Markov game, we know that $z^*$ is also the state-wise RQE of $\mathbf{Q}^*$. Theorem 3.1 implies:

$$\|z_t^* - z^*\|_2 \leq L_{RQE}\|\mathbf{Q}_t - \mathbf{Q}^*\|_{\max}, \tag{143}$$

so that

$$
\begin{aligned}
\|z_t - z^*\|_2 \leq & \|z_t - z_t^*\|_2 + \|z_t^* - z^*\|_2 \\
\leq & \left(1 - \frac{\tilde{\alpha}}{2}\right)^t \left(\left(\sqrt{|\mathcal{A}_1|} + \sqrt{|\mathcal{A}_2|}\right) + C_2 Q_{\max}\right) \\
& + L_{RQE}\left(1 - \frac{\tilde{\alpha}}{2}\right)^t \left(\frac{1}{C_2}\left(\sqrt{|\mathcal{A}_1|} + \sqrt{|\mathcal{A}_2|}\right) + \|\mathbf{Q}^*\|_{\max}\right) \\
\leq & \left(1 - \frac{\tilde{\alpha}}{2}\right)^t \left(1 + \frac{L_{RQE}}{C_2}\right)\left(\left(\sqrt{|\mathcal{A}_1|} + \sqrt{|\mathcal{A}_2|}\right) + C_2 Q_{\max}\right).
\end{aligned}
\tag{144}
$$

*Scenario 2:* Assume we are using diminishing step sizes where $\tilde{\alpha}_t = \frac{\tilde{\alpha}}{t+h}$ and $\tilde{\beta}_t = \frac{\tilde{\beta}}{t+h}, \tilde{\alpha} \ll \tilde{\beta}$ for some fixed integer $h$. The recursion now has the form:

$$
\begin{bmatrix} u_{t+1} \\ v_{t+1} \end{bmatrix} \leq \begin{bmatrix} (1 + C_1\frac{\tilde{\alpha}}{t+h})(1 - \frac{\tilde{\beta}}{t+h}) & C_2\frac{\tilde{\alpha}}{t+h} \\ C_3\frac{\tilde{\alpha}}{t+h}(1 - \frac{\tilde{\beta}}{t+h}) & 1 - \frac{\tilde{\alpha}}{t+h} \end{bmatrix} \cdot \begin{bmatrix} u_t \\ v_t \end{bmatrix}.
\tag{145}
$$

Similarly, let $W_t = u_t + kv_t$ for some $k > 0$, the recursion on $W_t$ becomes:

$$
\begin{aligned}
W_{t+1} = & u_{t+1} + kv_{t+1} \\
= & \left((1 + C_1\frac{\tilde{\alpha}}{t+h})(1 - \frac{\tilde{\beta}}{t+h}) + kC_3\frac{\tilde{\alpha}}{t+h}(1 - \frac{\tilde{\beta}}{t+h})\right)u_t + \left(C_2\frac{\tilde{\alpha}}{t+h} + k(1 - \frac{\tilde{\alpha}}{t+h})\right)v_t \\
\leq & \max\left\{(1 + C_1\frac{\tilde{\alpha}}{t+h})(1 - \frac{\tilde{\beta}}{t+h}) + kC_3\frac{\tilde{\alpha}}{t+h}(1 - \frac{\tilde{\beta}}{t+h}), \frac{C_2}{k}\frac{\tilde{\alpha}}{t+h} + (1 - \frac{\tilde{\alpha}}{t+h})\right\}(u_t + kv_t) \\
\leq & \max\left\{(1 + (C_1 + kC_3)\frac{\tilde{\alpha}}{t+h})(1 - \frac{\tilde{\beta}}{t+h}), \frac{C_2}{k}\frac{\tilde{\alpha}}{t+h} + (1 - \frac{\tilde{\alpha}}{t+h})\right\}W_t \\
\leq & \max\left\{1 - \frac{\tilde{\beta} - \tilde{\alpha}(C_1 + kC_3)}{t+h}, 1 - \frac{(1 - C_2/k)\tilde{\alpha}}{t+h}\right\}W_t.
\end{aligned}
\tag{146}
$$

Setting $k = 2C_2$ and $\tilde{\beta} \geq \left(\frac{1}{2} + C_1 + 2C_2C_3\right)\tilde{\alpha}$, we have:

$$W_{t+1} \leq \left(1 - \frac{\tilde{\alpha}}{2(t+h)}\right)W_t \tag{147}$$

and as a result,

$$
\begin{aligned}
W_{t+1} \leq & W_0 \prod_{i=0}^{t}\left(1 - \frac{\tilde{\alpha}}{2(i+h)}\right) \\
\leq & W_0 \prod_{i=0}^{t}\exp\left(-\frac{\tilde{\alpha}}{2(i+h)}\right) \\
\leq & W_0 \exp\left(-\frac{\tilde{\alpha}}{2}\sum_{i=0}^{t}\frac{1}{i+h}\right) \\
= & W_0 \exp\left(-\frac{\tilde{\alpha}}{2}(H_{h+t} - H_{h-1})\right).
\end{aligned}
\tag{148}
$$

where $H_t$ denotes the harmonic series. Since we have the following upper bound:

$$H_{h-1} - H_{h+t} \leq \log \frac{h}{h+t+1} \tag{149}$$

we obtain the upper bound of:

$$W_{t+1} \leq W_0 \left( \frac{h}{h+t+1} \right)^{\frac{\tilde{\alpha}}{2}} \tag{150}$$

which leads to the final result:

$$\|z_t - z_t^*\|_2 \leq 2 \left( \left( \sqrt{|\mathcal{A}_1|} + \sqrt{|\mathcal{A}_2|} \right) + C_2 Q_{\max} \right) \left( \frac{h}{h+t+1} \right)^{\frac{\tilde{\alpha}}{2}}; \tag{151}$$

$$\|\mathbf{Q}_t - \mathbf{Q}^*\|_{\max} \leq \left( \frac{1}{C_2} \left( \sqrt{|\mathcal{A}_1|} + \sqrt{|\mathcal{A}_2|} \right) + Q_{\max} \right) \left( \frac{h}{h+t+1} \right)^{\frac{\tilde{\alpha}}{2}}. \tag{152}$$

and:

$$\|z_t - z^*\|_2 \leq \left( 1 + \frac{L_{RQE}}{C_2} \right) \left( \left( \sqrt{|\mathcal{A}_1|} + \sqrt{|\mathcal{A}_2|} \right) + C_2 Q_{\max} \right) \left( \frac{h}{h+t+1} \right)^{\frac{\tilde{\alpha}}{2}}. \tag{153}$$

$\square$

## H. Detailed Statement and Proof of Theorem 4.7

We first present a more detailed statement of Theorem 4.7 as follows:

**Theorem H.1** (Theorem 4.7, detailed). *Under Assumptions 4.5 and 4.6, if there exists $\gamma_0$ such that the Bellman optimality operator $\mathcal{T}$ is a contraction mapping, and additionally $D_i(\cdot, \cdot)$ are $L_D$-Lipschitz in either argument when the other is fixed, $\nu_i(\cdot)$ are $L_\nu$-Lipschitz with respect to its input, and the gradient operator of the modified 4-player game is $(\mu, \lambda)$-strongly monotone and $L_F$-Lipschitz for every state $s \in \mathcal{S}$, then both on- and off-policy variants of Algorithm 1 with the following parameters:*

$$K \geq \frac{4\sqrt{2}C}{(1-\rho)(1-\gamma_0)d_{\min}}; \tag{154}$$

$$\alpha_t < 1; \tag{155}$$

$$\beta_t \leq \min\{\frac{1}{\mu}, \frac{\mu}{L_F^2}\}; \tag{156}$$

$$\frac{\beta_t}{\alpha_t} \geq \frac{(1-\gamma_0)d_{\min}/2 + \left( 1 + (1+\gamma_0)^2 + 2C_{\mathcal{T}} + \frac{8\gamma^2 C_{\mathcal{T}}^2}{d_{\min}^3} \left( \frac{1+\gamma_0}{1-\gamma_0} \right)^2 \right) L_{RQE}}{\mu + \left( 1 + (1+\gamma_0)^2 + 2C_{\mathcal{T}} + \frac{8\gamma^2 C_{\mathcal{T}}^2}{d_{\min}^3} \left( \frac{1+\gamma_0}{1-\gamma_0} \right)^2 \right) L_{RQE}}. \tag{157}$$

*then in expectation, the sequence of $z_t$ converges to the RQE $z^*$ of the Markov game and the Q function $\mathbf{Q}_t$ converges to the corresponding $\mathbf{Q}^*$ to $z^*$ at the following rates:*

1. *If we use constant step sizes $\alpha_t = \alpha, \beta_t = \beta$, then $\forall s \in \mathcal{S}$:*

$$\mathbb{E}[\|\mathbf{Q}_t - \mathbf{Q}^*\|_{\max}^2] \leq (1 - \frac{1}{2}(1-\gamma_0)d_{\min}\alpha)^t (Q_{\max}^2 + 8k) + \frac{2(C_3 + kC_6)}{(1-\gamma_0)d_{\min}}\alpha;$$

$$\mathbb{E}[\|z_t(\cdot|s) - z^*(\cdot|s)\|_2^2] \leq (2 + 2kL_{RQE}^2) \left( (1 - \frac{1}{2}(1-\gamma_0)d_{\min}\alpha)^t (Q_{\max}^2/k + 8) + \frac{2(C_3/k + C_6)}{(1-\gamma_0)d_{\min}}\alpha \right). \tag{158}$$

2. *If we use diminishing step sizes $\alpha_t = \frac{\alpha}{t+h}, \beta_t = \frac{\beta}{t+h}$ for some $h \geq 1$, then $\forall s \in \mathcal{S}$:*

$$\mathbb{E}[\|\mathbf{Q}_t - \mathbf{Q}^*\|_{\max}^2] \leq \left(Q_{\max}^2 + 8k + \frac{(C_3 + kC_6)\alpha^2}{h^2} \frac{h + 2 - \frac{1}{2}(1-\gamma_0)d_{\min}\alpha}{1 - \frac{1}{2}(1-\gamma_0)d_{\min}\alpha}\right) \left(\frac{h+1}{h+t}\right)^{\frac{1}{2}(1-\gamma_0)d_{\min}\alpha};$$

$$\mathbb{E}[\|z_t(\cdot|s) - z^*(\cdot|s)\|_2^2] \leq (2/k + 2L_{RQE}^2) \left(Q_{\max}^2 + 8k + \frac{(C_3 + kC_6)\alpha^2}{h^2} \frac{h + 2 - \frac{1}{2}(1-\gamma_0)d_{\min}\alpha}{1 - \frac{1}{2}(1-\gamma_0)d_{\min}\alpha}\right) \left(\frac{h+1}{h+t}\right)^{\frac{1}{2}(1-\gamma_0)d_{\min}\alpha}.$$

$$\tag{159}$$

*where in both cases,*

$$
\begin{aligned}
d_{\min} =& \underline{\mu}\underline{\pi}; \\
k =& \frac{(1-\gamma_0)d_{\min}}{4L_{RQE}(1+\gamma_0)^2}; \\
C_3 =& 4(1-\gamma_0)^2 \left(\frac{1}{K} + \frac{64C\rho}{(1-\rho)K} + 2\left(\frac{C}{(1-\rho)K}\right)^2\right) Q_{\max}^2 \\
& + \left(1 + \frac{1}{2}(1-\gamma_0)d_{\min}\alpha + \frac{1}{K} + \frac{64C\rho}{(1-\rho)K} + 2\left(\frac{C}{(1-\rho)K}\right)^2\right) \\
& \times 8\gamma^2 C_{\mathcal{T}}^2(1-\beta)^2 + \frac{8}{K}Q_{span}^2; \\
C_6 =& (2 + C_{\mathcal{T}})L_{RQE}^2 C_{\mathcal{T}}(1-\beta)^2 + L_{RQE}^3 C_{\mathcal{T}}^2 \alpha_t (1-\beta)^2 + 4L_{RQE}^2(1+\gamma_0)^2 Q_{\max}
\end{aligned}
$$

*here $Q_{\max}, Q_{span}$ are those provided in Lemma G.3, and $L_{RQE}, C_{\mathcal{T}}$ are the same as those in Theorem G.2.*

*Proof of Theorem H.1:*

In this section, we provide a proof of Theorem H.1 following similar ideas as that of Theorem G.2. However, since the iterates of Algorithm 1 incurs coupled random updates, we use a coupled Lyapunov drift approach following prior work in stochastic approximation that considers the Lyapunov function of $\|\mathbf{Q} - \mathbf{Q}'\|_{\max}^2$ as opposed to $\|\mathbf{Q} - \mathbf{Q}'\|_{\max}$ in the proof of Theorem G.2. We present the coupled Lyapunov drift inequalities respectively in this section.

### H.1. Lyapunov Drift Inequality for $Q$ Functions

Before presenting the Lyapunov drift inequality, we first prove a lemma that will be useful throughout:

**Lemma H.2.** *Assume the Bellman optimality operator satisfy $\gamma_0$-contraction property, the following bound holds for all $t$ and arbitrary constant $c > 0$:*

$$\|\mathcal{T}_{z_{t+1}}\mathbf{Q}_t - \mathbf{Q}_t\|_{\max}^2 \leq (1+c)\|\mathcal{T}_{z_{t+1}}\mathbf{Q}_t - \mathcal{T}_{z_t^*}\mathbf{Q}_t\|_{\max}^2 + \left(1 + \frac{1}{c}\right)(1-\gamma_0)^2\|\mathbf{Q}_t - \mathbf{Q}^*\|_{\max}^2. \tag{160}$$

*Proof.* We can write the difference term as:

$$
\begin{aligned}
&\|\mathcal{T}_{z_{t+1}}\mathbf{Q}_t - \mathbf{Q}_t\|_{\max} \\
=&\|\mathcal{T}_{z_{t+1}}\mathbf{Q}_t - \mathcal{T}_{z_t^*}\mathbf{Q}_t + \mathcal{T}_{z_t^*}\mathbf{Q}_t - \mathbf{Q}^* - \mathbf{Q}_t + \mathbf{Q}^*\|_{\max} \\
\leq&\|\mathcal{T}_{z_{t+1}}\mathbf{Q}_t - \mathcal{T}_{z_t^*}\mathbf{Q}_t\|_{\max} + \|\mathcal{T}_{z_t^*}\mathbf{Q}_t - \mathcal{T}\mathbf{Q}^* - \mathbf{Q}_t + \mathbf{Q}^*\|_{\max} \\
\leq&\|\mathcal{T}_{z_{t+1}}\mathbf{Q}_t - \mathcal{T}_{z_t^*}\mathbf{Q}_t\|_{\max} + (1-\gamma_0)\|\mathbf{Q}_t - \mathbf{Q}^*\|_{\max}
\end{aligned}
\tag{161}
$$

therefore, we can bound the square term as:

$$
\begin{aligned}
&\|\mathcal{T}_{z_{t+1}}\mathbf{Q}_t - \mathbf{Q}_t\|_{\max}^2 \\
\leq& \left(\|\mathcal{T}_{z_{t+1}}\mathbf{Q}_t - \mathcal{T}_{z_t^*}\mathbf{Q}_t\|_{\max} + (1-\gamma_0)\|\mathbf{Q}_t - \mathbf{Q}^*\|_{\max}\right)^2 \\
\leq&(1+c)\|\mathcal{T}_{z_{t+1}}\mathbf{Q}_t - \mathcal{T}_{z_t^*}\mathbf{Q}_t\|_{\max}^2 + \left(1 + \frac{1}{c}\right)(1-\gamma_0)^2\|\mathbf{Q}_t - \mathbf{Q}^*\|_{\max}^2
\end{aligned}
\tag{162}
$$

where we have used Lemma I.7 in the last inequality. $\square$

Let $\mathcal{F}_{t,k}$ denote the filtration at timestep $k$ in episode $t$ and $\mathcal{F}_t$ denote $\mathcal{F}_{t,0}$, for $w_{i,k} = \hat{q}_{i,k} - \mathcal{T}_{z_{t+1}} Q_{i,t}(s_k, \mathbf{a}_k)$, we have that $\mathbb{E}[w_{i,k}|\mathcal{F}_{t,k}] = 0$. Therefore, let $\xi_{i,k}(s,\mathbf{a})$ denote $\xi_{i,k}(s,\mathbf{a}) = w_{i,k}\mathbf{1}[(s,\mathbf{a}) = (s_k,\mathbf{a}_k)]$, for the conditional expectation given $\mathcal{F}_t$, it holds that:

$$
\begin{aligned}
\mathbb{E}[\xi_{i,k}(s,\mathbf{a})|\mathcal{F}_{t,k}] &= \mathbb{E}[w_{i,k}\mathbf{1}[(s,\mathbf{a}) = (s_k,\mathbf{a}_k)]|\mathcal{F}_{t,k}] \\
&= \mathbb{E}[w_{i,k}|\mathcal{F}_{t,k}]\mathbf{1}[(s,\mathbf{a}) = (s_k,\mathbf{a}_k)] \\
&= 0
\end{aligned}
\tag{163}
$$

and

$$
\begin{aligned}
\mathbb{E}[\xi_{i,k}(s,\mathbf{a})|\mathcal{F}_t] &= \mathbb{E}[w_{i,k}\mathbf{1}[(s,\mathbf{a}) = (s_k,\mathbf{a}_k)]|\mathcal{F}_t] \\
&= \mathbb{E}[\mathbb{E}[w_{i,k}\mathbf{1}[(s,\mathbf{a}) = (s_k,\mathbf{a}_k)]|\mathcal{F}_{t,k}]|\mathcal{F}_t] \\
&= \mathbb{E}[\mathbb{E}[w_{i,k}|\mathcal{F}_{t,k}]\mathbf{1}[(s,\mathbf{a}) = (s_k,\mathbf{a}_k)]|\mathcal{F}_t] \\
&= 0
\end{aligned}
\tag{164}
$$

therefore, $\xi_{i,k}(s,\mathbf{a})$ is a martingale difference sequence with respect to $\{\mathcal{F}_{t,k}\}$. Let $M_{i,t}(s,\mathbf{a}) = \frac{1}{K}\sum_{k=0}^{K-1}\xi_{i,k}(s,\mathbf{a})$, we can see that $\mathbb{E}[M_{i,t}(s,\mathbf{a})|\mathcal{F}_t] = 0$. For the update $\hat{\delta}_i$, we have:

$$
\begin{aligned}
&\mathbb{E}[\hat{\delta}_i(s,\mathbf{a})|\mathcal{F}_t] \\
=&\frac{1}{K}\sum_{k=0}^{K-1} \mathbb{E}\left[(\hat{q}_{i,k} - Q_{i,t}(s_k,\mathbf{a}_k))\,\mathbf{1}[(s,\mathbf{a}) = (s_k,\mathbf{a}_k)]|\mathcal{F}_t\right] \\
=&\frac{1}{K}\sum_{k=0}^{K-1} \mathbb{E}\left[\left(w_{i,k} + \mathcal{T}_{z_{t+1}}Q_{i,t}(s_k,\mathbf{a}_k) - Q_{i,t}(s_k,\mathbf{a}_k)\right)\mathbf{1}[(s,\mathbf{a}) = (s_k,\mathbf{a}_k)]\big|\mathcal{F}_t\right] \\
=&\frac{1}{K}\sum_{k=0}^{K-1} \mathbb{E}\left[\left(\mathcal{T}_{z_{t+1}}Q_{i,t}(s_k,\mathbf{a}_k) - Q_{i,t}(s_k,\mathbf{a}_k)\right)\mathbf{1}[(s,\mathbf{a}) = (s_k,\mathbf{a}_k)]\big|\mathcal{F}_t\right] + \mathbb{E}[M_{i,t}(s,\mathbf{a})|\mathcal{F}_t] \\
=&\frac{1}{K}\sum_{k=0}^{K-1} \mathbb{E}\left[\left(\mathcal{T}_{z_{t+1}}Q_{i,t}(s_k,\mathbf{a}_k) - Q_{i,t}(s_k,\mathbf{a}_k)\right)\mathbf{1}[(s,\mathbf{a}) = (s_k,\mathbf{a}_k)]\big|\mathcal{F}_t\right] \\
=&\frac{1}{K}\sum_{k=0}^{K-1} \mathbb{E}\left[\left(\mathcal{T}_{z_{t+1}}Q_{i,t}(s,\mathbf{a}) - Q_{i,t}(s,\mathbf{a})\right)\mathbf{1}[(s,\mathbf{a}) = (s_k,\mathbf{a}_k)]\big|\mathcal{F}_t\right] \\
=&\left(\mathcal{T}_{z_{t+1}}Q_{i,t}(s,\mathbf{a}) - Q_{i,t}(s,\mathbf{a})\right)\frac{1}{K}\sum_{k=0}^{K-1}\Pr((s_k,\mathbf{a}_k) = (s,\mathbf{a})|\mathcal{F}_t)
\end{aligned}
\tag{165}
$$

let $\bar{P}_K(s,\mathbf{a}|\mathcal{F}_t)$ denote $\frac{1}{K}\sum_{k=0}^{K-1}\Pr((s_k,\mathbf{a}_k) = (s,\mathbf{a})|\mathcal{F}_t)$, we conclude that

$$
\mathbb{E}[\hat{\delta}_i(s,\mathbf{a})|\mathcal{F}_t] = \bar{P}_K(s,\mathbf{a}|\mathcal{F}_t)\left(\mathcal{T}_{z_{t+1}}Q_{i,t}(s,\mathbf{a}) - Q_{i,t}(s,\mathbf{a})\right)
\tag{166}
$$

**Critic Update Decomposition.** We first provide a decomposition result for $(Q_{i,t+1}(s,\mathbf{a}) - Q_i^*(s,\mathbf{a}))^2$, notice that the update rule of Algorithm 1 can be rewritten as:

$$
\begin{aligned}
Q_{i,t+1}(s,\mathbf{a}) =& Q_{i,t}(s,\mathbf{a}) + \alpha_t \hat{\delta}_i(s,\mathbf{a}) \\
=& Q_{i,t}(s,\mathbf{a}) + \alpha_t \frac{1}{K} \sum_{k=0}^{K-1} (\hat{q}_{i,k} - Q_{i,t}(s_k,\mathbf{a}_k)) \mathbf{1}[(s,\mathbf{a}) = (s_k,\mathbf{a}_k)] \\
=& Q_{i,t}(s,\mathbf{a}) + \alpha_t \frac{1}{K} \sum_{k=0}^{K-1} (w_{i,k} + \mathcal{T}_{z_{t+1}} Q_{i,t}(s_k,\mathbf{a}_k) - Q_{i,t}(s_k,\mathbf{a}_k)) \mathbf{1}[(s,\mathbf{a}) = (s_k,\mathbf{a}_k)] \\
=& Q_{i,t}(s,\mathbf{a}) + \alpha_t \frac{1}{K} \sum_{k=0}^{K-1} \xi_{i,k}(s,\mathbf{a}) \\
& + \alpha_t \frac{1}{K} \sum_{k=0}^{K-1} (\mathcal{T}_{z_{t+1}} Q_{i,t}(s_k,\mathbf{a}_k) - Q_{i,t}(s_k,\mathbf{a}_k)) \mathbf{1}[(s,\mathbf{a}) = (s_k,\mathbf{a}_k)] \\
=& Q_{i,t}(s,\mathbf{a}) + \alpha_t \mathbb{E}[\hat{\delta}_i(s,\mathbf{a})|\mathcal{F}_t] \\
& + \alpha_t \frac{1}{K} \sum_{k=0}^{K-1} (\mathcal{T}_{z_{t+1}} Q_{i,t}(s_k,\mathbf{a}_k) - Q_{i,t}(s_k,\mathbf{a}_k)) \mathbf{1}[(s,\mathbf{a}) = (s_k,\mathbf{a}_k)] - \alpha_t \mathbb{E}[\hat{\delta}_i(s,\mathbf{a})|\mathcal{F}_t] \\
& + \alpha_t \frac{1}{K} \sum_{k=0}^{K-1} \xi_{i,k}(s,\mathbf{a})
\end{aligned}
\tag{167}
$$

since we have that

$$
\begin{aligned}
& \mathbb{E}\left[ \left(Q_{i,t}(s,\mathbf{a}) + \alpha_t \mathbb{E}[\hat{\delta}_i(s,\mathbf{a})|\mathcal{F}_t] - Q_i^*(s,\mathbf{a})\right) \left(\alpha_t \frac{1}{K} \sum_{k=0}^{K-1} \xi_{i,k}(s,\mathbf{a})\right) \middle| \mathcal{F}_t \right] \\
=& \left(Q_{i,t}(s,\mathbf{a}) + \alpha_t \mathbb{E}[\hat{\delta}_i(s,\mathbf{a})|\mathcal{F}_t] - Q_i^*(s,\mathbf{a})\right) \mathbb{E}\left[ \left(\alpha_t \frac{1}{K} \sum_{k=0}^{K-1} \xi_{i,k}(s,\mathbf{a})\right) \middle| \mathcal{F}_t \right] \\
=& \left(Q_{i,t}(s,\mathbf{a}) + \alpha_t \mathbb{E}[\hat{\delta}_i(s,\mathbf{a})|\mathcal{F}_t] - Q_i^*(s,\mathbf{a})\right) \alpha_t \frac{1}{K} \sum_{k=0}^{K-1} \mathbb{E}[\xi_{i,k}(s,\mathbf{a})|\mathcal{F}_t] \\
=& 0;
\end{aligned}
\tag{168}
$$

and

$$
\begin{aligned}
& \mathbb{E}\left[ \left(Q_{i,t}(s,\mathbf{a}) + \alpha_t \mathbb{E}[\hat{\delta}_i(s,\mathbf{a})|\mathcal{F}_t] - Q_i^*(s,\mathbf{a})\right) \times \right. \\
& \left. \left(\alpha_t \frac{1}{K} \sum_{k=0}^{K-1} (\mathcal{T}_{z_{t+1}} Q_{i,t}(s_k,\mathbf{a}_k) - Q_{i,t}(s_k,\mathbf{a}_k)) \mathbf{1}[(s,\mathbf{a}) = (s_k,\mathbf{a}_k)] - \alpha_t \mathbb{E}[\hat{\delta}_i(s,\mathbf{a})|\mathcal{F}_t]\right) \middle| \mathcal{F}_t \right] \\
=& \left(Q_{i,t}(s,\mathbf{a}) + \alpha_t \mathbb{E}[\hat{\delta}_i(s,\mathbf{a})|\mathcal{F}_t] - Q_i^*(s,\mathbf{a})\right) \times \\
& \mathbb{E}\left[ \alpha_t \frac{1}{K} \sum_{k=0}^{K-1} (\mathcal{T}_{z_{t+1}} Q_{i,t}(s_k,\mathbf{a}_k) - Q_{i,t}(s_k,\mathbf{a}_k)) \mathbf{1}[(s,\mathbf{a}) = (s_k,\mathbf{a}_k)] - \alpha_t \mathbb{E}[\hat{\delta}_i(s,\mathbf{a})|\mathcal{F}_t] \middle| \mathcal{F}_t \right] \\
=& \left(Q_{i,t}(s,\mathbf{a}) + \alpha_t \mathbb{E}[\hat{\delta}_i(s,\mathbf{a})|\mathcal{F}_t] - Q_i^*(s,\mathbf{a})\right) \times \\
& \left(\alpha_t \frac{1}{K} \sum_{k=0}^{K-1} \mathbb{E}\left[(\mathcal{T}_{z_{t+1}} Q_{i,t}(s_k,\mathbf{a}_k) - Q_{i,t}(s_k,\mathbf{a}_k)) \mathbf{1}[(s,\mathbf{a}) = (s_k,\mathbf{a}_k)]\middle|\mathcal{F}_t\right] - \alpha_t \mathbb{E}[\hat{\delta}_i(s,\mathbf{a})|\mathcal{F}_t]\right) \\
=& 0.
\end{aligned}
\tag{169}
$$

We can rewrite the squared difference term $(Q_{i,t+1}(s, \mathbf{a}) - Q_i^*(s, \mathbf{a}))^2$ into the sum of 4 different terms:

$$
\mathbb{E}\left[(Q_{i,t+1}(s, \mathbf{a}) - Q_i^*(s, \mathbf{a}))^2 \middle| \mathcal{F}_t\right]
$$

$$
= \underbrace{\mathbb{E}\left[\left(Q_{i,t}(s, \mathbf{a}) + \alpha_t \mathbb{E}[\hat{\delta}_i(s, \mathbf{a})|\mathcal{F}_t] - Q_i^*(s, \mathbf{a})\right)^2 \middle| \mathcal{F}_t\right]}_{\mathcal{L}_1'}
$$

$$
+ \underbrace{\mathbb{E}\left[\left(\alpha_t \frac{1}{K}\sum_{k=0}^{K-1}\left(\mathcal{T}_{z_{t+1}}Q_{i,t}(s_k, \mathbf{a}_k) - Q_{i,t}(s_k, \mathbf{a}_k)\right)\mathbf{1}[(s, \mathbf{a}) = (s_k, \mathbf{a}_k)] - \alpha_t \mathbb{E}[\hat{\delta}_i(s, \mathbf{a})|\mathcal{F}_t]\right)^2 \middle| \mathcal{F}_t\right]}_{\mathcal{L}_2'}
$$

$$
+ \underbrace{\mathbb{E}\left[\left(\alpha_t \frac{1}{K}\sum_{k=0}^{K-1}\xi_{i,k}(s, \mathbf{a})\right)^2 \middle| \mathcal{F}_t\right]}_{\mathcal{L}_3'} \tag{170}
$$

$$
+ 2\mathbb{E}\Bigg[\left(\alpha_t \frac{1}{K}\sum_{k=0}^{K-1}\xi_{i,k}(s, \mathbf{a})\right) \times
$$

$$
\underbrace{\left(\alpha_t \frac{1}{K}\sum_{k=0}^{K-1}\left(\mathcal{T}_{z_{t+1}}Q_{i,t}(s_k, \mathbf{a}_k) - Q_{i,t}(s_k, \mathbf{a}_k)\right)\mathbf{1}[(s, \mathbf{a}) = (s_k, \mathbf{a}_k)] - \alpha_t \mathbb{E}[\hat{\delta}_i(s, \mathbf{a})|\mathcal{F}_t]\right) \middle| \mathcal{F}_t\Bigg]}_{\mathcal{L}_4'}
$$

Here $\mathcal{L}_1'$ indicates the expected update of $Q_i$ conditioned on $\mathcal{F}_t$, $\mathcal{L}_2'$, $\mathcal{L}_3'$ and $\mathcal{L}_4'$ are error terms induced by the stochasticity induced by sampling in the update rule of Algorithm 1. We can take max-norm on both sides of (170) and get:

$$
\mathbb{E}\left[\|\mathbf{Q}_{t+1} - \mathbf{Q}^*\|_{\max}^2 \middle| \mathcal{F}_t\right]
$$

$$
\leq \underbrace{\mathbb{E}\left[\left\|\mathbf{Q}_t + \alpha_t \mathbb{E}[\hat{\boldsymbol{\delta}}|\mathcal{F}_t] - \mathbf{Q}^*\right\|_{\max}^2 \middle| \mathcal{F}_t\right]}_{\mathcal{L}_1}
$$

$$
+ \underbrace{\mathbb{E}\left[\left\|\alpha_t \frac{1}{K}\sum_{k=0}^{K-1}\left(\mathcal{T}_{z_{t+1}}\mathbf{Q}_t(s_k, \mathbf{a}_k) - \mathbf{Q}_t(s_k, \mathbf{a}_k)\right)\otimes e_{(s_k, \mathbf{a}_k)} - \alpha_t \mathbb{E}[\hat{\boldsymbol{\delta}}|\mathcal{F}_t]\right\|_{\max}^2 \middle| \mathcal{F}_t\right]}_{\mathcal{L}_2} \tag{171}
$$

$$
+ \underbrace{\mathbb{E}\left[\left\|\alpha_t \frac{1}{K}\sum_{k=0}^{K-1}\boldsymbol{\xi}_k\right\|_{\max}^2 \middle| \mathcal{F}_t\right]}_{\mathcal{L}_3}
$$

$$
+ 2\underbrace{\mathbb{E}\left[\left\|\alpha_t \frac{1}{K}\sum_{k=0}^{K-1}\left(\mathcal{T}_{z_{t+1}}\mathbf{Q}_t(s_k, \mathbf{a}_k) - \mathbf{Q}_t(s_k, \mathbf{a}_k)\right)\otimes e_{(s_k, \mathbf{a}_k)} - \alpha_t \mathbb{E}[\hat{\boldsymbol{\delta}}|\mathcal{F}_t]\right\|_{\max} \times \left\|\alpha_t \frac{1}{K}\sum_{k=0}^{K-1}\boldsymbol{\xi}_k\right\|_{\max} \middle| \mathcal{F}_t\right]}_{\mathcal{L}_4}
$$

To obtain a Lyapunov drift inequality on $\|\mathbf{Q}_t - \mathbf{Q}^*\|_{\max}^2$, we only have to bound $\mathcal{L}_1, \mathcal{L}_2, \mathcal{L}_3$ and $\mathcal{L}_4$, and each $\mathcal{L}_i$ term can be bounded individually, as we will show as follows.

**Further decomposing the expected update term $\mathcal{L}_1$.** Although the update expressed in term $\mathcal{L}_1$ is no longer stochastic conditioned on $\mathcal{F}_t$, it is not directly tied to the Bellman operator $\mathcal{T}_{z_{t+1}}$ and therefore cannot directly be processed by a contraction argument. Therefore, we have to further decompose $\mathcal{L}_1$ into terms that we can directly bound. To accommodate

a cleaner representation, notice that

$$
\begin{aligned}
&Q_{i,t}(s,\mathbf{a}) + \alpha_t \mathbb{E}[\hat{\delta}_i(s,\mathbf{a})|\mathcal{F}_t] - Q_i^*(s,\mathbf{a}) \\
=&Q_{i,t}(s,\mathbf{a}) + \alpha_t d_{t+1}(s,\mathbf{a})\left(\mathcal{T}_{z_{t+1}}Q_{i,t}(s,\mathbf{a}) - Q_{i,t}(s,\mathbf{a})\right) - Q_i^*(s,\mathbf{a}) \\
&+ \alpha_t \mathbb{E}[\hat{\delta}_i(s,\mathbf{a})|\mathcal{F}_t] - \alpha_t d_{t+1}(s,\mathbf{a})\left(\mathcal{T}_{z_{t+1}}Q_{i,t}(s,\mathbf{a}) - Q_{i,t}(s,\mathbf{a})\right)
\end{aligned}
\tag{172}
$$

we have the following decomposition for $\mathcal{L}_1'$:

$$
\mathcal{L}_1' = \mathcal{L}_{1,1}' + 2\alpha_t \mathcal{L}_{1,2}' + \alpha_t^2 \mathcal{L}_{1,3}'
\tag{173}
$$

where

$$
\mathcal{L}_{1,1}' = \mathbb{E}\left[\left(Q_{i,t}(s,\mathbf{a}) + \alpha_t d_{t+1}(s,\mathbf{a})\left(\mathcal{T}_{z_{t+1}}Q_{i,t}(s,\mathbf{a}) - Q_{i,t}(s,\mathbf{a})\right) - Q_i^*(s,\mathbf{a})\right)^2 \Big|\mathcal{F}_t\right];
\tag{174}
$$

$$
\begin{aligned}
\mathcal{L}_{1,2}' = \mathbb{E}\Big[&\left(Q_{i,t}(s,\mathbf{a}) + \alpha_t d_{t+1}(s,\mathbf{a})\left(\mathcal{T}_{z_{t+1}}Q_{i,t}(s,\mathbf{a}) - Q_{i,t}(s,\mathbf{a})\right) - Q_i^*(s,\mathbf{a})\right) \\
&\times \left(\mathbb{E}[\hat{\delta}_i(s,\mathbf{a})|\mathcal{F}_t] - d_{t+1}(s,\mathbf{a})\left(\mathcal{T}_{z_{t+1}}Q_{i,t}(s,\mathbf{a}) - Q_{i,t}(s,\mathbf{a})\right)\right)\Big|\mathcal{F}_t\Big];
\end{aligned}
\tag{175}
$$

$$
\mathcal{L}_{1,3}' = \mathbb{E}\left[\left(\mathbb{E}[\hat{\delta}_i(s,\mathbf{a})|\mathcal{F}_t] - d_{t+1}(s,\mathbf{a})\left(\mathcal{T}_{z_{t+1}}Q_{i,t}(s,\mathbf{a}) - Q_{i,t}(s,\mathbf{a})\right)\right)^2 \Big|\mathcal{F}_t\right].
\tag{176}
$$

here $d_{t+1}$ is the steady state-action distribution induced by the sampling policy at time step $t+1$: For on-policy updates,

$$
d_{t+1}(s,\mathbf{a}) = \mu_{\pi_{t+1}}(s)\pi_{t+1}(\mathbf{a}|s).
\tag{177}
$$

and for off-policy updates,

$$
d_{t+1}(s,\mathbf{a}) = \mu_{\pi^r}(s)\pi^r(\mathbf{a}|s).
\tag{178}
$$

for the corresponding state distribution $\mu_{\pi_{t+1}}$ or $\mu_{\pi^r}$. Taking max-norm of each term above, we obtain the following bound:

$$
\mathcal{L}_1 \leq \mathcal{L}_{1,1} + 2\alpha_t \mathcal{L}_{1,2} + \alpha_t^2 \mathcal{L}_{1,3}
\tag{179}
$$

where

$$
\mathcal{L}_{1,1} = \mathbb{E}\left[\left\|\mathbf{Q}_t + \alpha_t \mathbf{d}_{t+1} \odot \left(\mathcal{T}_{z_{t+1}}\mathbf{Q}_t - \mathbf{Q}_t\right) - \mathbf{Q}^*\right\|_{\max}^2 \Big|\mathcal{F}_t\right];
\tag{180}
$$

$$
\mathcal{L}_{1,2} = \mathbb{E}\left[\left\|\mathbf{Q}_t + \alpha_t \mathbf{d}_{t+1} \odot \left(\mathcal{T}_{z_{t+1}}\mathbf{Q}_t - \mathbf{Q}_t\right) - \mathbf{Q}^*\right\|_{\max} \times \left\|\mathbb{E}[\hat{\boldsymbol{\delta}}|\mathcal{F}_t] - \mathbf{d}_{t+1} \odot \left(\mathcal{T}_{z_{t+1}}\mathbf{Q}_t - \mathbf{Q}_t\right)\right\|_{\max} \Big|\mathcal{F}_t\right];
\tag{181}
$$

$$
\mathcal{L}_{1,3} = \mathbb{E}\left[\left\|\mathbb{E}[\hat{\boldsymbol{\delta}}|\mathcal{F}_t] - \mathbf{d}_{t+1} \odot \left(\mathcal{T}_{z_{t+1}}\mathbf{Q}_t - \mathbf{Q}_t\right)\right\|_{\max}^2 \Big|\mathcal{F}_t\right].
\tag{182}
$$

Notice that we have the following decomposition:

$$
\begin{aligned}
&Q_{i,t}(s,\mathbf{a}) + \alpha_t d_{t+1}(s,\mathbf{a})\left(\mathcal{T}_{z_{t+1}}Q_{i,t}(s,\mathbf{a}) - Q_{i,t}(s,\mathbf{a})\right) - Q_i^*(s,\mathbf{a}) \\
=&(1 - \alpha_t d_{t+1}(s,\mathbf{a}))(Q_{i,t}(s,\mathbf{a}) - Q_i^*(s,\mathbf{a})) + \alpha_t d_{t+1}(s,\mathbf{a})\left(\mathcal{T}_{z_{t+1}}Q_{i,t}(s,\mathbf{a}) - \mathcal{T}_{z^*}Q_i^*(s,\mathbf{a})\right) \\
=&(1 - \alpha_t d_{t+1}(s,\mathbf{a}))(Q_{i,t}(s,\mathbf{a}) - Q_i^*(s,\mathbf{a})) \\
&+ \alpha_t d_{t+1}(s,\mathbf{a})\left(\mathcal{T}_{z_{t+1}}Q_{i,t}(s,\mathbf{a}) - \mathcal{T}_{z_t^*}Q_{i,t}(s,\mathbf{a}) + \mathcal{T}_{z_t^*}Q_{i,t}(s,\mathbf{a}) - \mathcal{T}_{z^*}Q_i^*(s,\mathbf{a})\right) \\
=&(1 - \alpha_t d_{t+1}(s,\mathbf{a}))(Q_{i,t}(s,\mathbf{a}) - Q_i^*(s,\mathbf{a})) + \alpha_t d_{t+1}(s,\mathbf{a})(\mathcal{T}Q_{i,t}(s,\mathbf{a}) - \mathcal{T}Q_i^*(s,\mathbf{a})) \\
&+ \alpha_t d_{t+1}(s,\mathbf{a})\left(\mathcal{T}_{z_{t+1}}Q_{i,t}(s,\mathbf{a}) - \mathcal{T}_{z_t^*}Q_{i,t}(s,\mathbf{a})\right)
\end{aligned}
\tag{183}
$$

where we have used the fact that $\mathcal{T}_{z_t^*}Q_{i,t} = \mathcal{T}Q_{i,t}, \mathcal{T}_{z^*}Q_i^* = \mathcal{T}Q_i^*$ in the last equality. Therefore, we have:

$$
\begin{aligned}
&\left\|\mathbf{Q}_t + \alpha_t \mathbf{d}_{t+1} \odot \left(\mathcal{T}_{z_{t+1}}\mathbf{Q}_t - \mathbf{Q}_t\right) - \mathbf{Q}^*\right\|_{\max} \\
=&\max_{i,s,\mathbf{a}}\{Q_{i,t}(s,\mathbf{a}) + \alpha_t d_{t+1}(s,\mathbf{a})\left(\mathcal{T}_{z_{t+1}}Q_{i,t}(s,\mathbf{a}) - Q_{i,t}(s,\mathbf{a})\right) - Q_i^*(s,\mathbf{a})\} \\
=&\max_{i,s,\mathbf{a}}\{(1 - \alpha_t d_{t+1}(s,\mathbf{a}))(Q_{i,t}(s,\mathbf{a}) - Q_i^*(s,\mathbf{a})) + \alpha_t d_{t+1}(s,\mathbf{a})(\mathcal{T}Q_{i,t}(s,\mathbf{a}) - \mathcal{T}Q_i^*(s,\mathbf{a})) \\
&+ \alpha_t d_{t+1}(s,\mathbf{a})\left(\mathcal{T}_{z_{t+1}}Q_{i,t}(s,\mathbf{a}) - \mathcal{T}_{z_t^*}Q_{i,t}(s,\mathbf{a})\right)\} \\
\leq&(1 - \alpha_t d_{t+1,\min})\|\mathbf{Q}_t - \mathbf{Q}^*\|_{\max} + \gamma_0 \alpha_t d_{t+1,\max}\|\mathbf{Q}_t - \mathbf{Q}^*\|_{\max} + \alpha_t d_{t+1,\max}\|\mathcal{T}_{z_{t+1}}\mathbf{Q}_t - \mathcal{T}_{z_t^*}\mathbf{Q}_t\|_{\max} \\
\leq&(1 - (1 - \gamma_0)\alpha_t d_{t+1,\min})\|\mathbf{Q}_t - \mathbf{Q}^*\|_{\max} + \alpha_t d_{t+1,\max}\|\mathcal{T}_{z_{t+1}}\mathbf{Q}_t - \mathcal{T}_{z_t^*}\mathbf{Q}_t\|_{\max}
\end{aligned}
\tag{184}
$$

where $d_{t+1,\min} = \min_{s,\mathbf{a}} d_{t+1}(s, \mathbf{a})$ and $d_{t+1,\max} = \max_{s,\mathbf{a}} d_{t+1}(s, \mathbf{a})$. We can now rewrite the non-stochastic drift term $\mathcal{L}_{1,1}$ as:

$$
\begin{aligned}
&\mathcal{L}_{1,1} \\
&= \mathbb{E}\left[\left\|\mathbf{Q}_t + \alpha_t \mathbf{d}_{t+1} \odot \left(\mathcal{T}_{z_{t+1}}\mathbf{Q}_t - \mathbf{Q}_t\right) - \mathbf{Q}^*\right\|_{\max}^2 \Big| \mathcal{F}_t\right] \\
&\leq \left((1 - (1 - \gamma_0)\alpha_t d_{t+1,\min})\|\mathbf{Q}_t - \mathbf{Q}^*\|_{\max} + \alpha_t d_{t+1,\max}\|\mathcal{T}_{z_{t+1}}\mathbf{Q}_t - \mathcal{T}_{z_t^*}\mathbf{Q}_t\|_{\max}\right)^2 \\
&\leq (1 + c_1)(1 - (1 - \gamma_0)\alpha_t d_{t+1,\min})^2\|\mathbf{Q}_t - \mathbf{Q}^*\|_{\max}^2 + \left(1 + \frac{1}{c_1}\right)\alpha_t^2 d_{t+1,\max}^2 \|\mathcal{T}_{z_{t+1}}\mathbf{Q}_t - \mathcal{T}_{z_t^*}\mathbf{Q}_t\|_{\max}^2
\end{aligned}
\tag{185}
$$

where we have used (184) in the first inequality and Lemma I.7 in the second inequality. Recall Lemma G.4 and Lemma G.5, we have that (notice that we are still taking maximum over all $s \in \mathcal{S}$ for $z$, such that $\|z_t - z_t^*\|_2 = \max_{s \in \mathcal{S}}\|z_t(\cdot|s) - z_t^*(\cdot|s)\|_2$):

$$
\|\mathcal{T}_{z_{t+1}}\mathbf{Q}_t - \mathcal{T}_{z_t^*}\mathbf{Q}_t\|_{\max}^2 \leq \gamma^2 C_{\mathcal{T}}^2\|z_{t+1} - z_t^*\|_2^2 \leq \gamma^2 C_{\mathcal{T}}^2(1 - 2\beta_t\mu + \beta_t^2 L_F^2)\|z_t - z_t^*\|_2^2.
\tag{186}
$$

Combining (185) and (186) we obtain:

$$
\begin{aligned}
\mathcal{L}_{1,1} \leq &(1 + c_1)(1 - (1 - \gamma_0)\alpha_t d_{t+1,\min})^2\|\mathbf{Q}_t - \mathbf{Q}^*\|_{\max}^2 \\
&+ \left(1 + \frac{1}{c_1}\right)\alpha_t^2 d_{t+1,\max}^2 \gamma^2 C_{\mathcal{T}}^2(1 - 2\beta_t\mu + \beta_t^2 L_F^2)\|z_t - z_t^*\|_2^2.
\end{aligned}
\tag{187}
$$

For term $\mathcal{L}_{1,2}$, recall that $\bar{P}_K(s, \mathbf{a}|\mathcal{F}_t) = \frac{1}{K}\sum_{k=0}^{K-1}\Pr((s_k, \mathbf{a}_k) = (s, \mathbf{a})|\mathcal{F}_t)$, by Assumption 4.5, we know that $\bar{P}_K(s, \mathbf{a}|\mathcal{F}_t)$ converges to $d_{t+1}$ with a rate of (here we focus on the on-policy case, the same proof goes through for the off-policy case as well):

$$
\begin{aligned}
|\Pr((s_k, \mathbf{a}_k) = (s, \mathbf{a})|\mathcal{F}_t) - d_{t+1}(s, \mathbf{a})| &= |P_{\pi_{t+1}}^k(s_0, s|\mathcal{F}_t)\pi_{t+1}(\mathbf{a}|s) - d_{t+1}(s, \mathbf{a})| \\
&\leq \|P_{\pi_{t+1}}^k(s_0, s|\mathcal{F}_t) - \mu_{\pi_{t+1}}(\cdot)\|_{\mathrm{TV}} \\
&\leq C\rho^k
\end{aligned}
\tag{188}
$$

and therefore:

$$
\begin{aligned}
&|\bar{P}_K(s, \mathbf{a}|\mathcal{F}_t) - d_{t+1}(s, \mathbf{a})| \\
&= |\frac{1}{K}\sum_{k=0}^{K-1}\Pr((s_k, \mathbf{a}_k) = (s, \mathbf{a})|\mathcal{F}_t) - d_{t+1}(s, \mathbf{a})| \\
&\leq \frac{1}{K}\sum_{k=0}^{K-1}|\Pr((s_k, \mathbf{a}_k) = (s, \mathbf{a})|\mathcal{F}_t) - d_{t+1}(s, \mathbf{a})| \\
&\leq \frac{C}{K}\sum_{k=0}^{K-1}\rho^k \\
&\leq \frac{C}{K}\frac{1 - \rho^K}{1 - \rho} \\
&\leq \frac{C}{(1 - \rho)K}.
\end{aligned}
\tag{189}
$$

Recall (166), we have the following upper bound on $\mathcal{L}_{1,2}$:

$$\mathcal{L}_{1,2}$$
$$=\mathbb{E}\Big[\big\|\mathbf{Q}_t + \alpha_t \mathbf{d}_{t+1} \odot (\mathcal{T}_{z_{t+1}}\mathbf{Q}_t - \mathbf{Q}_t) - \mathbf{Q}^*\big\|_{\max} \times \big\|\mathbb{E}[\hat{\boldsymbol{\delta}}|\mathcal{F}_t] - \mathbf{d}_{t+1} \odot (\mathcal{T}_{z_{t+1}}\mathbf{Q}_t - \mathbf{Q}_t)\big\|_{\max}\Big|\mathcal{F}_t\Big]$$
$$=\mathbb{E}\Big[\big\|\mathbf{Q}_t + \alpha_t \mathbf{d}_{t+1} \odot (\mathcal{T}_{z_{t+1}}\mathbf{Q}_t - \mathbf{Q}_t) - \mathbf{Q}^*\big\|_{\max} \times \big\|(\bar{P}_K(\cdot,\cdot|\mathcal{F}_t) - \mathbf{d}_{t+1}) \odot (\mathcal{T}_{z_{t+1}}\mathbf{Q}_t - \mathbf{Q}_t)\big\|_{\max}\Big|\mathcal{F}_t\Big]$$
$$=\mathbb{E}\Big[\big\|\mathbf{Q}_t + \alpha_t \mathbf{d}_{t+1} \odot (\mathcal{T}_{z_{t+1}}\mathbf{Q}_t - \mathbf{Q}_t) - \mathbf{Q}^*\big\|_{\max}\Big|\mathcal{F}_t\Big] \times \big\|(\bar{P}_K(\cdot,\cdot|\mathcal{F}_t) - \mathbf{d}_{t+1}) \odot (\mathcal{T}_{z_{t+1}}\mathbf{Q}_t - \mathbf{Q}_t)\big\|_{\max}$$
$$\stackrel{(i)}{\leq} \big((1-(1-\gamma_0)\alpha_t d_{t+1,\min})\|\mathbf{Q}_t - \mathbf{Q}^*\|_{\max} + \alpha_t d_{t+1,\max}\|\mathcal{T}_{z_{t+1}}\mathbf{Q}_t - \mathcal{T}_{z_t^*}\mathbf{Q}_t\|_{\max}\big)$$
$$\quad \times \frac{C}{(1-\rho)K}\|\mathcal{T}_{z_{t+1}}\mathbf{Q}_t - \mathbf{Q}_t\|_{\max} \tag{190}$$
$$\stackrel{(ii)}{\leq} \frac{c_2}{2}\big((1-(1-\gamma_0)\alpha_t d_{t+1,\min})\|\mathbf{Q}_t - \mathbf{Q}^*\|_{\max} + \alpha_t d_{t+1,\max}\|\mathcal{T}_{z_{t+1}}\mathbf{Q}_t - \mathcal{T}_{z_t^*}\mathbf{Q}_t\|_{\max}\big)^2$$
$$\quad + \frac{1}{2c_2}\left(\frac{C}{(1-\rho)K}\|\mathcal{T}_{z_{t+1}}\mathbf{Q}_t - \mathbf{Q}_t\|_{\max}\right)^2$$
$$\stackrel{(iii)}{\leq} \frac{c_2}{2}(1+c_3)(1-(1-\gamma_0)\alpha_t d_{t+1,\min})^2\|\mathbf{Q}_t - \mathbf{Q}^*\|_{\max}^2 + \frac{c_2}{2}\left(1 + \frac{1}{c_3}\right)\alpha_t^2 d_{t+1,\max}^2\|\mathcal{T}_{z_{t+1}}\mathbf{Q}_t - \mathcal{T}_{z_t^*}\mathbf{Q}_t\|_{\max}^2$$
$$\quad + \frac{1}{2c_2}\left(\frac{C}{(1-\rho)K}\|\mathcal{T}_{z_{t+1}}\mathbf{Q}_t - \mathbf{Q}_t\|_{\max}\right)^2$$

where the (i) holds by (184), (ii) holds because $ab = \sqrt{c_2}a \cdot \frac{b}{\sqrt{c_2}} < \frac{c_2}{2}a^2 + \frac{1}{2c_2}b^2$ and (iii) uses Lemma I.7. We further apply (186) and Lemma H.2 to the last line and and obtain:

$$\mathcal{L}_{1,2}$$
$$\leq \frac{c_2}{2}(1+c_3)(1-(1-\gamma_0)\alpha_t d_{t+1,\min})^2\|\mathbf{Q}_t - \mathbf{Q}^*\|_{\max}^2$$
$$\quad + \frac{c_2}{2}\left(1 + \frac{1}{c_3}\right)\alpha_t^2 d_{t+1,\max}^2\|\mathcal{T}_{z_{t+1}}\mathbf{Q}_t - \mathcal{T}_{z_t^*}\mathbf{Q}_t\|_{\max}^2$$
$$\quad + \frac{1}{2c_2}\left(\frac{C}{(1-\rho)K}\right)^2\left((1+c_4)\|\mathcal{T}_{z_{t+1}}\mathbf{Q}_t - \mathcal{T}_{z_t^*}\mathbf{Q}_t\|_{\max}^2 + \left(1 + \frac{1}{c_4}\right)(1-\gamma_0)^2\|\mathbf{Q}_t - \mathbf{Q}^*\|_{\max}^2\right)$$
$$= \left(\frac{c_2}{2}(1+c_3)(1-(1-\gamma_0)\alpha_t d_{t+1,\min})^2 + \frac{1}{2c_2}\left(\frac{C}{(1-\rho)K}\right)^2\left(1 + \frac{1}{c_4}\right)(1-\gamma_0)^2\right)\|\mathbf{Q}_t - \mathbf{Q}^*\|_{\max}^2 \tag{191}$$
$$\quad + \left(\frac{c_2}{2}\left(1 + \frac{1}{c_3}\right)\alpha_t^2 d_{t+1,\max}^2 + \frac{1+c_4}{2c_2}\left(\frac{C}{(1-\rho)K}\right)^2\right)\|\mathcal{T}_{z_{t+1}}\mathbf{Q}_t - \mathcal{T}_{z_t^*}\mathbf{Q}_t\|_{\max}^2$$
$$\leq \left(\frac{c_2}{2}(1+c_3)(1-(1-\gamma_0)\alpha_t d_{t+1,\min})^2 + \frac{1}{2c_2}\left(\frac{C}{(1-\rho)K}\right)^2\left(1 + \frac{1}{c_4}\right)(1-\gamma_0)^2\right)\|\mathbf{Q}_t - \mathbf{Q}^*\|_{\max}^2$$
$$\quad + \left(\frac{c_2}{2}\left(1 + \frac{1}{c_3}\right)\alpha_t^2 d_{t+1,\max}^2 + \frac{1+c_4}{2c_2}\left(\frac{C}{(1-\rho)K}\right)^2\right)\gamma^2 C_{\mathcal{T}}^2(1-2\beta_t\mu + \beta_t^2 L_F^2)\|z_t - z_t^*\|_2^2$$

Similarly, for term $\mathcal{L}_{1,3}$, combining (166) and (189) we obtain:

$$\mathcal{L}_{1,3} \leq \left(\frac{C}{(1-\rho)K}\|\mathcal{T}_{z_{t+1}}\mathbf{Q}_t - \mathbf{Q}_t\|_{\max}\right)^2, \tag{192}$$

which can be further transformed using (186) and Lemma H.2 to:

$$\mathcal{L}_{1,3} \leq \left(\frac{C}{(1-\rho)K}\right)^2\left((1+c_5)\|\mathcal{T}_{z_{t+1}}\mathbf{Q}_t - \mathcal{T}_{z_t^*}\mathbf{Q}_t\|_{\max}^2 + \left(1 + \frac{1}{c_5}\right)(1-\gamma_0)^2\|\mathbf{Q}_t - \mathbf{Q}^*\|_{\max}^2\right)$$
$$\leq \left(\frac{C}{(1-\rho)K}\right)^2\left((1+c_5)\gamma^2 C_{\mathcal{T}}^2(1-2\beta_t\mu + \beta_t^2 L_F^2)\|z_t - z_t^*\|_2^2 + \left(1 + \frac{1}{c_5}\right)(1-\gamma_0)^2\|\mathbf{Q}_t - \mathbf{Q}^*\|_{\max}^2\right). \tag{193}$$

Having each individual term bounded in (187), (191) and (193) respectively, we can now sum them up to get an overall bound for $\mathcal{L}_1$ using (179) as follows:

$$
\begin{aligned}
&\mathcal{L}_1 \\
&\leq \mathcal{L}_{1,1} + 2\alpha_t \mathcal{L}_{1,2} + \alpha_t^2 \mathcal{L}_{1,3} \\
&\leq (1+c_1)(1-(1-\gamma_0)\alpha_t d_{t+1,\min})^2 \|\mathbf{Q}_t - \mathbf{Q}^*\|_{\max}^2 \\
&\quad + \left(1 + \frac{1}{c_1}\right) \alpha_t^2 d_{t+1,\max}^2 \gamma^2 C_\mathcal{T}^2 (1 - 2\beta_t\mu + \beta_t^2 L_F^2)\|z_t - z_t^*\|_2^2 \\
&\quad + \alpha_t \left(c_2(1+c_3)(1-(1-\gamma_0)\alpha_t d_{t+1,\min})^2 + \frac{1}{c_2}\left(\frac{C}{(1-\rho)K}\right)^2 \left(1+\frac{1}{c_4}\right)(1-\gamma_0)^2\right)\|\mathbf{Q}_t - \mathbf{Q}^*\|_{\max}^2 \\
&\quad + \alpha_t \left(c_2\left(1+\frac{1}{c_3}\right)\alpha_t^2 d_{t+1,\max}^2 + \frac{(1+c_4)}{c_2}\left(\frac{C}{(1-\rho)K}\right)^2\right)\gamma^2 C_\mathcal{T}^2(1-2\beta_t\mu + \beta_t^2 L_F^2)\|z_t - z_t^*\|_2^2 \\
&\quad + \alpha_t^2 \left(\frac{C}{(1-\rho)K}\right)^2 (1+c_5)\gamma^2 C_\mathcal{T}^2 (1-2\beta_t\mu + \beta_t^2 L_F^2)\|z_t - z_t^*\|_2^2 \\
&\quad + \alpha_t^2 \left(\frac{C}{(1-\rho)K}\right)^2 \left(1+\frac{1}{c_5}\right)(1-\gamma_0)^2\|\mathbf{Q}_t - \mathbf{Q}^*\|_{\max}^2 \\
&= \Bigg( (1+c_1+c_2\alpha_t(1+c_3))(1-(1-\gamma_0)\alpha_t d_{t+1,\min})^2 \\
&\qquad + \left(\frac{1}{c_2}\alpha_t\left(1+\frac{1}{c_4}\right) + \alpha_t^2\left(1+\frac{1}{c_5}\right)\right)\left(\frac{C}{(1-\rho)K}\right)^2 (1-\gamma_0)^2 \Bigg)\|\mathbf{Q}_t - \mathbf{Q}^*\|_{\max}^2 \\
&\quad + \Bigg( \left(\left(1+\frac{1}{c_1}+c_2\alpha_t\left(1+\frac{1}{c_3}\right)\right)\alpha_t^2 d_{t+1,\max}^2 \\
&\qquad + (\frac{1}{c_2}\alpha_t(1+c_4) + \alpha_t^2(1+c_5))\left(\frac{C}{(1-\rho)K}\right)^2\Bigg)\gamma^2 C_\mathcal{T}^2(1-2\beta_t\mu + \beta_t^2 L_F^2)\|z_t - z_t^*\|_2^2
\end{aligned}
\tag{194}
$$

For simplicity, we take $c_1 = c\alpha_t$, $c_2 = \frac{c}{2}$ where $c = \frac{1}{2}(1-\gamma_0)d_{t+1,\min}$ and $c_3 = c_4 = c_5 = 1$, when

$K \geq \frac{2\sqrt{2}C}{(1-\rho)(1-\gamma_0)d_{t+1,\min}}$, we have:

$$\mathcal{L}_1 \leq$$

$$(1 + (c + 2c_2)\alpha_t)(1 - (1-\gamma_0)\alpha_t d_{t+1,\min})^2 \|\mathbf{Q}_t - \mathbf{Q}^*\|_{\max}^2 + 2(\frac{1}{c_2}\alpha_t + \alpha_t^2)\left(\frac{C}{(1-\rho)K}\right)^2 (1-\gamma_0)^2 \|\mathbf{Q}_t - \mathbf{Q}^*\|_{\max}^2$$

$$+ \frac{\alpha_t}{c}d_{t+1,\max}^2 \gamma^2 C_{\mathcal{T}}^2 (1 - 2\beta_t\mu + \beta_t^2 L_F^2)\|z_t - z_t^*\|_2^2 + (1 + 2c_2\alpha_t)\alpha_t^2 d_{t+1,\max}^2 \gamma^2 C_{\mathcal{T}}^2 (1 - 2\beta_t\mu + \beta_t^2 L_F^2)\|z_t - z_t^*\|_2^2$$

$$+ 2(\frac{1}{c_2}\alpha_t + \alpha_t^2)\left(\frac{C}{(1-\rho)K}\right)^2 \gamma^2 C_{\mathcal{T}}^2 (1 - 2\beta_t\mu + \beta_t^2 L_F^2)\|z_t - z_t^*\|_2^2$$

$$\leq (1 + (1-\gamma_0)\alpha_t d_{t+1,\min}\alpha_t)(1 - (1-\gamma_0)\alpha_t d_{t+1,\min})^2 \|\mathbf{Q}_t - \mathbf{Q}^*\|_{\max}^2$$

$$+ 2(\frac{4}{(1-\gamma_0)d_{t+1,\min}}\alpha_t + \alpha_t^2)\left(\frac{C}{(1-\rho)K}\right)^2 (1-\gamma_0)^2 \|\mathbf{Q}_t - \mathbf{Q}^*\|_{\max}^2$$

$$+ \frac{2\alpha_t}{(1-\gamma_0)d_{t+1,\min}}d_{t+1,\max}^2 \gamma^2 C_{\mathcal{T}}^2 (1 - 2\beta_t\mu + \beta_t^2 L_F^2)\|z_t - z_t^*\|_2^2$$

$$+ (1 + \frac{1}{2}(1-\gamma_0)d_{t+1,\min}\alpha_t)\alpha_t^2 d_{t+1,\max}^2 \gamma^2 C_{\mathcal{T}}^2 (1 - 2\beta_t\mu + \beta_t^2 L_F^2)\|z_t - z_t^*\|_2^2$$

$$+ 2(\frac{4}{(1-\gamma_0)d_{t+1,\min}}\alpha_t + \alpha_t^2)\left(\frac{C}{(1-\rho)K}\right)^2 \gamma^2 C_{\mathcal{T}}^2 (1 - 2\beta_t\mu + \beta_t^2 L_F^2)\|z_t - z_t^*\|_2^2$$

$$\leq (1 - (1-\gamma_0)\alpha_t d_{t+1,\min})\|\mathbf{Q}_t - \mathbf{Q}^*\|_{\max}^2$$

$$+ 2(\frac{4}{(1-\gamma_0)d_{t+1,\min}}\alpha_t + \alpha_t^2)\left(\frac{C}{(1-\rho)K}\right)^2 (1-\gamma_0)^2 \|\mathbf{Q}_t - \mathbf{Q}^*\|_{\max}^2$$

$$+ \frac{2\alpha_t}{(1-\gamma_0)d_{t+1,\min}}d_{t+1,\max}^2 \gamma^2 C_{\mathcal{T}}^2 (1 - 2\beta_t\mu + \beta_t^2 L_F^2)\|z_t - z_t^*\|_2^2$$

$$+ (1 + \frac{1}{2}(1-\gamma_0)d_{t+1,\min}\alpha_t)\alpha_t^2 d_{t+1,\max}^2 \gamma^2 C_{\mathcal{T}}^2 (1 - 2\beta_t\mu + \beta_t^2 L_F^2)\|z_t - z_t^*\|_2^2$$

$$+ 2(\frac{4}{(1-\gamma_0)d_{t+1,\min}}\alpha_t + \alpha_t^2)\left(\frac{C}{(1-\rho)K}\right)^2 \gamma^2 C_{\mathcal{T}}^2 (1 - 2\beta_t\mu + \beta_t^2 L_F^2)\|z_t - z_t^*\|_2^2$$

$$\leq \left(1 - \left((1-\gamma_0)d_{t+1,\min} - \frac{8}{(1-\gamma_0)d_{t+1,\min}}\left(\frac{C}{(1-\rho)K}\right)^2\right)\alpha_t\right)\|\mathbf{Q}_t - \mathbf{Q}^*\|_{\max}^2$$

$$+ 2\alpha_t^2 \left(\frac{C}{(1-\rho)K}\right)^2 (1-\gamma_0)^2 \|\mathbf{Q}_t - \mathbf{Q}^*\|_{\max}^2$$

$$+ \left(\frac{2d_{t+1,\max}^2}{(1-\gamma_0)d_{t+1,\min}} + \frac{8}{(1-\gamma_0)d_{t+1,\min}}\left(\frac{C}{(1-\rho)K}\right)^2\right)\alpha_t\gamma^2 C_{\mathcal{T}}^2 (1 - 2\beta_t\mu + \beta_t^2 L_F^2)\|z_t - z_t^*\|_2^2$$

$$+ \left((1 + \frac{1}{2}(1-\gamma_0)d_{t+1,\min}\alpha_t)d_{t+1,\max}^2 + 2\left(\frac{C}{(1-\rho)K}\right)^2\right)\alpha_t^2\gamma^2 C_{\mathcal{T}}^2 (1 - 2\beta_t\mu + \beta_t^2 L_F^2)\|z_t - z_t^*\|_2^2$$

$$(195)$$

**Bounding the additional error terms $\mathcal{L}_2$, $\mathcal{L}_3$ and $\mathcal{L}_4$.** We now focus on the additional bias terms $\mathcal{L}_2$, $\mathcal{L}_3$ and $\mathcal{L}_4$. Since each of the terms is in the order of $\alpha_t^2$, we only need to bound the scales of each term in order to obtain the convergence result. For $\mathcal{L}_2$, notice that $\mathcal{L}_2$ can be obtained by directly taking max-norm on $\mathcal{L}_2'$, we analyze $\mathcal{L}_2'$ instead. Recall the

definition of $\bar{P}_K(s, \mathbf{a}|\mathcal{F}_t)$ and (166), we have:

$$
\begin{aligned}
&\mathcal{L}_2' \\
&= \mathbb{E}\left[\left(\alpha_t \frac{1}{K} \sum_{k=0}^{K-1} \left(\mathcal{T}_{z_{t+1}} Q_{i,t}(s_k, \mathbf{a}_k) - Q_{i,t}(s_k, \mathbf{a}_k)\right) \mathbf{1}[(s, \mathbf{a}) = (s_k, \mathbf{a}_k)] - \alpha_t \mathbb{E}[\hat{\delta}_i(s, \mathbf{a})|\mathcal{F}_t]\right)^2 \middle| \mathcal{F}_t\right] \\
&= \mathbb{E}\left[\left(\alpha_t \frac{1}{K} \sum_{k=0}^{K-1} \left(\mathcal{T}_{z_{t+1}} Q_{i,t}(s, \mathbf{a}) - Q_{i,t}(s, \mathbf{a})\right) \left(\mathbf{1}[(s, \mathbf{a}) = (s_k, \mathbf{a}_k)] - \bar{P}_K(s, \mathbf{a}|\mathcal{F}_t)\right)\right)^2 \middle| \mathcal{F}_t\right] \\
&= \mathbb{E}\left[\alpha_t^2 \left(\mathcal{T}_{z_{t+1}} Q_{i,t}(s, \mathbf{a}) - Q_{i,t}(s, \mathbf{a})\right)^2 \left(\frac{1}{K} \sum_{k=0}^{K-1} \left(\mathbf{1}[(s, \mathbf{a}) = (s_k, \mathbf{a}_k)] - \bar{P}_K(s, \mathbf{a}|\mathcal{F}_t)\right)\right)^2 \middle| \mathcal{F}_t\right] \\
&= \alpha_t^2 \left(\mathcal{T}_{z_{t+1}} Q_{i,t}(s, \mathbf{a}) - Q_{i,t}(s, \mathbf{a})\right)^2 \mathbb{E}\left[\left(\frac{1}{K} \sum_{k=0}^{K-1} \left(\mathbf{1}[(s, \mathbf{a}) = (s_k, \mathbf{a}_k)] - \bar{P}_K(s, \mathbf{a}|\mathcal{F}_t)\right)\right)^2 \middle| \mathcal{F}_t\right]
\end{aligned}
\tag{196}
$$

The term inside the expectation can be bounded using geometric ergodicity: Let $Y_k = \mathbf{1}[(s_k, \mathbf{a}_k) = (s, \mathbf{a})] - \Pr((s_k, \mathbf{a}_k) = (s, \mathbf{a})|\mathcal{F}_t)$

$$
\begin{aligned}
&\mathbb{E}\left[\left(\frac{1}{K} \sum_{k=0}^{K-1} \left(\mathbf{1}[(s, \mathbf{a}) = (s_k, \mathbf{a}_k)] - \bar{P}_K(s, \mathbf{a}|\mathcal{F}_t)\right)\right)^2 \middle| \mathcal{F}_t\right] \\
&= \mathbb{E}\left[\left(\frac{1}{K} \sum_{k=0}^{K-1} Y_k\right)^2 \middle| \mathcal{F}_t\right] \\
&= \frac{1}{K^2} \left(\sum_{k=0}^{K-1} \text{Var}(Y_k) + \sum_{0 \le k_1 \ne k_2 \le K-1} \text{Cov}(Y_{k_1}, Y_{k_2})\right)
\end{aligned}
\tag{197}
$$

Since $\mathbb{E}[Y_k] = 0$ and $Y_k$ only takes two values, we have $\text{Var}(Y_k) \le \frac{1}{4}$. For $\text{Cov}(Y_{k_1}, Y_{k_2})$ we obtain from Lemma I.5 and Lemma I.6 that

$$
\text{Cov}(Y_{k_1}, Y_{k_2}) \le 8C\rho^{|k_1 - k_2|}
\tag{198}
$$

combining the two equations above, we have:

$$
\mathbb{E}\left[\left(\frac{1}{K} \sum_{k=0}^{K-1} \left(\mathbf{1}[(s, \mathbf{a}) = (s_k, \mathbf{a}_k)] - \bar{P}_K(s, \mathbf{a}|\mathcal{F}_t)\right)\right)^2 \middle| \mathcal{F}_t\right] \le \frac{1}{4K} + \frac{16C\rho}{(1-\rho)K}
\tag{199}
$$

and as a result,

$$
\mathcal{L}_2' \le \alpha_t^2 \left(\mathcal{T}_{z_{t+1}} Q_{i,t}(s, \mathbf{a}) - Q_{i,t}(s, \mathbf{a})\right)^2 \left(\frac{1}{4K} + \frac{16C\rho}{(1-\rho)K}\right).
\tag{200}
$$

Now we take max-norm and obtain the bound for $\mathcal{L}_2$:

$$
\mathcal{L}_2 \le \alpha_t^2 \left(\frac{1}{4K} + \frac{16C\rho}{(1-\rho)K}\right) \|\mathcal{T}_{z_{t+1}} \mathbf{Q}_t - \mathbf{Q}_t\|_{\max}^2.
\tag{201}
$$

Using (186) and Lemma H.2 with $c = 1$ we obtain:

$$
\begin{aligned}
\mathcal{L}_2 \le & \alpha_t^2 \left(\frac{1}{2K} + \frac{32C\rho}{(1-\rho)K}\right) \gamma^2 C_{\mathcal{T}}^2 (1 - 2\beta_t\mu + \beta_t^2 L_F^2) \|z_t - z_t^*\|_2^2 \\
& + \alpha_t^2 \left(\frac{1}{2K} + \frac{32C\rho}{(1-\rho)K}\right) (1 - \gamma_0)^2 \|\mathbf{Q}_t - \mathbf{Q}^*\|_{\max}^2
\end{aligned}
\tag{202}
$$

For $\mathcal{L}_3$, recall the definition $M_{i,t}(s,\mathbf{a}) = \frac{1}{K}\sum_{k=0}^{K-1}\xi_{i,k}(s,\mathbf{a})$ and the property that $\mathbb{E}[M_{i,t}(s,\mathbf{a})|\mathcal{F}_t] = 0$, we have:

$$
\begin{aligned}
\mathbb{E}[M_{i,t}^2(s,\mathbf{a})|\mathcal{F}_t] =& \mathbb{E}\left[\left(\frac{1}{K}\sum_{k=0}^{K-1}\xi_{i,k}(s,\mathbf{a})\right)^2\Bigg|\mathcal{F}_t\right] \\
=&\frac{1}{K^2}\mathbb{E}\left[\sum_{k=0}^{K-1}\xi_{i,k}^2(s,\mathbf{a})\Bigg|\mathcal{F}_t\right] + \frac{1}{K^2}\mathbb{E}\left[\sum_{k_1\neq k_2}^{K-1}\xi_{i,k_1}(s,\mathbf{a})\xi_{i,k_2}(s,\mathbf{a})\Bigg|\mathcal{F}_t\right] \\
=&\frac{1}{K^2}\sum_{k=0}^{K-1}\mathbb{E}\left[\xi_{i,k}^2(s,\mathbf{a})\big|\mathcal{F}_t\right] + \frac{1}{K^2}\sum_{k_1\neq k_2}^{K-1}\mathbb{E}\left[\xi_{i,k_1}(s,\mathbf{a})\xi_{i,k_2}(s,\mathbf{a})|\mathcal{F}_t\right] \\
=&\frac{1}{K^2}\sum_{k=0}^{K-1}\mathbb{E}\left[\xi_{i,k}^2(s,\mathbf{a})\big|\mathcal{F}_t\right] + \frac{2}{K^2}\sum_{k_1<k_2}^{K-1}\mathbb{E}\left[\mathbb{E}[\xi_{i,k_1}(s,\mathbf{a})\xi_{i,k_2}(s,\mathbf{a})|\mathcal{F}_{t,k_1}]\big|\mathcal{F}_t\right] \\
=&\frac{1}{K^2}\sum_{k=0}^{K-1}\mathbb{E}\left[\xi_{i,k}^2(s,\mathbf{a})\big|\mathcal{F}_t\right] + \frac{2}{K^2}\sum_{k_1<k_2}^{K-1}\mathbb{E}\left[\mathbb{E}[\xi_{i,k_2}(s,\mathbf{a})|\mathcal{F}_{t,k_1}]\xi_{i,k_1}(s,\mathbf{a})\big|\mathcal{F}_t\right] \\
=&\frac{1}{K^2}\sum_{k=0}^{K-1}\mathbb{E}\left[\xi_{i,k}^2(s,\mathbf{a})\big|\mathcal{F}_t\right] + \frac{2}{K^2}\sum_{k_1<k_2}^{K-1}\mathbb{E}\left[\mathbb{E}[\mathbb{E}[\xi_{i,k_2}(s,\mathbf{a})|\mathcal{F}_{t,k_2}]|\mathcal{F}_{t,k_1}]\xi_{i,k_1}(s,\mathbf{a})\big|\mathcal{F}_t\right] \\
=&\frac{1}{K^2}\sum_{k=0}^{K-1}\mathbb{E}\left[\xi_{i,k}^2(s,\mathbf{a})\big|\mathcal{F}_t\right]
\end{aligned}
\tag{203}
$$

and as a result,

$$
\mathbb{E}\left[\left(\alpha_t\frac{1}{K}\sum_{k=0}^{K-1}\xi_{i,k}(s,\mathbf{a})\right)^2\Bigg|\mathcal{F}_t\right] \leq \frac{\alpha_t^2}{K^2}\sum_{k=0}^{K-1}\mathbb{E}[\xi_{i,k}^2(s,\mathbf{a})|\mathcal{F}_t] \leq \frac{\alpha_t^2}{K}\max_k\|\xi_{i,k}\|_{\max}^2
\tag{204}
$$

and since this bound hold for all $(s,\mathbf{a})$ pairs, we have:

$$
\mathcal{L}_3 \leq \frac{\alpha_t^2}{K}\max_k\|\xi_{i,k}\|_{\max}^2 \leq \frac{4\alpha_t^2}{K}\mathrm{sp}(\mathbf{Q})^2.
\tag{205}
$$

For $\mathcal{L}_4$, by using the fact that $ab \leq \frac{1}{2}a^2 + \frac{1}{2}b^2$, we obtain:

$$
\begin{aligned}
2\mathcal{L}_4 =& 2\mathbb{E}\left[\left\|\alpha_t\frac{1}{K}\sum_{k=0}^{K-1}\left(\mathcal{T}_{z_{t+1}}\mathbf{Q}_t(s_k,\mathbf{a}_k)-\mathbf{Q}_t(s_k,\mathbf{a}_k)\right)\otimes e_{(s_k,\mathbf{a}_k)}-\alpha_t\mathbb{E}[\hat{\boldsymbol{\delta}}|\mathcal{F}_t]\right\|_{\max}\times\left\|\alpha_t\frac{1}{K}\sum_{k=0}^{K-1}\boldsymbol{\xi}_k\right\|_{\max}\Bigg|\mathcal{F}_t\right] \\
\leq& \mathbb{E}\left[\left\|\alpha_t\frac{1}{K}\sum_{k=0}^{K-1}\left(\mathcal{T}_{z_{t+1}}\mathbf{Q}_t(s_k,\mathbf{a}_k)-\mathbf{Q}_t(s_k,\mathbf{a}_k)\right)\otimes e_{(s_k,\mathbf{a}_k)}-\alpha_t\mathbb{E}[\hat{\boldsymbol{\delta}}|\mathcal{F}_t]\right\|_{\max}^2\Bigg|\mathcal{F}_t\right] \\
&+ \mathbb{E}\left[\left\|\alpha_t\frac{1}{K}\sum_{k=0}^{K-1}\boldsymbol{\xi}_k\right\|_{\max}^2\Bigg|\mathcal{F}_t\right] \\
=& \mathcal{L}_2 + \mathcal{L}_3 \\
\leq& \alpha_t^2\left(\frac{1}{2K}+\frac{32C\rho}{(1-\rho)K}\right)\gamma^2C_{\mathcal{T}}^2(1-2\beta_t\mu+\beta_t^2 L_F^2)\|z_t-z_t^*\|_2^2 \\
&+ \alpha_t^2\left(\frac{1}{2K}+\frac{32C\rho}{(1-\rho)K}\right)(1-\gamma_0)^2\|\mathbf{Q}_t-\mathbf{Q}^*\|_{\max}^2 + \frac{4\alpha_t^2}{K}\mathrm{sp}(\mathbf{Q})^2
\end{aligned}
\tag{206}
$$

**Obtaining the Lyapunov Drift Inequality for $Q$.** Having obtained the upper bounds for $\mathcal{L}_1, \mathcal{L}_2, \mathcal{L}_3$ and $\mathcal{L}_4$, we finally combine the error terms to obtain the Lyapunov drift inequality for $Q$ functions. Recall (171) and substituting in (195),

(202), (205), and (206), we have:

$$
\begin{aligned}
&\mathbb{E}\left[\|\mathbf{Q}_{t+1} - \mathbf{Q}^*\|_{\max}^2 \big| \mathcal{F}_t\right] \\
&\leq \mathcal{L}_1 + \mathcal{L}_2 + \mathcal{L}_3 + 2\mathcal{L}_4 \\
&= \left(1 - \left((1-\gamma_0)d_{t+1,\min} - \frac{8}{(1-\gamma_0)d_{t+1,\min}}\left(\frac{C}{(1-\rho)K}\right)^2\right)\alpha_t\right)\|\mathbf{Q}_t - \mathbf{Q}^*\|_{\max}^2 \\
&\quad + \left(\frac{1}{K} + \frac{64C\rho}{(1-\rho)K} + 2\left(\frac{C}{(1-\rho)K}\right)^2\right)\alpha_t^2(1-\gamma_0)^2\|\mathbf{Q}_t - \mathbf{Q}^*\|_{\max}^2 \\
&\quad + \left(\frac{2d_{t+1,\max}}{(1-\gamma_0)d_{t+1,\min}^2} + \frac{8}{(1-\gamma_0)d_{t+1,\min}}\left(\frac{C}{(1-\rho)K}\right)^2\right)\alpha_t\gamma^2 C_{\mathcal{T}}^2(1 - 2\beta_t\mu + \beta_t^2 L_F^2)\|z_t - z_t^*\|_2^2 \\
&\quad + \left((1 + \tfrac{1}{2}(1-\gamma_0)d_{t+1,\min}\,\alpha_t)d_{t+1,\max}^2 + \frac{1}{K} + \frac{64C\rho}{(1-\rho)K} + 2\left(\frac{C}{(1-\rho)K}\right)^2\right) \\
&\quad \times \alpha_t^2\gamma^2 C_{\mathcal{T}}^2(1 - 2\beta_t\mu + \beta_t^2 L_F^2)\|z_t - z_t^*\|_2^2 + \frac{8\alpha_t^2}{K}\mathrm{sp}(\mathbf{Q})^2
\end{aligned}
\tag{207}
$$

## H.2. Lyapunov Drift Inequality for Policies

We now shift our focus on the Lyapunov drift inequality for policies. Notice that given the $Q$ function $\mathbf{Q}_t$ at time step $t$, the policy update is exactly the same as the non-stochastic update rule (17) because:

$$
\Lambda F(z; -\mathbf{Q}_t(s,\cdot)) = \begin{bmatrix}
\lambda_1(Q_1(s,\cdot)p_1(\cdot|s) + \epsilon_1\nabla\nu_1(\pi_1;s)) \\
\lambda_2(Q_2(s,\cdot)p_2(\cdot|s) + \epsilon_2\nabla\nu_2(\pi_2;s)) \\
\lambda_1(-Q_1^T(s,\cdot)\pi_1(\cdot|s) + \frac{1}{\tau_1}\nabla_{p_1}D_1(p_1,\pi_2;s)) \\
\lambda_2(-Q_2^T(s,\cdot)\pi_2(\cdot|s) + \frac{1}{\tau_2}\nabla_{p_2}D_2(p_2,\pi_1;s))
\end{bmatrix}
\tag{208}
$$

this provides the same iteration dynamic as (132):

$$
\|z_{t+1} - z_{t+1}^*\|_2 \leq (1 + \alpha_t L_{RQE}C_{\mathcal{T}})\sqrt{1 - 2\beta_t\mu + \beta_t^2 L_F^2}\|z_t - z_t^*\|_2 + (1+\gamma_0)\alpha_t L_{RQE}\|\mathbf{Q}_t - \mathbf{Q}^*\|_{\max}.
\tag{209}
$$

However, in order to fit the squared term in the drift inequality for $Q$ functions, we apply Lemma I.7 to the above inequality and obtain:

$$
\begin{aligned}
\|z_{t+1} - z_{t+1}^*\|_2^2 \leq{}& (1 + c_z)(1 + \alpha_t L_{RQE}C_{\mathcal{T}})^2\left(1 - 2\beta_t\mu + \beta_t^2 L_F^2\right)\|z_t - z_t^*\|_2^2 \\
&+ \left(1 + \frac{1}{c_z}\right)(1+\gamma_0)^2\alpha_t^2 L_{RQE}^2\|\mathbf{Q}_t - \mathbf{Q}^*\|_{\max}^2.
\end{aligned}
\tag{210}
$$

Taking $c_z = c_6\alpha_t L_{RQE}$, we obtain:

$$
\begin{aligned}
\|z_{t+1} - z_{t+1}^*\|_2^2 \leq{}& (1 + c_6\alpha_t L_{RQE})(1 + \alpha_t L_{RQE}C_{\mathcal{T}})^2\left(1 - 2\beta_t\mu + \beta_t^2 L_F^2\right)\|z_t - z_t^*\|_2^2 \\
&+ \left(\alpha_t^2 L_{RQE}^2 + \frac{\alpha_t L_{RQE}}{c_6}\right)(1+\gamma_0)^2\|\mathbf{Q}_t - \mathbf{Q}^*\|_{\max}^2.
\end{aligned}
\tag{211}
$$

## H.3. Solving the Coupled Lyapunov Inequalities

Having obtained the Lyapunov drift inequality (207) for $Q$ and (211) for $z$, we now proceed to solve the coupled Lyapunov drift inequalities. We first simplify (207) as follows: For $Q$ functions, assume the iterates $\mathbf{Q}_t$ are bounded, the coefficient with term $\alpha_t^2$ is bounded, and we can write (207) as:

$$
\mathbb{E}\left[\|\mathbf{Q}_{t+1} - \mathbf{Q}^*\|_{\max}^2 \big| \mathcal{F}_t\right] \leq (1 - C_1\alpha_t)\|\mathbf{Q}_t - \mathbf{Q}^*\|_{\max}^2 + C_2\alpha_t(1 - \tilde{\beta}_t)^2\|z_t - z_t^*\|_2^2 + C_3\alpha_t^2
\tag{212}
$$

where

$$C_1 = (1-\gamma_0)d_{t+1,\min} - \frac{8}{(1-\gamma_0)d_{t+1,\min}}\left(\frac{C}{(1-\rho)K}\right)^2; \tag{213}$$

$$C_2 = \gamma^2 C_{\mathcal{T}}^2 \left(\frac{2d_{t+1,\max}}{(1-\gamma_0)d_{t+1,\min}^2} + \frac{8}{(1-\gamma_0)d_{t+1,\min}}\left(\frac{C}{(1-\rho)K}\right)^2\right); \tag{214}$$

$$C_3 = \max_t (1-\gamma_0)^2 \left(\frac{1}{K} + \frac{64C\rho}{(1-\rho)K} + 2\left(\frac{C}{(1-\rho)K}\right)^2\right)\|\mathbf{Q}_t - \mathbf{Q}^*\|_{\max}^2 \tag{215}$$

$$+ \left((1 + \frac{1}{2}(1-\gamma_0)d_{t+1,\min}\alpha_t)d_{t+1,\max}^2 + \frac{1}{K} + \frac{64C\rho}{(1-\rho)K} + 2\left(\frac{C}{(1-\rho)K}\right)^2\right)$$

$$\times \gamma^2 C_{\mathcal{T}}^2(1-\tilde{\beta}_t)^2\|z_t - z_t^*\|_2^2 + \frac{8}{K}\mathrm{sp}(\mathbf{Q})^2.$$

Similarly, we can rewrite (211) as:

$$\|z_{t+1} - z_{t+1}^*\|_2^2 \leq (1 + C_4\alpha_t)(1-\tilde{\beta}_t)^2\|z_t - z_t^*\|_2^2 + C_5\alpha_t\|\mathbf{Q}_t - \mathbf{Q}\|_{\max}^2 + C_6\alpha_t^2 \tag{216}$$

where (for simplicity we take $c_6 = 1$):

$$C_4 = (2C_{\mathcal{T}} + 1)L_{RQE}; \tag{217}$$

$$C_5 = L_{RQE}(1+\gamma_0)^2; \tag{218}$$

$$C_6 = \max_t (2 + C_{\mathcal{T}})L_{RQE}^2 C_{\mathcal{T}}(1-\tilde{\beta}_t)^2 + L_{RQE}^3 C_{\mathcal{T}}^2\alpha_t(1-\tilde{\beta}_t)^2 \tag{219}$$

$$+ L_{RQE}^2(1+\gamma_0)^2\|\mathbf{Q}_t - \mathbf{Q}^*\|_{\max}^2.$$

Let $U_t = \mathbb{E}[\|\mathbf{Q}_t - \mathbf{Q}^*\|_{\max}^2]$ and $V_t = \mathbb{E}[\|z_t - z_t^*\|_2^2]$, by taking expectations on both sides of (212) and (216), we obtain:

$$\begin{aligned}
U_{t+1} + kV_{t+1} &\leq (1 - C_1\alpha_t)U_t + C_2\alpha_t(1-\tilde{\beta}_t)^2 V_t + C_3\alpha_t^2 \\
&+ k(1 + C_4\alpha_t)(1-\tilde{\beta}_t)^2 V_t + kC_5\alpha_t U_t + kC_6\alpha_t^2 \\
&= (1 - (C_1 - kC_5)\alpha_t)U_t + (k + (C_2 + kC_4)\alpha_t)(1-\tilde{\beta}_t)^2 V_t + (C_3 + kC_6)\alpha_t^2 \\
&\leq \max\{(1 - (C_1 - kC_5)\alpha_t), (1 + (C_4 + C_2/k)\alpha_t)(1-\tilde{\beta}_t)^2\}(U_t + kV_t) + (C_3 + kC_6)\alpha_t^2,
\end{aligned} \tag{220}$$

We can now set conditions of $K$ and $k$ to simplify the above expression. When $K$ satisfies:

$$K \geq \frac{4\sqrt{2}C}{(1-\gamma_0)(1-\rho)d_{t+1,\min}}, \tag{221}$$

we have:

$$\frac{8}{(1-\gamma_0)d_{t+1,\min}}\left(\frac{C}{(1-\rho)K}\right)^2 \leq \frac{(1-\gamma_0)d_{t+1,\min}}{4}, \tag{222}$$

and when

$$k = \frac{(1-\gamma_0)d_{t+1,\min}}{4L_{RQE}(1+\gamma_0)^2}. \tag{223}$$

we further have:

$$\begin{aligned}
C_1 - kC_5 &\geq (1-\gamma_0)d_{t+1,\min} - \frac{1}{4}(1-\gamma_0)d_{t+1,\min} - \frac{1}{4}(1-\gamma_0)d_{t+1,\min} \\
&= \frac{1}{2}(1-\gamma_0)d_{t+1,\min}
\end{aligned} \tag{224}$$

so that

$$1 - (C_1 - kC_5)\alpha_t \leq 1 - \frac{1}{2}(1-\gamma_0)d_{t+1,\min}\alpha_t. \tag{225}$$

Also, we have:

$$
\begin{aligned}
C_4 + \frac{C_2}{k} \leq& (2C_{\mathcal{T}}+1)L_{RQE} + \gamma^2 C_{\mathcal{T}}^2 \left( \frac{2d_{t+1,\max}}{(1-\gamma_0)d_{t+1,\min}^2} + \frac{8}{(1-\gamma_0)d_{t+1,\min}} \left( \frac{C}{(1-\rho)K} \right)^2 \right) \frac{4L_{RQE}(1+\gamma_0)^2}{(1-\gamma_0)d_{t+1,\min}} \\
\leq& (2C_{\mathcal{T}}+1)L_{RQE} + \gamma^2 C_{\mathcal{T}}^2 \left( \frac{2d_{t+1,\max}}{(1-\gamma_0)d_{t+1,\min}^2} + \frac{(1-\gamma_0)d_{t+1,\min}}{4} \right) \frac{4L_{RQE}(1+\gamma_0)^2}{(1-\gamma_0)d_{t+1,\min}} \\
=& \left(2C_{\mathcal{T}}+1+(1+\gamma_0)^2\right) L_{RQE} + 8L_{RQE}\gamma^2 C_{\mathcal{T}}^2 \left( \frac{1+\gamma_0}{1-\gamma_0} \right)^2 \frac{d_{t+1,\max}}{d_{t+1,\min}^3} \\
=& \left( 1 + (1+\gamma_0)^2 + 2C_{\mathcal{T}} + 8\gamma^2 C_{\mathcal{T}}^2 \left( \frac{1+\gamma_0}{1-\gamma_0} \right)^2 \frac{d_{t+1,\max}}{d_{t+1,\min}^3} \right) L_{RQE}
\end{aligned}
\tag{226}
$$

recall that when $\beta_t \leq \min\{\frac{1}{\mu}, \frac{\mu}{L_F^2}\}$, we have $(1-\tilde{\beta}_t)^2 = 1 - 2\beta_t\mu + \beta_t^2 L_F^2 \leq 1 - \beta_t\mu$, and when

$$
(C_1 - kC_5 + C_4 + \frac{C_2}{k})\alpha_t \leq \mu\beta_t + (C_4 + \frac{C_2}{k})\alpha_t\beta_t
\tag{227}
$$

we would have $1 - (C_1 - kC_5)\alpha_t \geq (1 + (C_4 + C_2/k)\alpha_t)(1-\tilde{\beta}_t)^2$. Let $\alpha_t < 1$ and combining the conditions above, and then (227) simplifies to:

$$
\begin{aligned}
\frac{\beta_t}{\alpha_t} \geq& \frac{C_1 - kC_5 + C_4 + C_2/k}{\mu + C_4 + C_2/k} \\
=& \frac{(1-\gamma_0)d_{t+1,\min}/2 + \left( 1 + (1+\gamma_0)^2 + 2C_{\mathcal{T}} + 8\gamma^2 C_{\mathcal{T}}^2 \left( \frac{1+\gamma_0}{1-\gamma_0} \right)^2 \frac{d_{t+1,\max}}{d_{t+1,\min}^3} \right) L_{RQE}}{\mu + \left( 1 + (1+\gamma_0)^2 + 2C_{\mathcal{T}} + 8\gamma^2 C_{\mathcal{T}}^2 \left( \frac{1+\gamma_0}{1-\gamma_0} \right)^2 \frac{d_{t+1,\max}}{d_{t+1,\min}^3} \right) L_{RQE}}.
\end{aligned}
\tag{228}
$$

When (221), (223) and (228) are satisfied, our recursion becomes:

$$
U_{t+1} + kV_{t+1} \leq (1 - \frac{1}{2}(1-\gamma_0)d_{t+1,\min}\alpha_t)(U_T + kV_t) + (C_3 + kC_6)\alpha_t^2
\tag{229}
$$

To clean up the recursion, notice that we naturally have $d_{t+1,\max} \leq 1$, and by Assumption 4.5 and Assumption 4.6, we have

$$
d_{t+1,\min} \geq d_{\min} := \underline{\mu}\underline{\pi}
\tag{230}
$$

and by a similar argument to Lemma G.3, we have $\|\mathbf{Q}_t\|_{\max} \leq Q_{\max}, \operatorname{sp}(\mathbf{Q}_t) \leq Q_{span}$ and $\|\mathbf{Q}^*\|_{\max} \leq Q_{\max}$, and therefore, $C_3$ can be rewritten as:

$$
\begin{aligned}
C_3 =& 4(1-\gamma_0)^2 \left( \frac{1}{K} + \frac{64C\rho}{(1-\rho)K} + 2 \left( \frac{C}{(1-\rho)K} \right)^2 \right) Q_{\max}^2 \\
& + \left( 1 + \frac{1}{2}(1-\gamma_0)d_{\min}\alpha_t + \frac{1}{K} + \frac{64C\rho}{(1-\rho)K} + 2 \left( \frac{C}{(1-\rho)K} \right)^2 \right) \\
& \times \gamma^2 C_{\mathcal{T}}^2 (1-\tilde{\beta}_t)^2 \|z_t - z_t^*\|_2^2 + \frac{8}{K} Q_{span}^2,
\end{aligned}
$$

and $C_6$ can be written as:

$$
C_6 = (2 + C_{\mathcal{T}})L_{RQE}^2 C_{\mathcal{T}}(1-\beta)^2 + L_{RQE}^3 C_{\mathcal{T}}^2 \alpha_t(1-\beta)^2 + 4L_{RQE}^2(1+\gamma_0)^2 Q_{\max}.
\tag{231}
$$

Additionally, Equation (228) further simplifies to:

$$
\frac{\beta_t}{\alpha_t} \geq \frac{(1-\gamma_0)d_{\min}/2 + \left( 1 + (1+\gamma_0)^2 + 2C_{\mathcal{T}} + \frac{8\gamma^2 C_{\mathcal{T}}^2}{d_{\min}^3} \left( \frac{1+\gamma_0}{1-\gamma_0} \right)^2 \right) L_{RQE}}{\mu + \left( 1 + (1+\gamma_0)^2 + 2C_{\mathcal{T}} + \frac{8\gamma^2 C_{\mathcal{T}}^2}{d_{\min}^3} \left( \frac{1+\gamma_0}{1-\gamma_0} \right)^2 \right) L_{RQE}}.
\tag{232}
$$

Now by applying Lemma I.8, we obtain:

*Scenario 1:* Assuming we are using constant step sizes $\alpha_t = \alpha$, $\beta_t = \beta$, we have:

$$U_t + kV_t \leq (1 - \frac{1}{2}(1 - \gamma_0)d_{\min}\alpha)^t(U_0 + kV_0) + \frac{2(C_3 + kC_6)}{(1 - \gamma_0)d_{\min}}\alpha \tag{233}$$

and therefore,

$$\mathbb{E}[\|\mathbf{Q}_t - \mathbf{Q}^*\|_{\max}^2] \leq (1 - \frac{1}{2}(1 - \gamma_0)d_{\min}\alpha)^t(Q_{\max}^2 + 8k) + \frac{2(C_3 + kC_6)}{(1 - \gamma_0)d_{\min}}\alpha \tag{234}$$

and

$$\mathbb{E}[\|z_t - z_t^*\|_2^2] \leq (1 - \frac{1}{2}(1 - \gamma_0)d_{\min}\alpha)^t(Q_{\max}^2/k + 8) + \frac{2(C_3/k + C_6)}{(1 - \gamma_0)d_{\min}}\alpha \tag{235}$$

and since

$$
\begin{aligned}
\|z_t - z^*\|_2^2 &\leq (\|z_t - z_t^*\|_2 + \|z_t^* - z^*\|_2)^2 \\
&\leq (\|z_t - z_t^*\|_2 + L_{RQE}\|\mathbf{Q}_t - \mathbf{Q}^*\|_{\max})^2 \\
&\leq 2\|z_t - z_t^*\|_2^2 + 2L_{RQE}^2\|\mathbf{Q}_t - \mathbf{Q}^*\|_{\max}^2
\end{aligned} \tag{236}
$$

we have:

$$
\begin{aligned}
\mathbb{E}[\|z_t - z^*\|_2^2] &\leq 2\mathbb{E}[\|z_t - z_t^*\|_2^2] + 2L_{RQE}^2\mathbb{E}[\|\mathbf{Q}_t - \mathbf{Q}^*\|_{\max}^2] \\
&\leq (2 + 2kL_{RQE}^2)\left((1 - \frac{1}{2}(1 - \gamma_0)d_{\min}\alpha)^t(Q_{\max}^2/k + 8) + \frac{2(C_3/k + C_6)}{(1 - \gamma_0)d_{\min}}\alpha\right)
\end{aligned} \tag{237}
$$

*Scenario 2:* Assuming we are using diminishing step sizes $\alpha_t = \frac{\alpha}{t+h}$, $\beta_t = \frac{\beta}{t+h}$, we have:

$$U_t + kV_t \leq \left(U_0 + kV_0 + \frac{(C_3 + kC_6)\alpha^2}{h^2}\frac{h + 2 - \frac{1}{2}(1 - \gamma_0)d_{\min}\alpha}{1 - \frac{1}{2}(1 - \gamma_0)d_{\min}\alpha}\right)\left(\frac{h+1}{h+t}\right)^{\frac{1}{2}(1-\gamma_0)d_{\min}\alpha}, \tag{238}$$

as a result,

$$\mathbb{E}[\|\mathbf{Q}_t - \mathbf{Q}^*\|_{\max}^2] \leq \left(Q_{\max}^2 + 8k + \frac{(C_3 + kC_6)\alpha^2}{h^2}\frac{h + 2 - \frac{1}{2}(1 - \gamma_0)d_{\min}\alpha}{1 - \frac{1}{2}(1 - \gamma_0)d_{\min}\alpha}\right)\left(\frac{h+1}{h+t}\right)^{\frac{1}{2}(1-\gamma_0)d_{\min}\alpha}, \tag{239}$$

and

$$\mathbb{E}[\|z_t - z^*\|_2^2] \leq (2/k + 2L_{RQE}^2)\left(Q_{\max}^2 + 8k + \frac{(C_3 + kC_6)\alpha^2}{h^2}\frac{h + 2 - \frac{1}{2}(1 - \gamma_0)d_{\min}\alpha}{1 - \frac{1}{2}(1 - \gamma_0)d_{\min}\alpha}\right)\left(\frac{h+1}{h+t}\right)^{\frac{1}{2}(1-\gamma_0)d_{\min}\alpha} \tag{240}$$

## I. Auxiliary Lemmas

In this section we state and prove some auxiliary lemmas that are used in our proofs.

**Lemma I.1.** *Let* $\Lambda = \text{Diag}(\lambda)$*, a differentiable mapping* $F : \mathcal{Z} \to \mathbb{R}^N$ *is* $\lambda$*-strictly monotone if for all* $z \in \mathcal{Z}$*,*

$$\Lambda \nabla F(z) + \nabla F(z)^T \Lambda \succ 0. \tag{241}$$

*Further, it is* $(\mu, \lambda)$*-strongly monotone if and only if for all* $z \in \mathcal{Z}$*,*

$$\Lambda \nabla F(z) + \nabla F(z)^T \Lambda \succeq 2\mu I. \tag{242}$$

*Here* $\nabla F(z)$ *denotes the Jacobian of* $F(z)$*.*

*Proof.* We provide a proof for general $\Lambda$. For $z, z' \in \mathcal{Z}$ and $v = z' - z$, let

$$S(z) = \frac{\Lambda \nabla F(z) + \nabla F(z)^T \Lambda}{2}, \tag{243}$$

define a function $\varphi(t) := z + tv, t \in [0, 1]$, we have that $\varphi(0) = z$ and $\varphi(1) = z'$. Consider the function

$$g(t) := F(\varphi(t))^T \Lambda v, \tag{244}$$

we have that:

$$\begin{aligned}
&\langle F(z') - F(z), z' - z \rangle_\Lambda \\
=&\langle F(z') - F(z), v \rangle_\Lambda \\
=&(F(z') - F(z))^T \Lambda v \\
=&g(1) - g(0).
\end{aligned} \tag{245}$$

For $g(1) - g(0)$, we can write is as the following integral:

$$\begin{aligned}
g(1) - g(0) &= \int_0^1 g'(t)\, \mathrm{d}t \\
&= \int_0^1 \frac{\mathrm{d}}{\mathrm{d}t}\left(F(\varphi(t))^T \Lambda v\right) \mathrm{d}t \\
&= \int_0^1 v^T \nabla F(\varphi(t))^T \Lambda v\, \mathrm{d}t.
\end{aligned} \tag{246}$$

Also, a symmetric expansion of $\frac{\mathrm{d}}{\mathrm{d}t}\left(F(\varphi(t))^T \Lambda v\right)$ yields:

$$g(1) - g(0) = \int_0^1 v^T \Lambda \nabla F(\varphi(t)) v\, \mathrm{d}t. \tag{247}$$

Therefore, we have that:

$$g(1) - g(0) = \int_0^1 v^T S(\varphi(t)) v\, \mathrm{d}t. \tag{248}$$

**From Jacobian condition to monotonicity.** If $S(z) \succ 0$, we have that $v^T S(\varphi(t)) v > 0$ for all $t$, and therefore, $\langle F(z') - F(z), z' - z \rangle_\Lambda = g(1) - g(0) > 0$, $\lambda$-strict monotonicity holds.

If we further have

$$S(z) \succeq \mu I, \tag{249}$$

it holds that

$$\begin{aligned}
&\langle F(z') - F(z), z' - z \rangle_\Lambda \\
=&g(1) - g(0) \\
=&\int_0^1 v^T S(\varphi(t)) v\, \mathrm{d}t \\
\geq&\mu \int_0^1 v^T v\, \mathrm{d}t \\
=&\mu \|v\|_2^2 \\
=&\mu \|z' - z\|_2^2,
\end{aligned} \tag{250}$$

indicating that $(\mu, \lambda)$-strong monotonicity holds.

**From strong monotonicity to Jacobian Condition.** If $F$ is $(\mu, \lambda)$-strongly monotone, for all $t_1 < t_2, t_1, t_2 \in [0, 1]$, we have that

$$(t_2 - t_1)(g(t_2) - g(t_1)) = \langle F(z + t_2 v) - F(z + t_1 v), t_2 v - t_1 v \rangle_\Lambda \geq \mu(t_2 - t_1)^2 \|v\|_2^2 \tag{251}$$

and therefore,

$$g(t_2) - g(t_1) \geq \mu(t_2 - t_1)\|v\|_2^2. \tag{252}$$

Since $F$ is differentiable, we have that $g'(t) \geq \mu\|v\|_2^2$. Now we expand $g'(t)$ to obtain:

$$g'(t) = \frac{\mathrm{d}}{\mathrm{d}t}\left(F(\varphi(t))^T \Lambda v\right) = v^T \nabla F(\varphi(t))^T \Lambda v \tag{253}$$

and similarly using symmetry,

$$g'(t) = v^T \Lambda \nabla F(\varphi(t)) v \tag{254}$$

we have that:

$$v^T\left(\Lambda \nabla F(\varphi(t)) + \nabla F(\varphi(t))^T \Lambda\right) v \geq 2\mu v^T v, \tag{255}$$

that gives:

$$\Lambda \nabla F(\varphi(t)) + \nabla F(\varphi(t))^T \Lambda \succeq 2\mu I. \tag{256}$$

This completes the proof. $\qquad\square$

**Lemma I.2.** *Consider a normal form game with bounded payoff matrices $\mathbf{R}$, under the following cases the RQE of the game is bounded away from zero:*

1. *$D_i(\cdot, \cdot)$ are KL-divergence and $\nu_i(\cdot)$ are log-barrier functions, then $\pi_i^* \geq \frac{\epsilon_i}{\epsilon_i |\mathcal{A}_i| + \mathrm{sp}(\mathbf{R})}$;*

2. *$D_i(\cdot, \cdot)$ are reverse KL-divergence and $\nu_i(\cdot)$ are negative entropy, then $\pi_i^* \geq \frac{1}{|\mathcal{A}_i|} \exp\left(-\mathrm{sp}(\mathbf{R})/\epsilon_i\right)$ and $p_i \geq \frac{\exp(-\mathrm{sp}(\mathbf{R})/\epsilon_i)}{|\mathcal{A}_i|(|\mathcal{A}_{-i}| + \tau_i \mathrm{sp}(\mathbf{R})/\epsilon_i)}$.*

*Proof.* Consider the gradient operator of the game:

$$F(z; \mathbf{R}) = \begin{bmatrix} -R_1 p_1 + \epsilon_1 \nabla \nu_1(\pi_1) \\ -R_2 p_2 + \epsilon_2 \nabla \nu_2(\pi_2) \\ R_1^T \pi_1 + \frac{1}{\tau_1} \nabla_{p_1} D_1(p_1, \pi_2) \\ R_2^T \pi_2 + \frac{1}{\tau_2} \nabla_{p_2} D_2(p_2, \pi_1) \end{bmatrix} \tag{257}$$

*Case 1:* When $D_i$ are KL-divergence and $\nu_i$ are log-barrier functions, their gradients can be written as:

$$\nabla \nu_i(\pi_i) = -(\pi_i)^{-1}; \nabla_{p_i} D_i(p_i, \pi_{-i}) = \log\frac{p_i}{\pi_{-i}} + 1. \tag{258}$$

Since $\nu_1(\pi_1)$ and $\nu_2(\pi_2)$ goes to infinity on the boundary of the simplex, the RQE $\pi^*$ must satisfy the KKT condition for its optimality:

$$R_i p_i^* - \mu 1 = \epsilon_i \nabla \nu_i(\pi_i^*) = -\frac{\epsilon_i}{\pi_i^*}. \tag{259}$$

which implies that $\pi_i^*$ has the form:

$$\pi_i^*(a) = \frac{\epsilon_i}{\mu - R_i p_i^*(a)} \tag{260}$$

here we overload the notation $\mu$ to denote Lagrange multiplier such that:

$$\sum_{a \in \mathcal{A}_i} \frac{\epsilon_i}{\mu - R_i p_i^*(a)} = 1. \tag{261}$$

Let $a_{\max} = \arg\max_{a \in \mathcal{A}_i} R_i p_i^*(a)$, since the probability of choosing $a_{\max}$ is at least $\frac{1}{|\mathcal{A}_i|}$, we have:

$$\frac{\epsilon_i}{\mu - R_i p_i^*(a_{\max})} \geq \frac{1}{|\mathcal{A}_i|} \tag{262}$$

which implies:

$$\mu \leq R_i p_i^*(a_{\max}) + \epsilon_i |\mathcal{A}_i| \tag{263}$$

This leads to the conclusion that:

$$\pi_i^*(a) \geq \frac{\epsilon_i}{\epsilon_i |\mathcal{A}_i| + R_i p_i^*(a_{\max}) - R_i p_i^*(a)} \geq \frac{\epsilon_i}{\epsilon_i |\mathcal{A}_i| + \mathrm{sp}(\mathbf{R})}. \tag{264}$$

*Case 2:* When $D_i$ are reverse KL-divergence and $\nu_i$ are negative entropy, their gradients can be written as:

$$\nabla \nu_i(\pi_i) = \log \pi_i + 1; \nabla_{p_i} D_i(p_i, \pi_{-i}) = -\frac{\pi_{-i}}{p_i}. \tag{265}$$

Similarly, we have that

$$\pi_i^*(a) = \frac{\exp\left(\frac{R_i p_i^*(a)}{\epsilon_i}\right)}{\sum_{a'} \exp\left(\frac{R_i p_i^*(a)}{\epsilon_i}\right)} \geq \frac{1}{|\mathcal{A}_i|} \exp\left(-\mathrm{sp}(\mathbf{R})/\epsilon_i\right) \tag{266}$$

and additionally we have:

$$p_i \geq \frac{\exp\left(-\mathrm{sp}(\mathbf{R})/\epsilon_i\right)}{|\mathcal{A}_i|(|\mathcal{A}_{-i}| + \tau_i \mathrm{sp}(\mathbf{R})/\epsilon_i)}. \tag{267}$$

$\square$

**Lemma I.3.** *Let $F$ be a strictly $\lambda$-monotone mapping and $z^*$ be the unique solution to the $\lambda$-weighted variational inequality:*

$$\langle z^* - z, F(z^*) \rangle_\lambda \leq 0, \forall z \in \Delta \tag{268}$$

*in some closed convex set $\Delta$, let* Proj *denote the Euclidean projection onto $\Delta$, we have that:*

$$\mathrm{Proj}(z^* - \eta \Lambda F(z^*)) = z^*, \tag{269}$$

*where $\Lambda = \mathrm{Diag}(\lambda)$.*

*Proof.* Recall that Euclidean projection $\hat{z}$ of $z^* - \eta \Lambda F(z^*)$ is characterized by:

$$\langle z^* - \eta \Lambda F(z^*) - \hat{z}, z - \hat{z} \rangle \leq 0, \forall z \in \Delta \tag{270}$$

Notice that the optimality condition of $z_t^*$:

$$\langle z^* - z, F(z^*) \rangle_\lambda = (z^* - z)^T \Lambda F(z^*) \leq 0, \forall z \in \Delta \tag{271}$$

would immediately imply that taking $\hat{z} = z^*$ satisfies the characterization, conclude that $z^*$ is the projection. $\square$

**Lemma I.4.** *If the payoff matrix $\mathbf{R}$ is bounded and the regularizers $\nu_i(\cdot)$ are $S_\nu$-smooth and $D_i(p_i, \pi_{-i})$ are $S_D$-smooth, then the gradient operator given by:*

$$F(z; \mathbf{R}) = \begin{bmatrix} -R_1 p_1 + \epsilon_1 \nabla \nu_1(\pi_1) \\ -R_2 p_2 + \epsilon_2 \nabla \nu_2(\pi_2) \\ R_1^T \pi_1 + \frac{1}{\tau_1} \nabla_{p_1} D_1(p_1, \pi_2) \\ R_2^T \pi_2 + \frac{1}{\tau_2} \nabla_{p_2} D_2(p_2, \pi_1) \end{bmatrix} \tag{272}$$

*is $L_F$-Lipschitz continuous for*

$$L_F = 2\sqrt{|\mathcal{A}_1||\mathcal{A}_2|} \mathrm{sp}(\mathbf{R}) + (\epsilon_1 + \epsilon_2) S_\nu + \left(\frac{1}{\tau_1} + \frac{1}{\tau_2}\right) S_D. \tag{273}$$

*Proof.* To prove the Lipschitz continuity of $F$, we consider the difference between $F(z; \mathbf{R})$ and $F(z'; \mathbf{R})$ for $z = (\pi_1, \pi_2, p_1, p_2)$ and $z' = (\pi_1', \pi_2', p_1', p_2')$. We have:

$$
\begin{aligned}
&\|F(z; \mathbf{R}) - F(z'; \mathbf{R})\|_2 \\
&= \left\| \begin{bmatrix} -R_1(p_1 - p_1') + \epsilon_1(\nabla\nu_1(\pi_1) - \nabla\nu_1(\pi_1')) \\ -R_2(p_2 - p_2') + \epsilon_2(\nabla\nu_2(\pi_2) - \nabla\nu_2(\pi_2')) \\ R_1^T(\pi_1 - \pi_1') + \frac{1}{\tau_1}(\nabla_{p_1}D_1(p_1, \pi_2) - \nabla_{p_1}D_1(p_1', \pi_2')) \\ R_2^T(\pi_2 - \pi_2') + \frac{1}{\tau_2}(\nabla_{p_2}D_2(p_2, \pi_1) - \nabla_{p_2}D_2(p_2', \pi_1')) \end{bmatrix} \right\|_2 \\
&\leq \left\| \begin{bmatrix} -R_1(p_1 - p_1') \\ -R_2(p_2 - p_2') \\ R_1^T(\pi_1 - \pi_1') \\ R_2^T(\pi_2 - \pi_2') \end{bmatrix} \right\|_2 + \left\| \begin{bmatrix} \epsilon_1(\nabla\nu_1(\pi_1) - \nabla\nu_1(\pi_1')) \\ \epsilon_2(\nabla\nu_2(\pi_2) - \nabla\nu_2(\pi_2')) \\ \frac{1}{\tau_1}(\nabla_{p_1}D_1(p_1, \pi_2) - \nabla_{p_1}D_1(p_1', \pi_2')) \\ \frac{1}{\tau_2}(\nabla_{p_2}D_2(p_2, \pi_1) - \nabla_{p_2}D_2(p_2', \pi_1')) \end{bmatrix} \right\|_2 \\
&\overset{(i)}{\leq} 2\sqrt{|\mathcal{A}_1||\mathcal{A}_2|}\mathrm{sp}(\mathbf{R})\|z - z'\|_2 + \epsilon_1\|\nabla\nu_1(\pi_1) - \nabla\nu_1(\pi_1')\|_2 + \epsilon_2\|\nabla\nu_2(\pi_2) - \nabla\nu_2(\pi_2')\|_2 \\
&\quad + \frac{1}{\tau_1}\|\nabla_{p_1}D_1(p_1, \pi_2) - \nabla_{p_1}D_1(p_1', \pi_2')\|_2 + \frac{1}{\tau_2}\|\nabla_{p_2}D_2(p_2, \pi_1) - \nabla_{p_2}D_2(p_2', \pi_1')\|_2 \\
&\leq 2\sqrt{|\mathcal{A}_1||\mathcal{A}_2|}\mathrm{sp}(\mathbf{R})\|z - z'\|_2 + \epsilon_1 S_\nu\|\pi_1 - \pi_1'\|_2 + \epsilon_2 S_\nu\|\pi_2 - \pi_2'\|_2 \\
&\quad + \frac{S_D}{\tau_1}\|(p_1, \pi_2) - (p_1', \pi_2')\|_2 + \frac{S_D}{\tau_2}\|(p_2, \pi_1) - (p_2', \pi_1')\|_2 \\
&\leq \left( 2\sqrt{|\mathcal{A}_1||\mathcal{A}_2|}\mathrm{sp}(\mathbf{R}) + (\epsilon_1 + \epsilon_2)S_\nu + (\frac{1}{\tau_1} + \frac{1}{\tau_2})S_D \right) \|z - z'\|_2,
\end{aligned}
\tag{274}
$$

where we have used the fact that $\pi_i, p_i$ are on the simplex in (i). This completes the proof. $\square$

**Lemma I.5.** *[Covariance bound for $\alpha$-mixing sequences, Lemma 1.2 in ([Ibragimov, 1962](#))] Let $U$ and $V$ be bounded random variables such that $U$ is measurable with respect to $\mathcal{F}_{-\infty}^t$ and $V$ with respect to $\mathcal{F}_{t+k}^{+\infty}$. Then*

$$
\big| \mathrm{Cov}(U, V) \big| \leq 4\,\alpha(k)\,\|U\|_\infty\,\|V\|_\infty,
\tag{275}
$$

*where*

$$
\alpha(k) = \sup_{t \in \mathbb{Z}} \sup_{A \in \mathcal{F}_{-\infty}^t, B \in \mathcal{F}_{t+k}^{+\infty}} |\Pr(A \cap B) - \Pr(A)\Pr(B)|
\tag{276}
$$

$\alpha(k)$ *is the $\alpha$-mixing coefficient at lag $k$.*

**Lemma I.6.** *[([Meyn & Tweedie, 2012](#))] If a Markov chain $\{X_t\}$ is uniformly geometrically ergodic, then the chain is geometrically $\alpha$-mixing with $\alpha(k) \leq 2C\rho^k$.*

**Lemma I.7.** *For real numbers $a, b$ and constant $c > 0$, it holds that:*

$$
(a + b)^2 \leq (1 + c)a^2 + (1 + \frac{1}{c})b^2.
\tag{277}
$$

*Furthermore, for real matrices $A$ and $B$, it holds that:*

$$
\|A + B\|_{\max}^2 \leq (1 + c)\|A\|_{\max}^2 + (1 + \frac{1}{c})\|B\|_{\max}^2.
\tag{278}
$$

*Proof.* We have that

$$
\begin{aligned}
(a + b)^2 &= a^2 + 2ab + b^2 \\
&= a^2 + b^2 + 2(\sqrt{c}a \cdot \frac{b}{\sqrt{c}}) \\
&\overset{(i)}{\leq} (1 + c)a^2 + (1 + \frac{1}{c})b^2
\end{aligned}
\tag{279}
$$

where (i) comes from the AM-GM inequality. The inequality for the matrix holds by applying the above inequality element-wise:

$$
(A_{ij} + B_{ij})^2 \leq (1 + c)A_{ij}^2 + (1 + \frac{1}{c})B_{ij}^2 \leq (1 + c)\|A\|_{\max}^2 + (1 + \frac{1}{c})\|B\|_{\max}^2.
\tag{280}
$$

and the result holds because the inequality above holds for all $i, j$. $\square$

**Lemma I.8.** *Consider a series $\{w_t\}_{t \geq 0}$ satisfying the following inequality:*

$$w_{t+1} \leq (1 - k\alpha_t)w_t + C\alpha_t^2 \tag{281}$$

*for some $k > 0$ such that $k\alpha_t < 1$, we have for all $t$:*

$$w_t \leq (1 - k\alpha)^t w_0 + \frac{C}{k}\alpha, \tag{282}$$

*if we use constant step size $\alpha_t = \alpha$, and*

$$w_t \leq \left(w_0 + \frac{C\alpha^2}{h^2} \frac{h + 2 - k\alpha}{1 - k\alpha}\right)\left(\frac{h + 1}{h + t}\right)^{k\alpha}, \tag{283}$$

*if we use diminishing step sizes $\alpha_t = \frac{\alpha}{t + h}$.*

*Proof.* Unrolling the recursion we obtain:

$$
\begin{aligned}
w_{T+1} &\leq (1 - k\alpha_T)w_T + C\alpha_T^2 \\
&\leq (1 - k\alpha_T)(1 - k\alpha_{T-1})w_{T-1} + C\alpha_T^2 + C(1 - k\alpha_T)\alpha_{T-1}^2 \\
&\leq \ldots \\
&\leq \prod_{t=0}^{T}(1 - k\alpha_t)w_0 + C\sum_{t=0}^{T}\alpha_t^2 \prod_{s=t+1}^{T}(1 - k\alpha_s).
\end{aligned}
\tag{284}
$$

For constant step size $\alpha_t = \alpha$, the inequality above can be simplified as:

$$
\begin{aligned}
w_{T+1} &\leq (1 - k\alpha)^{T+1}w_0 + C\sum_{t=0}^{T}\alpha^2(1 - k\alpha)^{T-t} \\
&= (1 - k\alpha)^{T+1}w_0 + C\alpha^2 \sum_{t=0}^{T}(1 - k\alpha)^t \\
&= (1 - k\alpha)^{T+1}w_0 + C\alpha^2 \frac{1 - (1 - k\alpha)^{T+1}}{k\alpha} \\
&\leq (1 - k\alpha)^{T+1}w_0 + \frac{C}{k}\alpha.
\end{aligned}
\tag{285}
$$

For deminishing step sizes $\alpha_t = \frac{\alpha}{t+h}$, we have:

$$
\begin{aligned}
\prod_{t=0}^{T}(1 - k\alpha_t) &= \prod_{t=0}^{T}(1 - \frac{k\alpha}{t + h}) \\
&= \exp\left(\sum_{t=0}^{T}\log(1 - \frac{k\alpha}{t + h})\right) \\
&\leq \exp\left(-k\alpha \sum_{t=0}^{T}\frac{1}{t + h}\right) \\
&\leq \exp\left(-k\alpha(H_{h+T} - H_{h-1})\right) \\
&\leq \left(\frac{h}{h + T + 1}\right)^{k\alpha}
\end{aligned}
\tag{286}
$$

and similarly,

$$
\begin{aligned}
\prod_{s=t+1}^{T} (1 - k\alpha_s) &= \exp\left( \sum_{s=t+1}^{T} \log(1 - \frac{k\alpha}{s+h}) \right) \\
&\leq \exp\left( -k\alpha \sum_{s=t+1}^{T} \frac{1}{s+h} \right) \\
&\leq \exp\left( -k\alpha(H_{h+T} - H_{h+t}) \right) \\
&\leq \left( \frac{h+t+1}{h+T+1} \right)^{k\alpha}.
\end{aligned}
\tag{287}
$$

Therefore, the bound for $w_{T+1}$ can be simplified as:

$$
\begin{aligned}
w_{T+1} &\leq \left( \frac{h}{h+T+1} \right)^{k\alpha} w_0 + C\alpha^2 \sum_{t=0}^{T} \frac{1}{(t+h)^2} \left( \frac{h+t+1}{h+T+1} \right)^{k\alpha} \\
&= \left( \frac{h}{h+T+1} \right)^{k\alpha} w_0 + C\alpha^2 \frac{1}{(h+T+1)^{k\alpha}} \sum_{t=0}^{T} \frac{(t+h+1)^{k\alpha}}{(t+h)^2} \\
&\leq \left( \frac{h}{h+T+1} \right)^{k\alpha} w_0 + C\alpha^2 \frac{1}{(h+T+1)^{k\alpha}} \left( \frac{h+1}{h} \right)^2 \sum_{t=0}^{T} (t+h+1)^{k\alpha-2}
\end{aligned}
\tag{288}
$$

Notice that:

$$
\sum_{t=0}^{T} (t+h+1)^{k\alpha-2} \leq (h+1)^{k\alpha-2} + \int_{h+1}^{h+T+2} x^{k\alpha-2}\, \mathrm{d}x \leq (h+1)^{k\alpha-2} + \frac{(h+1)^{k\alpha-1}}{1-k\alpha}
\tag{289}
$$

we obtain the final bound of:

$$
\begin{aligned}
w_{T+1} &\leq \left( \frac{h}{h+T+1} \right)^{k\alpha} w_0 + C\alpha^2 \frac{1}{(h+T+1)^{k\alpha}} \left( \frac{h+1}{h} \right)^2 \left( (h+1)^{k\alpha-2} + \frac{(h+1)^{k\alpha-1}}{1-k\alpha} \right) \\
&\leq \left( w_0 + \frac{C\alpha^2}{h^2} \frac{h+2-k\alpha}{1-k\alpha} \right) \left( \frac{h+1}{h+T+1} \right)^{k\alpha}
\end{aligned}
\tag{290}
$$

$\square$

