# OpenReview forum: "Provably Convergent Actor-Critic for MARL through Risk-aversion"
_ICML.cc/2026/Conference — ICML 2026 spotlight_

### Official Review · Reviewer_zHAJ · 2026-02-22

**Soundness:** 3
**Presentation:** 3
**Significance:** 4
**Originality:** 3
**Overall Recommendation:** 5
**Confidence:** 4

**Summary:**

This paper focuses on Multi-Agent Reinforcement Learning (MARL). Previous works in MARL often struggle to guarantee stable convergence in general-sum Markov games. This paper, however, introduces the concept of Risk-averse Quantal Response Equilibrium (RQE). It transforms the RQE problem into finding the Nash Equilibrium of an extended game with strong monotonicity properties, and establishes a novel two-timescale Actor-Critic algorithm as the solution with finite-sample global convergence guarantees.

**Compliance With Llm Reviewing Policy:**

Affirmed.

**Final Justification:**

This work in RQE provides novel perspective, solid analysis, and satisfying result. Both its significance and originality contribute to my final recommendation.

The rebuttal is clear and helpful, answered my main concerns.

**Key Questions For Authors:**

I really enjoyed reading this paper, and here are a few questions I would like the authors to clarify:

1. As Risk-averse Quantal response Equilibria (RQE) fundamentally is different from traditional Nash Equilibria, is there any theoretical bound between these two (like the distance or payoff gap) under certain conditions?

2. How to understand the condition $16 \cdot \epsilon_1 \epsilon_2 \cdot \tau_1 \tau_2 > 1$ intrinsically? Is the behavior of agents analyzable beyond this condition (e.g., would the system exhibit periodic patterns or limit cycles if this condition is violated)?

3. What is the computational cost of this novel two-timescale Actor-Critic algorithm with respect to the number of agents / update step?

**Limitations:**

yes

**Strengths And Weaknesses:**

Strengths

Originality:

The formulation of Risk-averse Quantal Response Equilibria (RQE) is highly novel. By jointly integrating entropy regularization and risk aversion into a single solution concept, the authors provide a fresh and creative perspective on behavioral game theory in Multi-Agent Reinforcement Learning (MARL). The tailored two-timescale Actor-Critic algorithm designed specifically to exploit the regularity of RQE is also a highly original contribution.

Significance:

The paper makes a substantial and pioneering contribution to the field by addressing the pervasive issue of unavoidable instability in infinite-horizon general-sum Markov Games. Establishing a provable path to convergence in this notoriously difficult setting is of great significance and could inspire further research into alternative, learnable equilibrium concepts beyond standard Nash formulations.

Soundness:

The theoretical foundation of the paper is rigorous and very solid, with strong proofs supporting the global convergence of the proposed two-timescale algorithm. Furthermore, the empirical results convincingly corroborate the theoretical claims, demonstrating stable performance and clear advantages over existing baselines.

Weaknesses

Presentation & Significance (Theoretical Contextualization):

A major conceptual gap in the current manuscript is the lack of a thorough discussion regarding the relationship between the newly introduced Risk-averse Quantal Response Equilibria (RQE) and the classical Nash Equilibria (NE). Since RQE is fundamentally different from NE, readers and practitioners need to understand the theoretical trade-offs being made to achieve global convergence. The paper needs to explicitly discuss the theoretical gap (like under what conditions does RQE approximate a Nash Equilibrium? What is the performance bound or suboptimality when deploying an RQE policy in a true general-sum game compared to an exact NE?) Without this discussion, it is difficult to fully assess the broader significance of converging to an RQE.

---

> ### Author Rebuttal · Authors · 2026-03-28
>
> We thank the reviewer for carefully reading the paper and recognizing our contributions. We are extremely excited that you enjoyed reading this paper! We address the concerns and answer the questions below:
>
> ### Strengths and Weaknesses
> > *"A major conceptual gap in the current manuscript is the lack of a thorough discussion regarding the relationship between the newly introduced RQE and the classical NE. Since RQE is fundamentally different from NE, readers and practitioners need to understand the theoretical trade-offs being made to achieve global convergence. The paper needs to explicitly discuss the theoretical gap ... "*
>
> You are absolutely correct to perceive that our solution concept of RQE is fundamentally different from NE. However, we do not view the switch from RQE to NE as a performance-tractability trade-off, as RQE may itself be a more desirable property in real-world problems than NE. This is because it not only captures risk-aversion as a notion for robustness to different strategies of other players (which is true under most real-world multi-agent environments), but also captures bounded rationality and risk-aversion from behavioral economics, implying its ability to model real-world human behavior as well, compared to the idealized perfect rationality assumption in NE. For performance comparison between RQE and NE, we would like to point to a recent concurrent work Qu et al., (2026), which demonstrates when and how RQE can be more desirable than NE. In essense they show that RQE can actually yeild higher payoffs for players than Nash (something that is only possible in games and not single-agent regeims where robustness introduces sub-optimzality).
>
> Qu et al., (2026): Training Generalizable Collaborative Agents via Strategic Risk Aversion
>
>
> ### Key Questions for Authors
> > *Answer to Q1:*
>
> We note that since RQE captures risk-aversion in decision-making, it may be very different from NE, especially when the game structure itself involves high risk. To illustrate this, consider the pedestrian(column)-car(row) game example:
>
> | Utility | Wait | Cross |
> | -------- | -------- | -------- |
> | Keep speed   | (10,0)     | (-50,-100)     |
> | Slow down    | (8,0)     | (-10,-10)     |
> | Stop    | (0,0)     | (0,10)     |
>
> The Nash equilibria here are (Keep speed, Wait) and (Stop, Cross) but since the cell (Keep speed, Cross) has a very low payoff, in this case RQE will be different Nash because RQE considers possible deviations of opponent strategies. For example, depending on the level of risk-aversion, the RQE could be a mixture between (Slow down, Wait) and (Stop, Wait). If we set the payoffs of (Keep speed, Cross) to approach $-\infty$, we can see that NEs stays unchanged but neither of them can be RQE, as arbitrarily small deviation from NE could lead to $-\infty$ utility.
>
> > *Answer to Q2:*
>
> With proper regularizer pairs, the condition $16 \epsilon_1\epsilon_2\tau_1\tau_2>1$ is indeed necessary for the 4-player game to be strictly monotone. Since larger $\epsilon$ and larger $\tau$ means players are less rational and more risk-averse, the condition means when both players are not too rational and are risk-averse enough, RQE is tractable. While not satisfying strict monotonicity does not directly rule out the possibility of the game still being tractable (there are many such game instances), tractability for *general* games without monotonicity is hard (provably so since computing Nash and stationary CE/CCE is in PPAD) , and the learning dynamics can cycle even when the game is regularized in some other way, as shown in Mertikopoulos et al., (2017).
>
> > *Answer to Q3:*
>
> The computational cost of the algorithm depends highly on the implementation. In principle, if we simply just use tabular actor and critic, the space complexity of the critic (the dimension of the Q function) will be $O(|\mathcal{S}|\times |\mathcal{A}|^n)$ for each agent, which results in a total space complexity of $O(n|\mathcal{S}|\times |\mathcal{A}|^n)$, which is clearly not scalable (this is also known as the curse of multiagency). However, since in practice we use neural representations for both actor and critic, the overall complexity will be entirely dependent on the size of the networks. If we keep these sizes as constants, the overall complexity will become a highly scalable $O(n)$. Also, we note that compared to the standard multi-agent actor-critic algorithm, the only difference in computational complexity is in the actor update, where we additionally update an adversary for each agent. Notice that we only keep a shared single Q function for each agent-adversary pair, the critic update complexity stays the same, meaning that our method empirically shares a similar complexity to standard actor-critic.

---

> > ### Author Rebuttal · Reviewer_zHAJ · 2026-04-01
> >
> > Thanks for this thoughtful, clear, and highly convincing rebuttal. The pedestrian-car game example provided in the response to Q1 is exceptionally illustrative. The explanations regarding the intrinsic meaning of the $16\epsilon_1\epsilon_2\tau_1\tau_2 > 1$ condition (Q2) and the practical scalability of the neural implementation (Q3) are also very clear and satisfactory.
> >
> > The rebuttal further solidified my appreciation for this work, thus I am happily maintaining my positive score.

---

> > > ### Author Response · Authors · 2026-04-05
> > >
> > > Thank you very much for the thoughtful follow-up and for your careful reading of our rebuttal. We are truly grateful for your encouraging comments, and are also very glad that the pedestrian-car example and the additional clarifications helped address your questions clearly.
> > >
> > > We also sincerely appreciate your positive assessment of the paper’s originality, significance, and technical foundation. Your feedback has been very valuable in helping us better present the motivation and broader interpretation of the work. If there are any further questions or additional points, we would be very happy to clarify them.
> > >
> > > Thank you again for your time, feedback, and consideration.

---

### Official Review · Reviewer_AmU8 · 2026-03-03

**Soundness:** 4
**Presentation:** 3
**Significance:** 4
**Originality:** 4
**Overall Recommendation:** 4
**Confidence:** 2

**Summary:**

This paper tackles the computationally intractable problem of learning stationary policies in general-sum Markov Games by studying Risk-averse Quantal response Equilibria (RQE). RQE, which incorporates bounded rationality and risk aversion, makes these games mathematically solvable without requiring special game structures. Following that, this paper proposes a two-timescale Actor-Critic algorithm to compute RQE, and theoretically analyzes its global convergence in finite-time guarantees. Numerical experiments  further confirm the advancement of the proposed algorithm.

**Compliance With Llm Reviewing Policy:**

Affirmed.

**Final Justification:**

The response is detailed. I have decided to maintain the positive assessment.

**Key Questions For Authors:**

Two questions are in order:

* The authors claim that the necessity of storing history to find non-stationary equilibria substantially increases complexity. I wonder: could the techniques from POMDP-based RL be adapted here to replace the explicit history storage with more efficient surrogate methods? If so, this may lead to interesting results.

* The numerical experiments are considerably weaker than the theoretical component. All three experiments are conducted on a small scale. Providing larger-scale numerical evidence, such as an experiment involving Large Language Models (LLMs), would significantly enhance the persuasiveness of the work.

**Limitations:**

No additional limitations should be listed here in my opinion.

**Strengths And Weaknesses:**

**Strength:** The theoretical foundation of this paper is solid, and it contributes to the literature.

**Weakness 1:** A point of concern is that the authors emphasize the highly limited nature of existing works to the two-player setup; however, the main theoretical sections of this paper continue to examine the two-player case “for simplicity”. The detailed extension to the multi-player setting should be clarified.

**Weakness 2:** Another issue is that while the theoretical framework of this paper is solid, it lacks sufficient intuition. The presentation is dry, and the inclusion of illustrative figures could potentially address this limitation.

---

> ### Author Rebuttal · Authors · 2026-03-28
>
> We thank the reviewer for recognizing the soundness, significance, and originality of our work, and raising highly valuable further directions that could strengthen our work even more. Below we address the concerns and answer the questions:
>
> ### Strengths and Weaknesses
> > *Response to Weakness 1:*
>
> We first clarify that although we only present our result for the two-player case, our proof outline would generalize to an arbitrary $n$-player case. For Theorem 3.1, the adapted result would change $\sqrt{|\mathcal{A}_1|}+\sqrt{|\mathcal{A}_2|}$ to $\sum_i\sqrt{|\mathcal{A}_i|}$, through conducting the same analysis in Appendix F.1 on the expanded gradient operator $F$ from $2$ to $n$ players. The monotonicity condition for the $2n$-player game still won't rely on payoff matrices, and if we further assume the regularizers $D_i(p\_i,\pi\_{-i})$  is in the form of $\sum\_{j\neq i} D\_{i,j}(p\_{i,j}, \pi\_j)$ where each agent individually model $n-1$ adversaries each corresponding to one opponent, the matrix in part 1 of Theorem 3.2 would become:
>
> $$
> \begin{bmatrix}
>     2\lambda\_i\epsilon\_i \nabla^2\nu\_i & \left[ \frac{\lambda\_k}{\tau\_k}\nabla^2_{p\_{k,i},\pi\_i}D\_{k,i} \right]^T_{k\neq i} \\\\
>     \left[ \frac{\lambda\_k}{\tau\_k}\nabla^2_{p\_{k,i},\pi\_i}D\_{k,i} \right]\_{k\neq i} & \text{diag}\left( \left[ 2\frac{\lambda\_k}{\tau\_k}\nabla\_{p\_{k,i}}^2 D\_{k,i} \right]\_{k\neq i} \right)
> \end{bmatrix}
> $$
>
> and assume all $D_{i,j}$ are the same and $\nu_i, D_{i,j}$ are either log-barrier/KL or negative entropy/reverse KL, the condition $16\epsilon_1\epsilon_2\tau_1\tau_2>1$ generalizes to $\sum_{i=1}^n \frac{1}{1+4\epsilon_i\tau_i}<1$ (this is indeed equivalent to $16\epsilon_1\epsilon_2\tau_1\tau_2>1$ when $n=2$ by calculation).
>
> All results in Section 4 do not rely specifically on the two-player structure given the assumptions, with the only difference being that in the detailed versions, all $\sqrt{|\mathcal{A}_1|}+\sqrt{|\mathcal{A}_2|}$ will be changed to $\sum_i\sqrt{|\mathcal{A}_i|}$.
>
> We will state these explicitly as well as the complete proofs in the final version.
>
> > *Response to Weakness 2:*
>
> We acknowledge that, due to the 8 page limit, we compressed the intuition of our RQE framework in order to present our main contributions. We will use the extra page in the final version to add more discussion on intuition behind our approach and the theorem proofs. For interpretations in preliminaries, please refer to our response to reviewer PGdw for more details. We will include the pedestrian-car example with figures in our response to reviewer zHAJ in the paper as an illustration.
>
>
> ### Key Questions for Authors
> > *Answer to Q1:*
>
> We would like to clarify that our approach has the merit of provably learning a stationary equilibria where all policies only depend on the current state. In comparison, even with full information, most existing methods (for example, the V-learning framework by Jin et al., (2021) and Liu et al., (2021)) only have provable guarantees to converge to non-stationary correlated equilibria (CE) and coarse correlated equilibria (CCE).
>
> To the best of our knowledge, the only existing theoretical framework studying game-theoretic setting with partial observability in MARL is Liu & Zhang (2023), which also studied convergence to CE and CCE, both being non-stationary equilibria depending on history. We believe it remains a largely open yet interesting future direction whether partial observability leads to less history dependence.
>
>
> Liu & Zhang (2023): Partially Observable Multi-Agent Reinforcement Learning with Information Sharing
>
> > *Answer to Q2:*
>
> We keep our environments and algorithms simple in order to verify the high-level intuition brought by our theoretical insights that risk-aversion helps convergence in MARL. While our work does not include LLM experiments, we note that a recent concurrent work Qu et al., (2026) includes experiments using the same RQE framework on fine-tuning LLMs, despite their work being more empirically-focused and used a PPO-based algorithm, while we focus more on proving the theoretical convergence of our proposed actor-critic algorithm.
>
> We additionally evaluate MAPPO and MADDPG on gridworld and tag, whose plots can be found in https://anonymous.4open.science/r/Provably_Convergent_Actor_Critic_Risk-averse_MARL-2F41. We can see that for gridworld, MAPPO diverge into two equilibria, while MADDPG has much worse performance and even fails to converge. In comparison, our risk-averse AC uniformly converges to the unique RQE. For tag, both MAPPO and MADDPG training curves has lower final MA100 reward and larger variance compared to risk-averse AC.
>
> | Algorithm | Risk-averse AC | MAPPO | MADDPG
> | -------- | -------- | -------- | -------- |
> | Simplementing Tag Reward   |   $36.74 \pm 2.65$   |  $35.50\pm 4.91$  | $30.20\pm 9.53$    |
>
>
> Qu et al., (2026): Training Generalizable Collaborative Agents via Strategic Risk Aversion

---

> > ### Author Rebuttal · Reviewer_AmU8 · 2026-03-31
> >
> > Thanks for the detailed response and the new experimental results. My primary concerns are resolved, and I decide to maintain my positive score.

---

> > > ### Author Response · Authors · 2026-04-05
> > >
> > > Thank you very much for the thoughtful follow-up and for your careful reading of our rebuttal. We are very grateful that our response, including the additional experimental results, addressed your primary concerns.
> > >
> > > We also sincerely appreciate your positive assessment of the paper’s theoretical foundation, significance, and originality, as well as your constructive feedback throughout the discussion. Your comments have been very helpful in improving how we present the work and its broader motivation. If there are any further concerns, questions or additional points, we would be more than happy to clarify them.
> > >
> > > Thank you again for your time and consideration.

---

### Official Review · Reviewer_PGdw · 2026-03-10

**Soundness:** 3
**Presentation:** 3
**Significance:** 3
**Originality:** 3
**Overall Recommendation:** 4
**Confidence:** 3

**Summary:**

The authors assess a central concept: risk-averse quantal response equilibrium (RQE) as a tractable stationary solution notion for infinite-horizon general-sum Markov games. Overall, this paper's central contribution concerns showing that, under behavioral regularization assumptions, RQE becomes unique and smooth enough to induce a contractive Bellman operator, which then supports a reverse-timescale actor-critic method with a fast actor, slow critic, and finite-sample convergence guarantees. The experimental section mainly argues that the risk-averse variant is more stable than its risk-neutral counterpart.

**Compliance With Llm Reviewing Policy:**

Affirmed.

**Final Justification:**

The author's rebuttal helped me to develop the further confidence on the evaluation.

**Key Questions For Authors:**

1. How restrictive is Assumption 4.1 in practice? In particular, for the tau, epsilon settings used in the experiments, can the authors verify that the sufficient monotonicity conditions really hold?

2. What should the reader conclude about Algorithm 2? Is it intended as a heuristic inspired by the theorem, or do the authors believe some part of the convergence argument extends to function approximation?
3. The experiments mainly compare against the risk-neutral version of the same method. Why is that the right empirical baseline? Even one or two standard MARL baselines on the small environments would make the practical claim much stronger.

**Limitations:**

yes

**Strengths And Weaknesses:**

Soundness:
The technical arc is strong. The paper gives a clean bridge from the 4-player reformulation to RQE, weakens prior monotonicity conditions via lambda-monotonicity, and derives both convergence of the two-timescale iteration and finite-sample guarantees for the tabular actor-critic. The reverse-timescale idea is interesting and well motivated.
My main reservation is that the guarantees rely on fairly strong assumptions: stage-wise (mu, lambda)-strong monotonicity, smooth regularizers, contraction of the Bellman operator, irreducibility / geometric ergodicity, and uniformly lower-bounded sampling policies. These assumptions are not hidden, but they are doing a lot of work, so the practical scope of the theorem is narrower than the headline may suggest.

Presentation:
The paper is organized and the main message is understandable, but the notation is heavy. The jump between the original 2-player game, the 4-player construction, and the Markov-game Bellman view takes effort to track.
I also think the practical status of Algorithm 2 is a bit underspecified relative to the theory: the theorems are for the tabular setting, while the experiments use the neural implementation. That gap is normal, but it should be stated more plainly.

Significance:
Provable convergence to a stationary equilibrium notion in discounted general-sum Markov games is genuinely hard, and the paper gives a coherent route by changing the equilibrium concept rather than the game class.
On the empirical side, the impact is less clear. The experiments are small-scale and the main comparison is to the risk-neutral version of the same framework, so the results support the paper’s mechanism story more than they establish broad practical superiority.

Originality:
The main novelty is not just “risk-sensitive MARL,” but the combination of behavioral-game-theoretic regularization, monotone-game analysis, Bellman contraction, and a fast-actor/slow-critic proof strategy. That combination feels fresh and technically nontrivial.

---

> ### Author Rebuttal · Authors · 2026-03-28
>
> We would first like to thank the reviewer for the careful reading and thoughtful review, as well as the strong recognition of our conceptual and technical contributions. We address the concerns and answer the questions below:
>
> ### Strengths and Weaknesses
>
> #### Soundness
> > *"My main reservation is that the guarantees rely on fairly strong assumptions ..."*
>
> We would like to clarify that the assumptions listed above are both standard and largely necessary for our analysis:
> * Strong monotonicity, as studied in Section 3, depends only on the risk-aversion and bounded rationality levels of each player, instead of the $\mathbf{Q}$ functions in each stage game itself. This implies that no matter what $\mathbf{Q}$ is in an arbitrary iteration of the algorithm, once the risk and regularization of each player is determined, the strong monotonicity assumption always holds.
> * Contraction of the Bellman operator (Prop 4.2) is a derived result from monotonicity instead of an assumption, and is central for proving Theorems 4.4 and 4.7. We again emphasize that contraction is not an assumption on the game structure.
> * Irreducibility and geometric ergodicity, as well as the uniformly lowre-bounded sampling policies guarantee that each state-action pair of the Markov game is sufficiently explored, and is usually required in stochastic approximation and actor-critic literature (e.g. Chen et al., (2022b), Wu et al., (2020), Zhang et al., (2020)).
> * Smoothness of the regularizers is required for analyzing the projected gradient descent step using direct parameterization, and holds for common regularizers such as $\ell_p$ norm, (reverse) KL divergence, log-barrier, and negative entropy in the strict interior of the simplex.
>
> #### Presentation
> > *"... the notation is heavy...I also think the practical status of Algorithm 2 is a bit underspecified relative to the theory..."*
>
> Due to the strict 8 page limit, we condensed the thorough intuitive explanation and connection between different views of RQE in order to better present our main theoretical contributions. We will use the extra page in the final version to include more specification for the preliminaries.
>
> Intuitive interpretations for RQE: First, the objective function for each agent is the value of a minimax optimization problem, so directly learning RQE as in Definition 2.1 involves solving a series of bilevel optimization problems, which is hard. Luckily, Proposition 2.2 tells us that we can introduce two adversaries and directly solve for a Nash equilibrium for the 4-player game instead. More importantly, the 4-player game has a zero-sum structure between each agent-adversary pair, making its Nash equilibrium easy to solve regardless of the payoff matrices. For the Markov-game Bellman view, we imagine at each time step, instead of dealing with multiple future timesteps, each player only plays a normal-form *stage game* whose payoff matrices are the current $\mathbf{Q}$ functions. Proposition 2.6 suggests that by doing this combined with the Bellman backups, the equilibrium that the updates converge to is precisely the RQE defined by raw Markov game objectives (7) as in Definition 2.4.
>
> Regarding Algorithm 2, please see the answer to Q2.
>
> #### Significance
> > *"On the empirical side, the impact is less clear ... "*
>
> Please see the answer to Q3.
>
>
> ### Key Questions for Authors
> > *Answer to Q1:*
>
> Notice that by Theorem 3.2, Assumption 4.1 boils down only to the risk-aversion and bounded rationality levels of each player. Let's take $\epsilon_1\epsilon_2\tau_1\tau_2>\frac{1}{16}$ as an example, it holds when $\epsilon_1=\epsilon_2=0.2$ and $\tau_1=\tau_2=5$, this is satisfied in our gridworld cooperation game experiment. For the MPE simple tag experiment, our parameters don't satisfy this condition, but Algorithm 2 still provides more consistent convergence behavior, indicating that in practice we may require even less risk-aversion and bounded rationality than what is theoretically needed.
>
> > *Answer to Q2:*
>
> We view Algorithm 2 as a practical scalable implementation inspired by Algorithm 1 that uses common RL tricks. Since neural networks are highly non-convex, it is generally not possible to give an exact theoretical guarantee for Algorithm 2. However, we believe it is possible to adapt our theoretical result to simple cases such as using linear function approximation for critic, and adopting softmax parameterization for the actor, as long as contraction still holds. We leave the formal theoretical characterization to future work.
>
> > *Answer to Q3:*
>
> Our goal is to motivate the high-level idea that risk-aversion helps convergence in MARL. Based on this, we design our experiments highlight the role of risk-aversion itself, while de-emphasizing various practical tricks, so for fair comparison, we keep the remainder of the algorithm exactly the same. We implement MAPPO and MADDPG as baselines, please see our response to reviewer AmU8 for more details.

---

> > ### Author Rebuttal · Reviewer_PGdw · 2026-04-02
> >
> > Thank you very much for the detailed clarification. I will increase the confidence to 3.

---

> > > ### Author Response · Authors · 2026-04-05
> > >
> > > Thank you very much for the thoughtful follow-up and for your careful reading of our rebuttal. We are very grateful that our clarifications addressed your concerns.
> > >
> > > We also greatly appreciate your constructive feedback and your encouraging assessment of the paper’s novelty and technical contributions. Your comments have been very helpful in improving how we present the work. If there are any further questions or additional points, we would be very happy to clarify them.
> > >
> > > Thank you again for your time and consideration.

---

### Official Review · Reviewer_RL4V · 2026-03-11

**Soundness:** 3
**Presentation:** 4
**Significance:** 3
**Originality:** 3
**Overall Recommendation:** 5
**Confidence:** 2

**Summary:**

To overcome the intractability associated with commonly considered equilibrium concepts, the authors introduce constraints on the agent rather than solving optimally the game itself, leading to the notion of RQE. After showing that the resulting surrogate game provides a lossless reduction of the original RQE problem, the framework is first developed for simple settings such as normal-form games and then extended to Markov games, where an optimal stationary policy may exist. Building on these results, (i.e. the contraction property of the operator) an actor–critic algorithm is proposed and proven to converge at a sublinear rate.

**Compliance With Llm Reviewing Policy:**

Affirmed.

**Key Questions For Authors:**

Do you think this framework could be extended to extensive-form games (EFG) or partially observable stochastic games (POSG)?

**Limitations:**

yes

**Strengths And Weaknesses:**

Strengths:
- Several experiments support the claims made in the paper
- Proof are complete and the paper is well organized
- New proofs are provided or complete already existing work (Prop 2.6, 4.2)

Weaknesses:
- Due to the number of proofs, I was not able to check all of them within the available time.

Typos:
- 1399: p^*_i instead of p_i?

---

> ### Author Rebuttal · Authors · 2026-03-28
>
> We genuinely respect and appreciate the reviewer for the great effort in carefully reading the paper and inspecting the correctness of the results. Please find our responses listed as below:
>
> ### Strengths and Weaknesses
>
> > *"Typos: 1399: $p_i^{\*}$ instead of $p_i$?"*
>
> We thank the reviewer for identifying this typo. We will correct it in the final version and check for the typos in other proofs as well.
>
> > *"Do you think this framework could be extended to extensive-form games (EFG) or partially observable stochastic games (POSG)?"*
>
> We note that there are existing works studying regularized equilibrim for EFGs like McKelvey & Palfrey, (1998) and Liu et al. (2023). Additionally, the original paper Mazumdar et al., (2024) also studied finite-horizon Markov games that are somewhat similar to EFGs. Therefore, we think it is possible to extend our RQE framework to EFGs as well. For POSGs, we definitely think that risk-aversion is intuitively beneficial under partial observability, but it remains less clear whether our notion of *strategically* risk-aversion (being risk-averse against other players' strategies) would directly work for POSGs, because on a high-level, to address partial observability, we should additionally consider the risk-aversion against environment uncertainty. We leave both interesting directions as future work.
>
> Liu et al. (2023): The Power of Regularization in Solving Extensive-Form Games

---

> > ### Author Rebuttal · Reviewer_RL4V · 2026-04-02
> >
> > The authors clarified all the ambiguous points. I keep my current evaluation.

---

> > > ### Author Response · Authors · 2026-04-05
> > >
> > > Thank you very much for reading our rebuttal carefully. We are very grateful that our response resolved the ambiguities you had raised.
> > >
> > > We also appreciate your constructive feedback on the paper, including the note on the typo and the interesting potential extensions. If there are any further questions or additional points during the discussion stage, we would be very happy to clarify them.
> > >
> > > Thank you again for your time and consideration.

---

### Decision · Program_Chairs · 2026-04-30

**Decision:**

Accept (spotlight)

**Comment:**

The paper shows that, under certain assumptions, RQE becomes unique and smooth enough to be solved. Then, they proposed a reverse-timescale actor-critic method with a fast actor, slow critic, and finite-sample convergence guarantees

All the reviewers think that the provable convergence result is an important result, and the paper's techniques are original.